# Single-cell and spatial transcriptome analyses reveal tertiary lymphoid structures linked to tumour progression and immunotherapy response in nasopharyngeal carcinoma

Yang Liu[1,10], Shuang-Yan Ye[2,10], Shuai He [1,10], Dong-Mei Chi [1,3,10], Xiu-Zhi Wang[1,10], Yue-Feng Wen[4], Dong Ma[1], Run-Cong Nie[1], Pu Xiang[1], You Zhou[1], Zhao-Hui Ruan[1], Rou-Jun Peng[1], Chun-Ling Luo [1], Pan-Pan Wei[1], Guo-Wang Lin[5], Jian Zheng [1], Qian Cui[6], Mu-Yan Cai [1], Jing-Ping Yun [1], Junchao Dong[7], Hai-Qiang Mai [1], Xiaojun Xia [1] & Jin-Xin Bei [1,8,9] ✉

Tertiary lymphoid structures are immune cell aggregates linked with cancer outcomes, but their interactions with tumour cell aggregates are unclear. Using nasopharyngeal carcinoma as a model, here we analyse single-cell transcriptomes of 343,829 cells from 77 biopsy and blood samples and spatially-resolved transcriptomes of 31,316 spots from 15 tumours to decipher their components and interactions with tumour cell aggregates. We identify essential cell populations in tertiary lymphoid structure, including CXCL13[+] cancer-associated fibroblasts, stem-like CXCL13[+]CD8[+] T cells, and B and T follicular helper cells. Our study shows that germinal centre reaction matures plasma cells. These plasma cells intersperse with tumour cell aggregates, promoting apoptosis of EBV-related malignant cells and enhancing immunotherapy response. CXCL13[+] cancer-associated fibroblasts promote B cell adhesion and antibody production, activating CXCL13[+]CD8[+] T cells that become exhausted in tumour cell aggregates. Tertiary lymphoid structure-related cell signatures correlate with prognosis and PD-1 blockade response, offering insights for therapeutic strategies in cancers.

Immune cell infiltration is a common pathological observation in cancer, where tumoral, immune, and stromal cells, as well as extracellular components, are assembled to form a tumour microenvironment (TME). The organisation of these components is essential to an effective immunity against tumour[1]. Tertiary lymphoid structure (TLS) represents such a well-organised ectopic compartment of immune cells that develops in inflamed, infected, or tumoral tissues[2,3]. Initially, both T and B cell zones are characterised in TLS using immunohistochemistry (IHC) assays[4]. Subsequently, several studies reveal that TLSs also contain antigen-presenting cells (APCs) as regulators and stromal cells as supportive infrastructures[5], in addition to T follicular helper (Tfh) cells and follicular dendritic cells (FDC) close to B cells[6]. Recently, single-cell RNA sequencing and spatial transcriptomics studies reveal specific cell populations of TLS in tumours, such as CXCL13[+] T-helper cells in ovarian cancer[7], SEMA4A[+] germinal centre (GC) B cells in head and neck squamous cell carcinoma[8], and CXCL12[+]

fibroblasts in renal cell cancer[9]. These findings suggest a complicated and heterogeneous composition of TLS in tumours from different technological perspectives, which await further investigations with larger sample sizes and cutting-edge technologies under higher resolution.

GC is a crucial feature of matured TLSs in tumours and is a well-known compartment for the selective activation and amplification of B cell clones and the production of long-lived plasma cells[6]. Although the coexistence of GC B cells and tumour-infiltrating B or plasma cells has been demonstrated in multiple tumours[9–11], little is known about the dynamic features of GC reaction and developmental trajectories of B cells within the tumours, especially at TLSs. A recent study has demonstrated GC B cells characterised by a continuum of gene expression states spanning the dark-light zone axis[12,13]. Apart from GC reaction, multiple cellular components, including lymphoid tissue inducer cells (LTi), fibroblastic reticular cells (FRC), and FDC[14,15], and molecular processes, including LTβR signalling[16], adhesion molecules (VCAM1 and ICAM1)[17], and lymphoid chemokines (CXCL13, CCL17, and CCL21)[18–20] have been reported to drive TLS formation in autoimmune diseases and chronic infection. Furthermore, previous studies also reveal that fibroblasts and endothelial cells orchestrate TLS formation in mouse tumour models[21,22]. By contrast, the critical components to drive TLS formation are poorly understood in human tumours.

TLSs could process local tumour antigens with colocalized APCs to generate effector memory T, memory B, and plasma cells, which infiltrate into tumours participating in anti-tumour responses[5]. Among these, T cells in TLSs or those in close proximity upregulate the expression of pro-survival and anti-apoptotic molecule BCL-2 to inhibit the growth of melanoma[23]. TLS-associated plasma cells could secrete antibodies, which recognise tumour-associated antigens to promote malignant cells apoptosis through antibody-dependent cellular cytotoxicity (ADCC) and antibody-dependent cellular phagocytosis (ADCP)[9]. These might explain the positive associations of higher TLS presentation with a higher degree of tumour-infiltrating immune cells[24], favourable patient survival, and better responses to immune checkpoint blockade (ICB) therapy in multiple human tumours[23,25,26]. Although the TLS-associated effector T and B cells exhibit anti-tumour phenotypes[9,23], how they intersperse with close spatial proximity to tumour cells and recognise them to execute their anti-tumour activities has not yet been characterised.

Here, we aim to uncover the cellular composition, formation, and function of TLS along with tumour progression and immunotherapy response at single-cell and spatial transcriptomic resolutions using nasopharyngeal carcinoma (NPC) as a model, which is a prevalent malignancy in Southern China with intensive immune cell infiltration[27]. We also evaluate the prognostic value of key TLS components for patient survival and immunotherapy response in multiple NPC cohorts.

## Results

### Landscape of TLS composition at single-cell and spatial resolutions in NPC

Given that CD20$^+$ B-cell follicle juxtaposing to a CD3$^+$ T cell aggregation is the hallmark of TLS in multiple cancers[4], we performed multiplex immunohistochemistry (IHC) staining assays and revealed prominent TLSs comprising B cells (CD20 as the signature maker), CD4$^+$ T cells (CD3, CD4), CD8$^+$ T cells (CD3, CD8), and fibroblasts (FAP-α) in NPC biopsies (Fig. 1a and Supplementary Fig. 1a). To characterise TLSs at single-cell resolution, we performed single-cell RNA sequencing (scRNA-seq) analysis of transcriptome and immune cell receptor profiles for 77 samples, including 56 tumours and 10 peripheral blood mononuclear cells (PBMC) from 56 patients with NPC and 11 nasopharyngeal non-cancerous (NPN) tissues (see Methods; Supplementary Fig. 1b and Data 1). After strict quality control filters (see Methods), 343,829 cells were identified from all the samples, including 221,357,

48,995, and 73,477 cells for tumour tissues, non-cancerous tissues, and PBMC, respectively (Supplementary Fig. 1b). We also identified various cell clusters according to their expression profiles using graph-based clustering implemented in the Seurat package, including 97,360 B cells, 88,009 CD4$^+$ T cells, 95,686 CD8$^+$ T cells, 20,051 NK cells, 21,088 myeloid cells, 1,652 cancer-associated fibroblasts (CAFs), 1,550 endothelial cells, and 15,666 malignant cells (Fig. 1b and Supplementary Fig. 1d-h). Specifically, we identified TLS-associated cell clusters according to the canonical TLS markers, including B cells (CD19, MS4A1, CD79A, and CD79B), T follicular helper (Tfh) cells (CD4_C8_CXCR5; CD4, CXCR5, CXCL13, and BCL6), CD8$^+$ T cells (CD8_C8_CXCL13; CD8A, CXCR5, CXCL13, and BCL6), and CAFs (iCAF_C2_CXCL13; COL1A1, CXCL13, CCL19, and CCL21; Fig. 1c and Supplementary Data 2).

Next, to explore the spatial cell composition and distribution of TLS in situ with a high resolution, we performed spatial transcriptomics (ST) analysis across multiple technological platforms, including 9563 data bins (containing 100×100 = 10,000 data spots; see Methods) from three fresh-frozen NPC primary tumours using Stereo-seq (Stereo-seq cohort) and 21,753 data spots from 12 NPC primary tumours using Visium formalin-fixed paraffin-embedded (FFPE) assay (Visium cohort; Supplementary Data 1). ST analyses and haematoxylin and eosin (H&E) staining assay revealed various locations of TLS, tumour cell aggregates (TCA), and stromal regions with irregular shapes across Stereo-seq and Visium cohorts in NPC tumours (Supplementary Figs. 2–4; see Methods). Meanwhile, ST analyses corroborated conspicuous aggregations of B lineage cells (Naïve B, Memory B, GC B, and plasma cells), CD4_C8_CXCR5 and CD8_C8_CXCL13 T cells, and CXCL13$^+$ CAFs (iCAF_C2_CXCL13) in the TLS regions of NPC tumours across Stereo-seq and Visium cohorts (Fig. 1d, e). Among these cell types, plasma cells demonstrated a variable distribution between the two cohorts, with the highest proportion in the stromal region of the Stereo-seq cohort (Fig. 1e), which is attributed to a high infiltration of plasma cells in one of the three samples examined (Supplementary Fig. 5a). Furthermore, the locations of all these TLS-associated cell clusters were corroborated in another independent NPC sample collection with RNA-Seq data derived from multiple micro-dissected tumour cell aggregates (TCA), immune cell aggregates (ICA), dysplastic epithelium (DYS), normal nasopharyngeal epithelium (NAT), and normal tumour-adjacent epithelium (NOR) regions[28] (Microdissection cohort, $n = 189$; see Methods), with significant enrichment of the TLS-associated cell clusters in ICA region compared with other regions (Supplementary Fig. 5b, c). Together, these observations suggest common spatial localisation of TLS in NPC, with essential components including B lineage cells (Naïve B, Memory B, GC B, and plasma cells), CD4_C8_CXCR5 and CD8_C8_CXCL13 T cells, and CXCL13$^+$ CAFs.

### Developmental trajectory and characteristics of GC B cells along tumour progression in NPC

Immune cell receptor profiling revealed 19,405 B cells with BCR clonotypes, among which 18,675 were unique in all samples (Supplementary Data 3). Higher somatic hypermutation (SHM) frequencies of IGHA1, IGHG1, and IGHG3, reflecting higher affinity of BCRs associated with antigen recognition and activation[29,30], were observed in B cells derived from NPC tissues compared with that from PBMC, as were higher clonality and proportion of IGHA1 and IGHG1 isotypes (Supplementary Fig. 6a). DEG analysis revealed distinct expression of signature genes for different B cell types (Supplementary Fig. 6b). To trace the development of B cells, we performed pseudotime trajectory analysis using diffusion map, which revealed a developmental trajectory of B cells from naïve B cells toward memory, GC, and plasma B cells, along with the increasing SHM frequencies of IgH, clonal expansion, and isotype switch (Fig. 2a). To investigate GC reaction, we applied principal component analysis (PCA) and circular trajectory

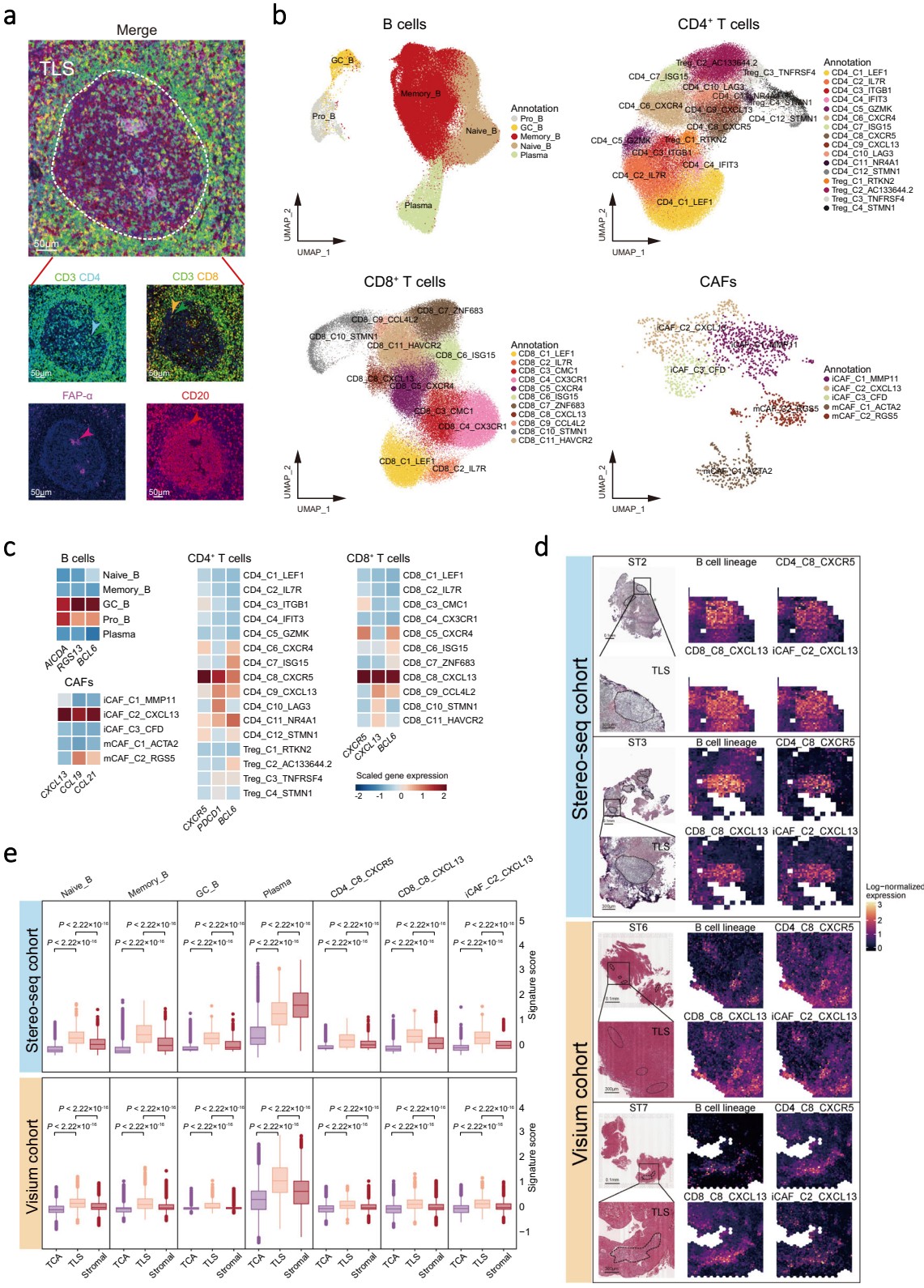

analysis of NPC GC B cells, revealing a characteristic continuum of gene expression states spanning the dark-light zone axis[31]. Furthermore, B cells in dark and light zones exhibited increased expression of *AICDA* and *CD83*, respectively, suggesting a common pattern of GC reaction in TLSs as previously reported[12,13] (Supplementary Fig. 6c-e). These findings together suggest that B cells infiltrating the tumour may undergo productive activation through GC reactions, affinity maturation, and antibody class switching in the NPC TME.

We noted that the proportion of GC B cells was significantly lower in the NPC tumours at the advanced stage (III/IV; median value: 1.0%) than that at the early stage (I/II; median value: 3.3%; Fig. 2b and Supplementary Fig. 6f) in the scRNA-seq samples and other two independent NPC cohorts (Bulk-RNA-seq cohort, n = 147 and Microarray cohort, n = 150; Supplementary Fig. 6g) as reflected by the transcription level of GC B cell signature (see Methods). By contrast, the proportions of plasma cells were similar across NPC samples at different

**Fig. 1 | TLS composition at single-cell and spatial-resolved resolution.**
**a** Representative images of multiplex IHC staining of TLS in NPC tissue biopsy. Cells were coloured according to their staining with CD3 (green), CD4 (cyan), CD8 (orange), CD20 (red), and FAP-α (purple) proteins as indicated on top. Images are representative data from three independent NPC samples. Scale bars is 50 μm.
**b** UMAP plots of cell clusters for B cells, CD4+ T cells, CD8+ T cells, and CAFs. Individual cell types were segregated from the major cell clusters identified in Supplementary Fig 1c and subsequently clustered independently. Each dot represents a single cell, with colour coded according to the cell types or clusters as indicated in the legend to the right (comprising five, 16, 11, and five clusters for 97,360 B cells, 88,009 CD4+ T cells, 95,686 CD8+ T cells, and 1652 CAFs, respectively). **c** Heatmap plots of the normalised mean expression of TLS canonical marker genes (columns) for different cell clusters (rows) in B, CD4+ T, CD8+ T cells, and CAFs. Filled colours from blue to red represent scaled expression levels from

low to high. **d** Representative images of Hematoxylin-Eosin staining (left) for TLSs and spatial feature plots of TLS-associated cell types (right) from the Stereo-seq (top; $n = 3$) and the Visium cohorts (bottom; $n = 12$). Dashed outlines indicate TLS areas on the H&E slides with scale bars of 0.1 mm and 300 μm. In spatial feature plots, the expression levels of the TLS-associated cluster signatures for each cell type (indicated on top) in each spot are filled with colours from black to yellow, representing scaled expression levels from low to high. **e** Box plots showing the signature scores of B lineage cells (Naïve_B, Memory_B, GC_B, and plasma cells), CD4_C8_CXCR5, CD8_C8_CXCL13, and iCAF_C2_CXCL13 in different tumour regions from NPC patients of the Stereo-seq (top; $n = 9563$) and the Visium cohorts (bottom; $n = 21,753$). Centre lines denote median values, and whiskers denote 1.5 × the interquartile range. $P$ values are derived from two-sided student t-tests. Source data are provided as a Source Data file.

stages (Fig. 2b). Given the link between GC B and plasma cells, we explored this seeming paradox's mechanism. First, we observed a dramatic reduction in the transitions of the same BCR clonotypes between GC B and plasma cells in the NPC tumours at the advanced stages compared with that at the early stages (Fig. 2c), suggesting that neonatal plasma cells through GC reaction were decreased in the advanced tumours. Second, pan-immunoglobulin (Ig) transcription levels were significantly lower in the NPC tumours at the advanced stages than at the early stages in all the three cohorts (Fig. 2d and Supplementary Fig. 6h). Third, IHC staining also revealed lower expression of IgA and IgG in NPC tumours at the advanced stage than that at the early stage in an additional sample collection ($n = 27$, Fig. 2e). Fourth, GSEA revealed the downregulation of genes for multiple pathways in plasma cells from tumours at the advanced stages, including BCR signalling, peptide biosynthetic process, and protein transport (Fig. 2f), which suggest decreased antibody production of plasma cells. We also noted that the pathways related to antibody production and immune response were downregulated in plasma cells from TCA compared with that from TLS in the Visium (Fig. 2g) and stereo-seq (Supplementary Fig. 6i) ST cohorts. Lastly, survival analysis revealed that higher proportions of GC B cells and transcriptional levels of pan-Ig were both associated with better prognosis of NPC in two independent cohorts (GC B cells: Bulk-RNA-seq cohort, progression-free survival, PFS: $P = 0.019$, hazard ratio, HR = 0.39; Microarray cohort, PFS: $P = 0.00095$, HR = 0.33, OS: $P = 0.00012$, HR = 0.19; Fig. 2h and Supplementary Fig. 6j). Taken together, these observations strongly suggest that the GC reaction of GC B cells is essential in sustaining antibody production amid reduced differentiation and function of plasma cells along tumour progression. Additionally, transcriptome and deconvolution analyses revealed a higher proportion of GC B cells in EBV+ GaC samples compared with EBV- GaC samples (TCGA and in-house data; Supplementary Fig. 6k), suggesting a link between EBV infection and GC reaction within the tumour.

## Antibodies secreted by plasma cells promote the apoptosis of malignant cells in EBV-related epithelial tumours

Antibodies produced by plasma cells mediate antitumor activity through ADCC or ADCP executed by immune cells such as NK cells or macrophages[9,32], respectively. ST analyses and multiplex IHC staining assays revealed that a substantial number of plasma cells were in close contact with NK cells and macrophages in NPC biopsy sections (Supplementary Fig. 7a, b). Furthermore, we observed a significant positive correlation between the proportions of plasma cells and that of macrophages or NK cells in most NPC ST samples (Stereo-seq cohort: 2/3; Visium cohort: 10/12; Supplementary Fig. 7c), suggesting a widespread colocalization of plasma cells with NK cells and macrophages in NPC tumours. These findings suggest that NK cells and macrophages may play an anti-tumour role in NPC through ADCC and ADCP, respectively.

We noted a lower pan-Ig transcription in tumoral plasma cells for NPC patients with high EBV load than those with low EBV load in either

tumour or peripheral blood sample from the Bulk-RNA-seq NPC cohort (Supplementary Fig. 8a; see Methods). We further evaluated the EBV infection status in the two ST cohorts, by using the top 100 DEGs between EBV-high and -low tumour cells derived from scRNA-seq data[27] as a surrogate signature for the Visium FFPE cohort that does not capture EBV genes and using directly EBV gene expression for the Stereo-seq cohort (see Methods). We identified two types of TCA containing distinct malignant cells with high or low EBV signature (EBV^high- or EBV^low-TCA, respectively), which were further corroborated by in situ hybridisation of EBV-encoded EBERs only in malignant cells (PanCK) in NPC biopsy samples (Supplementary Fig. 8b–e). Among them, EBV^high-TCA was negatively associated with a high proportion of plasma cells in the two ST cohorts (Stereo-seq cohort: 2/3; Visium cohort: 12/12) of NPC (Supplementary Fig. 8f). Additionally, we observed the proportion of plasma cell was correlated with the tumour apoptosis signature at single cell level (Supplementary Fig. 8g). Using ST data, we also observed that the TCA with plasma cells (TCA-wP) had higher apoptosis signatures (defined by the transcriptional level of extrinsic apoptosis-related genes) than those without plasma cells (TCA-woP), with the highest for EBV^high-TCA with plasma cells (EBV^high-TCA-wP; Fig. 3a, b and Supplementary Fig. 9a; see Methods), especially expressing *FAS* and *CASP3* (Supplementary Fig. 9b). Such patterns were also observed in an EBV+ gastric cancer (EBV+ GaC) cohort with ST data (Supplementary Fig. 9c–f) and the Microdissection NPC cohort (see Methods; Supplementary Fig. 9g, h). Multiplex IHC staining assays demonstrated a higher abundance of apoptotic tumour cells (CASP3+PanCK+) surrounded by macrophages (CD68), NK cells (CD56), and plasma cells (IgG) in EBV^high-TCA compared to EBV^low-TCA (Fig. 3c). Furthermore, spatial proximity analysis revealed that the number of apoptotic tumour cells (CASP3+PanCK+) interacting closely with plasma cells (IgG+; within 20 μm in distance) in the EBV^high-TCA was significantly higher than that in the EBV^low-TCA (Fig. 3d, e and Supplementary Fig. 10a, b). Together, these observations suggest a plausible mechanism that plasma cells generate antibodies to induce apoptosis and thus reduce the burden of malignant cells in EBV-related epithelial tumours.

Cell-cell interaction analysis revealed that EBV+ (LMP1) NPC malignant cells interact with plasma cells via CCL2-CCR2 (Supplementary Fig. 10c). Furthermore, transcriptional correlation analysis revealed a positive correlation between the plasma cell signature and the average expression of CCL2-CCR2 ligand/receptor pairs in a bulk RNA-seq cohort (Supplementary Fig. 10d). These data suggest a potential interaction where EBV+ NPC malignant cells may attract plasma cells via the CCL2-CCR2 axis, leading to the formation of a niche enriched with plasma cells in EBV+ TCA.

## Plasma cells contribute to immunotherapy response in NPC

To explore the potential role of antibodies generated by plasma cells in immunotherapy, we compared the above characteristics of antibody and plasma cells between responders and non-responders from the ST

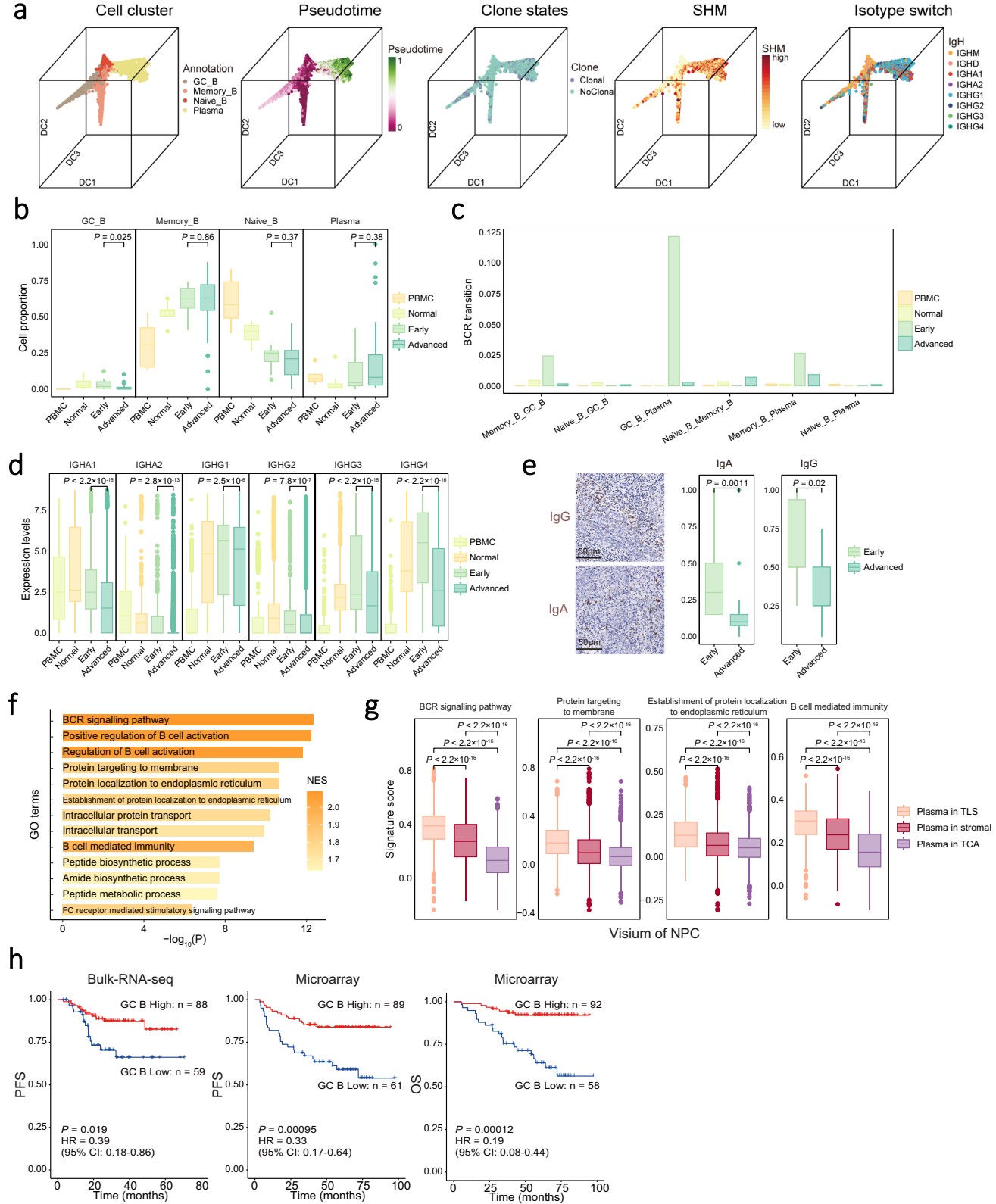

cohorts of 12 NPC patients (Visium cohort) with additional anti-PD1 immune checkpoint blockade (ICB) therapy (toripalimab plus chemotherapy; Supplementary Data 1). We observed that the responders had higher expression levels of tumoral pan-Ig and more spots with tumoral infiltration of plasma cells than the non-responders (Fig. 4a and Supplementary Fig. 10e). Moreover, we also observed that the responders had higher proportions of TCA-wP, especially EBV[high]-TCA-

wP (Fig. 4b) and significantly higher apoptosis signatures for EBV[high]-TCA-wP than the non-responders (Fig. 4c). These observations suggest that plasma cells promoting the apoptosis of EBV[high] malignant cells also contribute to the ICB therapy response in NPC. Of note, we observed higher immunosuppressive signatures for malignant cells in the non-responders than the responders and for EBV[high] malignant cells or TCA than the EBV[low] counterparts in EBV-related epithelial tumours

**Fig. 2 | Developmental trajectory and clinical implication of B cells. a** 3D plots showing the spatial distribution of B cell clusters. Each dot represents one single cell, coloured according to its cluster label, pseudotime score, clone state, SHM, and isotype as indicated. **b** Box plots showing the proportion of B cell clusters in all B cells derived from different sources and stages of NPC ($n = 77$; removing samples without detectable malignant cells in the NPC tissue and samples with fewer than 100 B cells). **c** Bar plot showing BCR transition scores among B cell clusters derived from different sources and stages of NPC. **d** Box plots showing the expression levels of antibodies in plasma cells ($n = 7060$) from different sources and stages of NPC. **e** Representative images of IgA and IgG staining (left) in tumour samples at different stages of NPC. Box plots showing the protein levels of IgA and IgG in tumour samples with different stages of NPC ($n = 27$). Scale bar, 50 µm. **f** Bar plot of GSVA results showing the down-regulated signalling pathways enriched in plasma cells from NPC at advanced than that at early stages. **g** Box plots showing the expression levels of antibody production and immune response related pathways of plasma cells ($n = 7694$) from different regions in the Visium cohort. **h** Kaplan-Meier survival curves of NPC cohorts (Bulk-RNA-seq, $n = 147$; Microarray, $n = 150$) with patients stratified by the proportion of GC B cells in the tumour. Survival duration and probability were indicated at the x- and y-axis, respectively. $P$ value and HR were calculated using a two-sided cox test. PFS, progression-free survival; OS, overall survival. In box plots, centre lines denote median values, and whiskers denote $1.5 \times$ the interquartile range. $P$ values are derived from two-sided student t-tests. Source data are provided as a Source Data file.

(NPC and EBV$^+$ GaC; Fig. 4d), suggesting that the ICB therapy response might be mediated through the ability of ICB to reverse the immunosuppression of EBV$^{high}$ malignant cells.

## CXCL13$^+$ TLS-associated CAFs promote B cells adhesion and antibody production in NPC

Although FDCs (follicular dendritic cells) have been demonstrated as an essential stromal component for TLS formation[24], the expression of their canonical marker (CR2) is scarcely detected in CAFs in NPC samples (Supplementary Fig. 11a). Instead, we identified a TLS-associated CXCL13$^+$ CAFs (iCAF_C2_CXCL13), with specific expression of TLS canonical markers, including *CXCL13, CCL19*, and *CCL21* (Fig. 5a and Supplementary Data 2). DEG analysis revealed that CXCL13$^+$ CAFs expressed cluster markers (*PTGDS, TNFSF13B, CYP1B1*, and *PDPN*), complement component associated genes (*C2* and *C3*), and cell adhesion molecules (*VCAM1* and *ICAM1*; Fig. 5a and Supplementary Data 2), which have been implicated in the activation and adhesion of B lymphocytes[33,34]. GSVA revealed a specific pattern of enriched pathways in CXCL13$^+$ CAFs, including 'B cell chemotaxis', 'Germinal centre formation', and 'Activation of C3 and C5' pathways (Fig. 5b), which have been involved in GC formation and the migration and activation of B cells[35,36]. Correlation analysis revealed that the proportions of CXCL13$^+$ CAFs were significantly correlated with B lineage clusters and the transcription levels of multiple IgH genes in the Bulk-RNA-seq and Microarray NPC cohorts (Supplementary Fig. 11b). We also observed a downtrend in the proportion of CXCL13$^+$ CAFs in NPC samples from early to advanced stages (Fig. 5c), resembling that of GC B cells and antibody level (Fig. 2b–d). Furthermore, multiplex IHC staining assays revealed the presence of CXCL13$^+$ CAFs in B cell aggerates (B cell zone of TLS; Fig. 5d). Spatial analysis also indicated that CXCL13$^+$ CAFs were located within TLS, while CXCL13$^-$ CAFs were primarily situated in the stromal area (Fig. 1e and Supplementary Fig. 11c). Taken together, these observations suggest that CXCL13$^+$ CAFs might be associated with GC formation and antibody production of B cells in TLS of NPC.

Cell-cell interaction analysis revealed that CXCL13$^+$ CAFs interacted with B cells through CXCL13-CXCR5, VCAM1-integrin, ICAM1-integrin, and TNFSF13B-related ligand-receptor pairs (Fig. 5e), which are well-known mediators for B cell chemotaxis, adhesion, and activation[37,38]. Supportively, ST and multiplex IHC analyses revealed that CXCL13$^+$ CAFs were embedded in plasma cell aggregates with strong expression of pan-Ig; and the cell proportion was strongly correlated with that of plasma cells in NPC biopsies (Supplementary Figs. 11d-f). SpaGene analysis revealed high expression of the ligand-receptor pairs in the intra-spots and inter-spots of TLS regions (Fig. 5f and Supplementary Fig. 12a), suggesting the presence of spatial contact between CXCL13$^+$ CAFs and B cells with interacting potential. To validate the function of CXCL13$^+$ CAFs on B cells, we performed in vitro co-culture of CXCL13$^+$ CAFs (VCAM1$^+$ CAFs) from NPC tissues and peripheral B cells from healthy donors (Supplementary Figs. 12b-d). CCK8 assays revealed significantly more B cells binding to VCAM1$^+$ than VCAM1$^-$ CAFs, and this increase in binding ability was diminished using a VCAM1 neutralising antibody (Fig. 6a). These data suggest that

VCAM1$^+$ CAFs have stronger adhesion capability to B cells than VCAM1$^-$ CAFs. Furthermore, co-culture assays also revealed that CXCL13 overexpression significantly enhanced B cell migration, which was impeded by the addition of a CXCL13 neutralising antibody (Fig. 6b), strongly suggest the critical role of CXCL13$^+$ fibroblasts in recruiting B cell aggregates through the CXCL13-CXCR5 axis.

Interestingly, co-culture assays also revealed significantly higher IgG production from B cells co-cultured with CXCL13$^+$ CAFs (VCAM1$^+$) than other (VCAM1$^-$) CAFs (Fig. 6c), suggesting that VCAM1$^+$ CAFs might induce differentiation and maturation of B cells. Notably, cell-cell interaction analysis revealed intensive interactions between CXCL13$^+$ CAFs and plasma cells via high expression of cytokine and cytokine receptors, including TNFSF13B-TNFRSF13B, which is well-known as B-cell maturation factor[39]. Subsequently, to determine whether CXCL13$^+$ TLS-associated CAFs enhance B cell antibody production via TNFSF13B signalling, we carried out similar co-culture experiments with additional TNFSF13B overexpression or blockade in fibroblasts. The co-culture assays revealed significant increase in IgG production of B cells co-cultured with fibroblasts overexpressing TNFSF13B compared to those with control fibroblasts (Fig. 6d). By contrast, the increased IgG production of B cells due to TNFSF13B overexpression was significantly abolished with addition of a TNFSF13B neutralising antibody (Fig. 6d). These findings corroborate that CXCL13$^+$ TLS-associated CAFs promote antibody production through TNFSF13B signalling. Additionally, survival analysis revealed that a higher proportion of CXCL13$^+$ CAFs with a higher expression of *IgG* was associated with a better prognosis of NPC (Supplementary Fig. 12e). These observations strongly suggest that CXCL13$^+$ CAFs interact with B cells and mediate their antibody production, thereby promoting better survival of NPC patients.

## Developmental potential of CXCL13$^+$CD8$^+$ T cells in TLS to exhausted CD8$^+$ T cells in TCA

We identified a TLS-associated CD8$^+$ T cell cluster, CD8_C8_CXCL13, with strong expression of some canonical makers, such as of activated T cell markers (*CD27, CD28, ICOS, TNFRSF4*, and *TNFRSF14*), stem-like markers (*TCF7, BCL6*, and *PLAGL1*), chemokines (*CXCL13, CXCR5, CCR4*, and *CXCR3*), and Toll-like receptors (*TLR4-6, TLR8*, and *TLR9*; Fig. 7a and Supplementary Data 2). We noted that CD8_C8_CXCL13 cells expressed *PDCD1*, but not other exhausted molecules (*HAVCR2, LAYN, CTLA4*, and *ENTPD1*) that were highly expressed in exhausted CD8$^+$ T cells (CD8_C11_HAVCR2; Fig. 7a and Supplementary Data 2). Multiplex IHC staining assays also revealed the presence of CXCL13$^+$CD8$^+$PD1$^+$HAVCR2$^-$ T cells (CD8_C8_CXCL13) surrounding B cells in NPC (Fig. 7b). Furthermore, we observed strong correlations of *PDCD1* expression with activation genes in CD8_C8_CXCL13 cells but with exhaustion genes in CD8_C11_HAVCR2 cells (Supplementary Fig. 13a). These observations suggest that *PDCD1* might prevent excessive activation of CD8_C8_CXCL13 cells rather than exhaustion, consistent with previous findings in mouse models[40,41]. Besides, GSVA revealed that signalling pathways related to epigenetic modification and metabolism were oppositely enriched in CD8_C8_CXCL13 cells

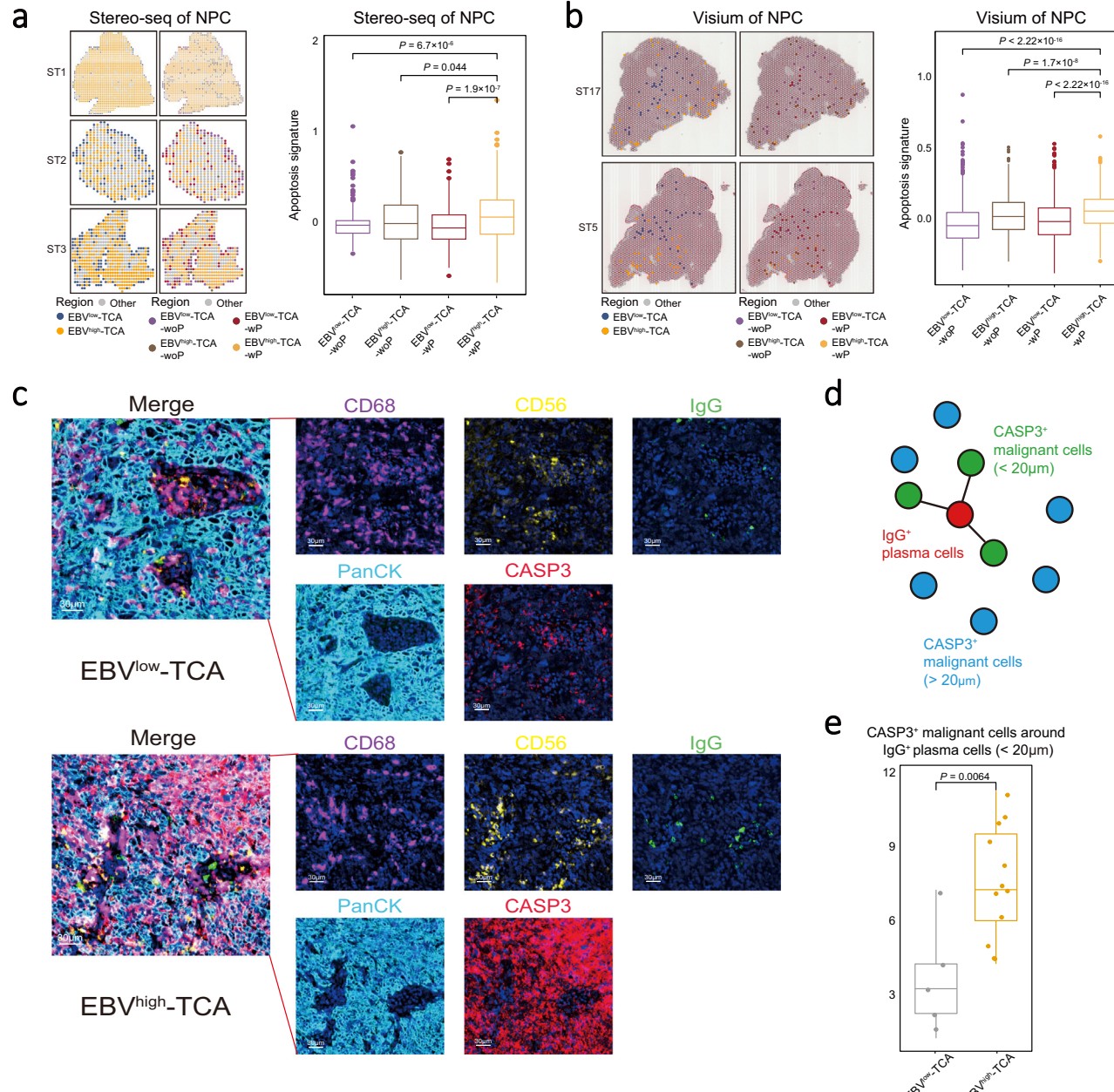

**Fig. 3 | Different distribution patterns of TCA in EBV-related epithelial tumours. a, b** Spatial distribution (left) and apoptosis signature level (right) of different TCA in representative NPC sections from the Stereo-seq (**a**; $n$ = 3491) or the Visium (**b**; $n$ = 3595) cohort. Each type of TCA is coloured as indicated. Box plots showing the apoptosis signatures in different TCA of NPC (right). **c** Representative images of multiplex IHC staining of molecules related to ADCC and ADCP at different regions of NPC tissue biopsy. Cells are coloured according to their staining with IgG (green), PanCK (cyan), CD56 (yellow), CD68 (purple), and CASP3 (red) proteins as indicated on top. Images are representative data from three independent NPC samples. Scale bars are 30 μm. **d** Schematic diagram showing the spatial distribution of CASP3+ malignant and IgG+ plasma cells, coloured by cell clusters and spatial distance. **e** Box plot showing the distribution of CASP3+ malignant cells in relation to their distance from IgG+ plasma cells ($n$ = 18). In box plots, centre lines denote median values, and whiskers denote 1.5 × the interquartile range. $P$ values are derived from two-sided student t-tests. Source data are provided as a Source Data file.

('DNA methylation', 'Glycogen degradation', and 'DNA synthesis') and the exhausted CD8_C11_HAVCR2 cells ('DNA demethylation' and 'O-Glycan biosynthesis'; Supplementary Fig. 13b). Since DNA demethylation and glycans biosynthesis induce T cell exhaustion[42–44], the data suggest the exhausted feature of CD8_C11_HAVCR2 cells but not CD8_C8_CXCL13 cells.

To further explore the development of CD8+ T cells in NPC, we first performed pseudotime trajectory analysis and revealed four developmental paths from CD8_C1_LEF1, CD8_C7_ZNF683,

CD8_C8_CXCL13, and CD8_C10_STMN1 cells at initial or mediate states to CD8_C11_HAVCR2 at terminal state (Fig. 7c and Supplementary Fig. 13c). Along with the developmental path from CD8_C8_CXCL13 to CD8_C11_HAVCR2 cells, we observed an increasing trend for exhaustion scores but a decreasing trend for stem-like scores (Fig. 7d, Supplementary Fig. 13c, and Supplementary Data 4). TCR sharing analysis revealed that CD8_C11_HAVCR2 cells had the most top 10 identical TCR clonotypes with CD8_C8_CXCL13 cells than other CD8+ T cells (Fig. 7c and e, and Supplementary Data 5). Furthermore, TCR diversity was

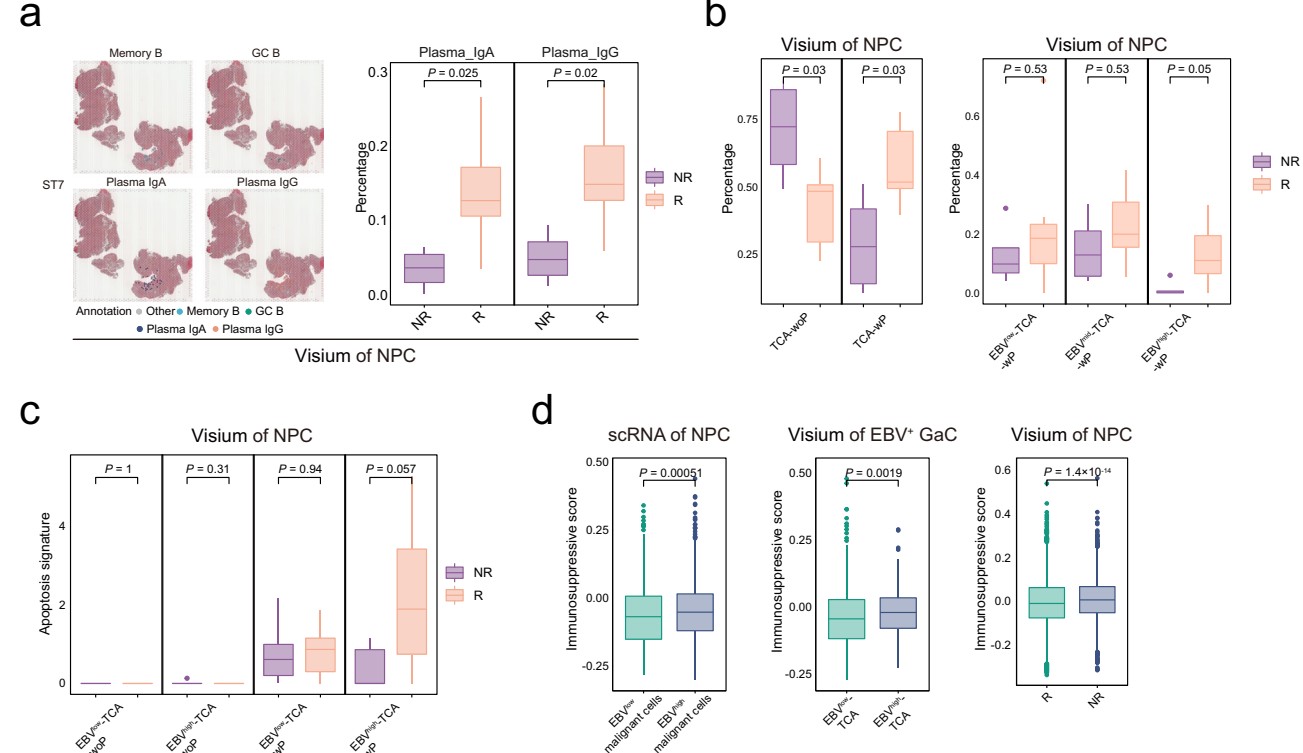

**Fig. 4 | Plasma cells associated with immunotherapy response in NPC. a** Spatial distribution of Memory_B, GC_B, Plasma_IgA, and Plasma_IgG in representative NPC sections (left panel) from the Visium cohort. Each B cell region is coloured as indicated. Box plots showing the proportions of Plasma_IgA and Plasma_IgG among B cells in the ICB responders and non-responders for NPC (right panel; n = 12). **b** Box plots showing the proportions of TCA-wP or TCA-woP (left panel) and EBV$^{high/mid/low}$-TCA-wP or EBV$^{high/mid/low}$-TCA-woP in the ICB responders and non-responders for NPC (right panel) from the Visium cohort (n = 12). **c** Box plots showing the apoptosis signature levels of EBV$^{high/low}$-TCA-wP or EBV$^{high/low}$-TCA-woP in the ICB responders and non-responders for NPC from the Visium cohort (n = 12). **d** Box plots showing the immunosuppressive scores of EBV$^{high/low}$ malignant cells (n = 2787) in NPC (left) or EBV$^+$ GaC (middle; n = 1152) and TCA in the ICB responders and non-responders for NPC (right; n = 7228). P values are derived from two-sided student t-tests. In box plots, centre lines denote median values, and whiskers denote 1.5 × the interquartile range. TCA, tumour cell aggregates; EBV$^{high}$-TCA-wP, EBV$^{high}$-TCA with plasma cells; EBV$^{low}$-TCA-wP, EBV$^{low}$-TCA with plasma cells; EBV$^{high}$-TCA-woP, EBV$^{high}$-TCA without plasma cells; EBV$^{low}$-TCA-woP, EBV$^{low}$-TCA without plasma cells. Source data are provided as a Source Data file.

decreased in CD8_C11_HAVCR2 compared with CD8_C8_CXCL13 cells (Supplementary Fig. 13d). ST analysis revealed that CD8_C8_CXCL13 T cells were aggregated and significantly enriched in TLS compared to CD8_C11_HAVCR2 T cells that were interspersed and enriched in TCA (Fig. 7f). Together, these observations suggest a developmental trajectory from CD8_C8_CXCL13 cells in TLS to exhausted CD8_C11_HAVCR2 cells widespread into TCA along with decreased stemness and TCR diversity in NPC.

To explore the anti-tumour mechanism of CD8$^+$ T cells, we reconstructed the potential connections between CD8$^+$ T and malignant NPC cells using CSOmap software (Supplementary Fig. 13e). We observed that CD8_C8_CXCL13 cells had the highest connections with malignant NPC cells among CD8$^+$ T cells (Supplementary Fig. 13f), suggesting that CD8_C8_CXCL13 cells had the most potential engagement in recognising and attacking malignant NPC cells. Furthermore, we noted that CD8_C8_CXCL13 cells had more chemokines (CXCR3-CCL20, CXCR3-CXCL10, and CXCR3-CCL19) and co-stimulatory (CD27-CD70) connections with malignant NPC cells than CD8_C11_HAVCR2 cells (Fig. 7g). Additionally, we performed multiplex IHC staining on NPC biopsies and confirmed the physical juxtapositions of CD70-expressing malignant cells (CD70$^+$PanCK$^+$) and CD27-expressing CXCL13$^+$CD8$^+$ T cells (CD8$^+$CXCL13$^+$CD27$^+$; Supplementary Fig. 13g). Although the prevalence of CD8_C8_CXCL13 cells within NPC tumours is relatively low (scRNA-seq cohort: 0-9.89%; Multiplex IHC cohort: 0.5-11%; Supplementary Fig. 13h), their proportion was notably higher in early stage NPC tumours compared to those at advanced stages

(Supplementary Fig. 13i). These observations suggest that CD8_C8_CXCL13 cells might be more actively recruited to TLS and have more substantial anti-tumour potential than the exhausted CD8$^+$ T cells in TCA of NPC, especially at the early stage.

### TLS-associated cell clusters fostering an immune-activated niche are associated with prognosis in NPC

Cell-cell interaction analysis revealed interactions between B cells and CD8_C8_CXCL13 T cells through MHC-CD8, ICOSLG-ICOS, CD86-CD28, and IL7-IL7R ligand-receptor pairs, whereby B cells act as antigen presenting cells and activate CD8$^+$ T cells in tumours[45,46] (Supplementary Fig. 14a, b). The analysis also revealed intensive interactions between B cells and CD4_C8_CXCR5 (Tfh) through CXCR5-CXCL13, CD40-CD40LG, and TNFRSF14-TNFSF14 ligand-receptor pairs (Supplementary Fig. 14c, d), which are known essential mediators for the activation and maturation of B cells[37,47]. This is consistent with a previous finding that Tfh cells play a critical role in mediating the selection and differentiation of B cells within germinal centres[48]. Given that B lineage cells, CD4_C8_CXCR5 and CD8_C8_CXCL13 T cells, and CXCL13$^+$ CAFs were the essential components of TLSs in NPC, we also observed strong pairwise correlations of cell proportions among these clusters in the three NPC cohorts (Supplementary Fig. 14e). Taken together, these data revealed an intercellular network among the TLS-associated cell clusters through intensive ligand-receptor pairs related to immune activations, strongly suggesting that the cross-talks might foster an immune-activated niche for the TME of NPC.

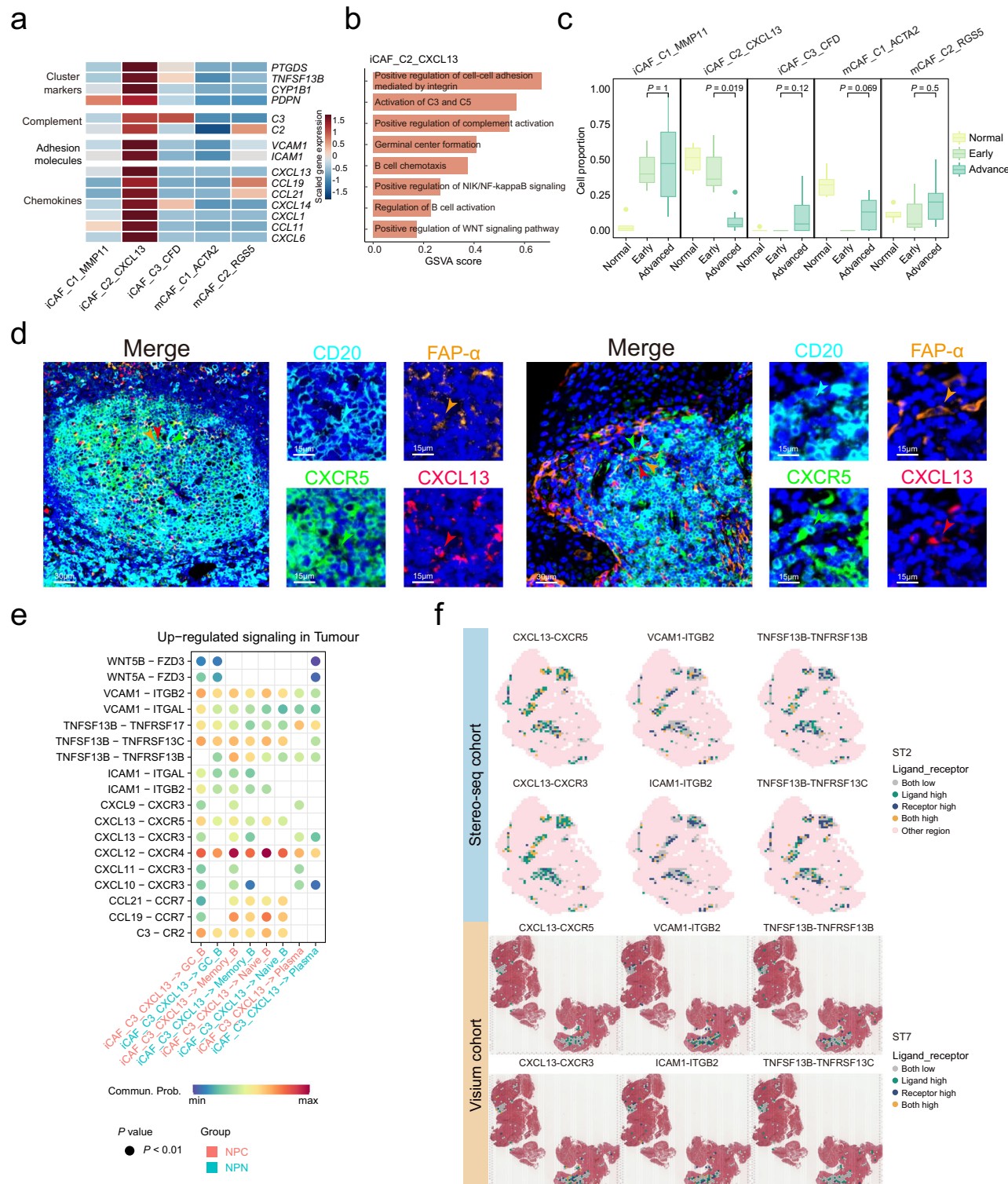

Based on the transcriptional signatures of TLS-associated cells derived from scRNA-seq data, we estimated the proportions of various TLS-associated cell types in the Microdissection, Bulk-RNA-seq, and Microarray NPC cohorts (Supplementary Fig. 15a; see Methods). These samples are comparable, as there are no differences in their clinical characteristics. (Supplementary Fig. 15b). With these, survival analysis revealed strong associations between the prognosis of NPC patients and the proportions of seven TLS-associated cell types, including B lineage cells (Naïve B, Memory B, GC B, and plasma cells), CD4_C8_CXCR5 and CD8_C8_CXCL13 T cells,

and CXCL13⁺ CAFs, with higher cell proportions for better survival (Fig. 8a). Moreover, we identified two groups of NPC patients with high and low TLS cell signature (TLS-CS) in three cohorts based on the transcriptional levels of signature genes of the TLS-associated cell clusters (Fig. 8b). Survival analysis revealed that NPC patients with higher TLS-CS had significantly a better prognosis than those with lower TLS-CS (Bulk-RNA-seq cohort, PFS: $P = 0.0018$, HR = 0.29; Microarray cohort, PFS: $P = 0.00033$, HR = 0.32, OS: $P = 0.00002$, HR = 0.19; Fig. 8c). We additionally conducted multivariate Cox regression analysis, revealing that the TLS signature independently

**Fig. 5 | Expression signature and functional characterisation of iCAF_C2_CXCL13 CAFs. a** Heatmap showing the normalised mean expression of cluster markers, complement, adhesion molecules, and chemokines (rows) for each CAF cluster (columns). Filled colours from blue to red represent scaled expression levels from low to high. **b** Bar plot of GSVA results showing the selected up-regulated signalling pathways enriched in iCAF_C2_CXCL13 CAFs. **c** Box plots showing the proportions of fibroblast clusters among all fibroblasts in NPC samples at different stages ($n = 16$; removing samples without detectable malignant cells in the NPC tissue and samples with fewer than 100 CAFs). Centre lines denote median values, and whiskers denote $1.5 \times$ the interquartile range. *P* values are derived from two-sided student t-tests. **d** Representative images for multiplex IHC staining of CXCL13[+] CAFs and CXCR5[+] B cells in NPC tissue biopsy. Cells were coloured according to their staining with CXCR5 (green), CD20 (cyan), FAP-α (orange), and CXCL13 (red) proteins as indicated on top. The green, cyan, orange, and red arrows

indicated positive cells with the expression of CXCR5, CD20, FAP-α, and CXCL13 proteins in NPC tissue, respectively. Images are representative data from three independent NPC samples. Scale bars are 30 μm or 15 μm, respectively. **e** Dot plot showing selected ligand-receptor interactions between iCAF_C2_CXCL13 CAFs and B cells (GC_B, Memory_B, Naïve_B, and plasma cells) in NPC (red) or NPN (blue) samples. Ligand-receptor interactions are indicated in rows. The means of the average expression levels of two interacting molecules are indicated by the colour heatmap (bottom panel), with blue to red representing low to high expression. *P* values are obtained by one-sided permutation tests and indicated by circle size. **f** Spatial distribution of ligand-receptor interactions in representative NPC sections from the Stereo-seq (top panel) and the Visium cohorts (bottom panel). Each ligand and its corresponding receptor are indicated on top, with different colours for low and high intensities, as indicated at the right panel. Source data are provided as a Source Data file.

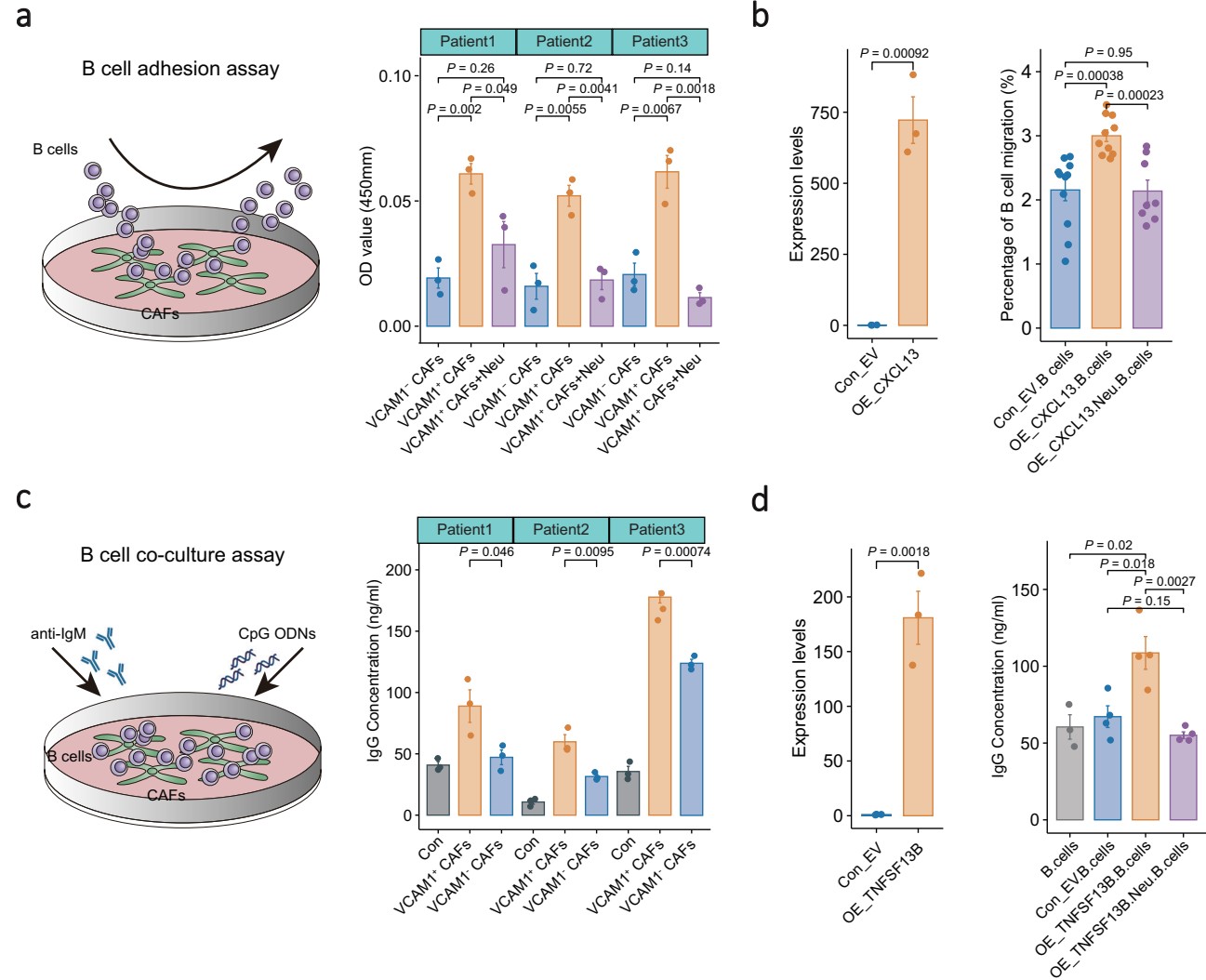

**Fig. 6 | Co-culture assays of CAFs and B cells. a** Schematic diagram showing the B cell adhesion assay (left panel). Bar plots showing the capability of B cell adhesion to CAFs (right panel; $n = 3$). B cells were cocultured with either VCAM1[−] CAFs, VCAM1[+] CAFs, or VCAM1[+] CAFs in combination with a VCAM1 neutralising antibody. The relative number (OD value) of B cells bound to CAFs were quantified using CCK8 assays. **b** Bar plot showing the relative expression level of CXCL13 in fibroblasts (left panel; $n = 3$). Fibroblasts were transduced with an empty vector or CXCL13 plasmid. Bar plot illustrates the migration capabilities of B cells under various co-culture conditions with fibroblasts (right panel; $n = 11$). The B cells were co-cultured with fibroblasts overexpressing either an empty vector (Con_EV), CXCL13 alone, or CXCL13 in combination with a CXCL13-specific neutralising antibody. **c** Schematic diagram showing the B cell co-culture assay (left panel). Bar

plots showing IgG production of B cells without (grey) or with coculture of VCAM1[+] (orange) or VCAM1[−] (blue) CAFs (right panel; $n = 3$). The IgG levels in the cocultured supernatant were determined using ELISA. **d** Bar plot showing the relative expression level of TNFSF13B in fibroblasts (left panel; $n = 3$). Fibroblasts were transduced with an empty vector or TNFSF13B plasmid. Bar plot showing the IgG production of B cells co-cultured with fibroblasts overexpressing either an empty vector (Con_EV), TNFSF13B alone, or TNFSF13B in combination with a TNFSF13B-specific neutralising antibody (right panel; $n = 4$). IgG level in supernatant was determined using ELISA. *P* values are derived from two-sided student t-tests. The data is presented as mean ± SD. OE overexpression, Neu neutralising antibody, Con control, EV empty vector. Source data are provided as a Source Data file.

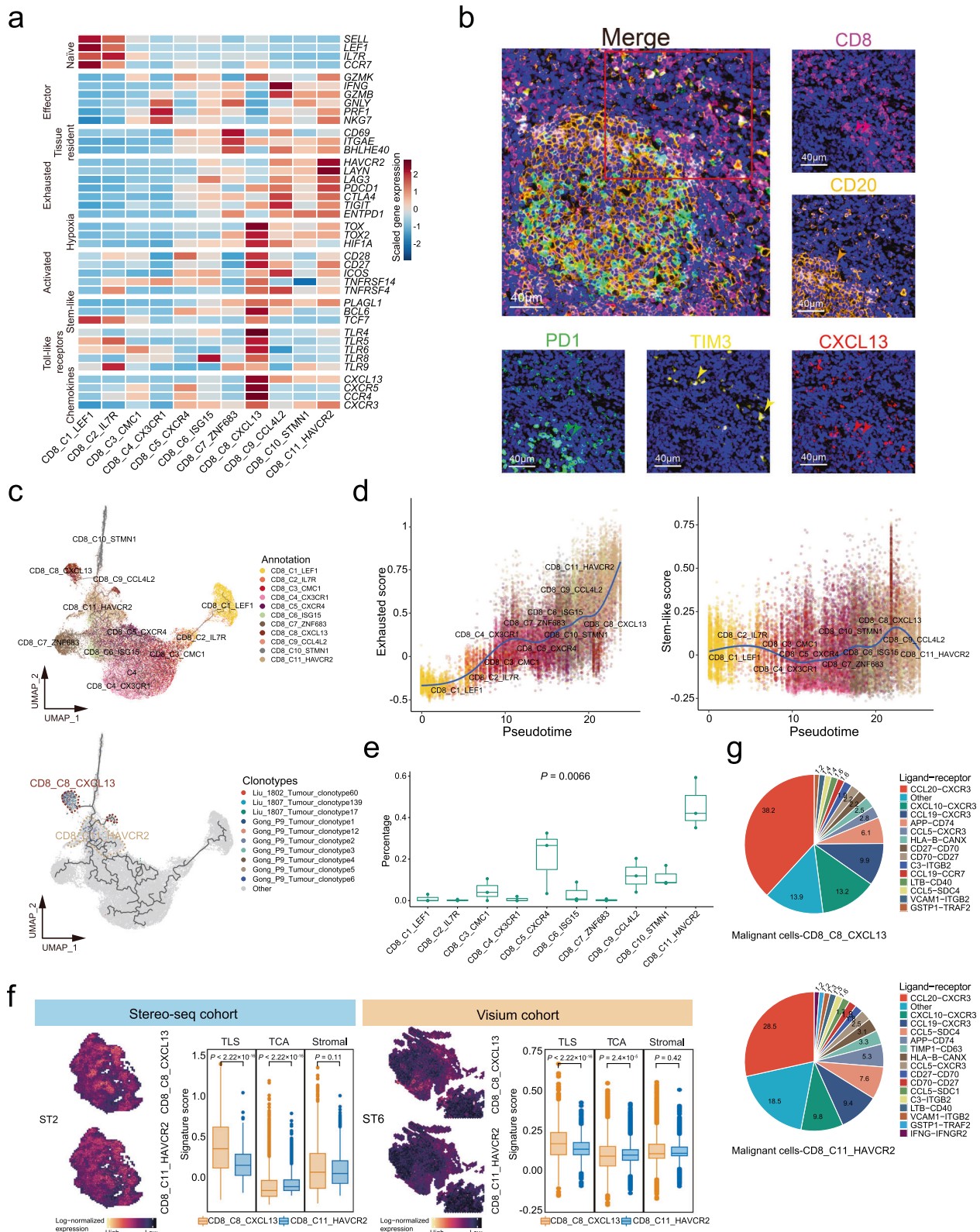

serves as a prognostic indicator for NPC survival (Supplementary Data 6). No significant difference in the clinical characteristics was detected between the high and low TLS-CS groups (Supplementary Fig. 15c−e). The findings were further corroborated by an independent cohort employing multiplex IHC staining for TLS, showing a better prognosis of patients with TLS compared to those lacking TLS (Supplementary Fig. 16a). This multiplex IHC staining also revealed

more immune cell infiltration in NPC patients with TLS compared to those without TLS (Supplementary Fig. 16b, c). Survival analysis demonstrated better prognosis for NPC patients with high immune cell infiltration (Supplementary Fig. 16d). Additionally, we observed significantly higher levels of the seven TLS-associated cell clusters in the ICB responders than in the non-responders (Fig. 8d). Together, these observations suggest that the seven essential cell clusters

**Fig. 7 | Transcriptional and developmental profiles of CD8_C8_CXCL13 cells.**
**a** Heatmap showing normalised mean expression of markers (naïve, effector, tissue resident, exhausted, hypoxia, activation, stem-like, toll-like receptor families, and chemokines) for each CD8+ T cell cluster. Colours from blue to red represent low to high scaled expression levels. **b** Representative images for multiplex IHC staining of CXCL13+CD8+ T cells in NPC tissue biopsy. Cells coloured by PD1 (green), TIM3 (yellow), CD20 (orange), CD8 (purple), and CXCL13 (red). Arrows indicate cells expressing PD1 (green), TIM3 (yellow), CD20 (orange), CD8 (purple), and CXCL13 (red) in NPC tissue. Representative images from three independent NPC samples. Scale bars = 40 μm. **c** Pseudotime trajectories of CD8 + T cells (CD8_C1-C11; $n = 41,736$) with TCR information. Each dot represents a single cell, coloured by cluster label (top) or top 10 TCR clonotypes of CD8_C8_CXCL13 cells (bottom). **d** Scatter plots showing exhausted (left) and stem-like (right) scores of CD8+ T cells along pseudotime scores (x-axis) among clusters. Each dot represents a single cell, coloured by cluster label as in panel (**c**). **e** Box plots showing the percentage of CD8+ T cells sharing TCR clonotypes with top 10 TCR clonotypes of CD8_C8_CXCL13 cells across all CD8+ T cells. Coloured dots represent patients ($n = 3$). Comparison made using a two-sided Kruskal-Wallis test across all clusters. **f** Representative spatial feature plots showing expression levels of CD8_C8_CXCL13 and CD8_C11_HAVCR2 signatures in each data spot from Stereo-seq (left; $n = 9563$) and Visium (right; $n = 21,753$) cohorts. Colours from black to yellow represent low to high scaled expression levels. Box plots show signature scores of CD8_C8_CXCL13 and CD8_C11_HAVCR2 in TLS, TCA, and stromal regions from Stereo-seq (left) and Visium (right) cohorts. Comparisons were made using a two-sided student t-test. **g** Pie charts showing contributions of ligand-receptor pairs among interactions between malignant and CD8_C8_CXCL13 cells (top) or CD8_C11_HAVCR2 cells (bottom) in NPC. Ligand-receptor pairs indicated on right panel with different colours. Percentages are shown in pies. In box plots, endpoints depict minimum and maximum values; centre lines denote median values; whiskers denote 1.5 × the interquartile range. Source data are provided as a Source Data file.

forming TLSs contribute to the prognosis and immunotherapy response of NPC patients.

## Discussion

Here we delineate the spatial locations of TLS, TCA, and stromal regions, as well as the cellular interactions among these tumour compartments through integrative analyses of multi-omics data from multiple independent sample cohorts with the largest sample size up to date (Supplementary Fig. 17a, b). Noteworthily, we identify unique cell populations, including CXCL13+ CAFs and CXCL13+CD8+ T cells, among other immune cells commonly reported in multiple cancers, suggesting the feasibility of using NPC as a model to study the heterogenous TLS in tumours. Furthermore, our study sheds light on the role of GC reaction in the developmental trajectories of antibody-secreting plasma cells from TLS to TCA. Moreover, our study reveals four types of TCA with juxtaposition of plasma cells generating antibodies responsible for the apoptosis of EBVhigh malignant cells. Our study also reveals that CXCL13+ CAFs promote B cell adhesion and antibody production, while B cells may activate CXCL13+CD8+ T cells in TLS.

GC as a mark of TLS maturation is essential for the humoral immune response to restrict tumour growth and metastasis[10,49]. In our study, we observed a 70% decrease in the proportion of GC B cells from early to late-stage NPC. Despite their generally low abundance across various cancer types[10,49], GC cells exhibit a robust proliferative capacity, capable generating a substantial number of memory B cells and plasma cells[50]. These cells play a pivotal role in modulating the tumour microenvironment through the production of antibodies, cytokines, and other factors[51]. However, their relationship is still unclear. Our study corroborated the presence of GC in NPC using multiple ST analyses, in addition to scRNA-seq and multiplex IHC staining reported previously[52]. Noteworthily, our study unveils a developmental trajectory of B cells from naïve B to antibody-secreting plasma cells via GC reaction. Given high clonality and extensive IgH with isotype switch and SHM, the plasma cells are more likely generated within tumours and experienced tumour antigen recognition[53]. Moreover, we observed that plasma cells in TLS had a more robust capacity for antibody production than that in TCA, and antibody levels were higher at early stages than at advanced stages. These observations at both spatial and developmental dimensions suggest an inhibitory effect of malignant cells on the antibody production of plasma cells, which is consistent with the increased immune suppression of malignant cells along with the increased spatial proximity to immune cells[54] and the tumour progression[55,56]. Together with the associations of GC B cells and antibodies with better survival of NPC, these data suggest that GC reaction is a dynamic and essential process giving rise to neonatal plasma cells and antibodies in TLS for effective immune responses in various tumour compartments, leading to favourable cancer survival. In addition, the link between EBV and GC B cells observed in GaC may imply that the virus triggers GC reaction and TLS formation in EBV-related cancers.

Although previous studies have demonstrated that antibodies secreted by plasma cells recognise tumour-associated antigens to promote apoptosis of malignant cells through ADCC or ADCP[9,11], a deeper understanding is needed regarding the distribution of antibodies and the antigen specificity of antibodies within tumours. Our study reveals a strong correlation of antibody level with NK cells and macrophages in NPC, consistent with observations in other cancers[9,32], suggesting a typical role of tumoral antibodies in mediating ADCC and ADCP. Strikingly, by combining scRNA-seq and ST analyses, we noted two distinct distribution patterns of plasma cells (with, wP; or without, woP) engaging either EBVhigh or EBVlow malignant cells, resulting in four types of TCA with diverse transcriptional characteristics and spatial heterogeneity in NPC. EBV infection is strongly linked with the development of EBV-related malignancies and EBV has evolved various strategies to evade both the innate and adaptive immune responses[27,57,58]. Antibodies against EBV molecules have been demonstrated in NPC patients as well as individuals at high-risk of NPC[59]. Given that the EBVhigh-TCA-wP had the highest apoptosis feature among all TCAs, the plasma cells might secrete antibodies recognizing specific antigens derived from EBV or tumour upon EBV infection and thus promote the apoptosis of EBV-infected malignant cells through ADCC or ADCP, which is consistent with the presence of HPV-specific plasma cells in HPV+ HNSCC[11]. Furthermore, we observed a higher apoptosis signature in EBVhigh-TCAwoP than in EBVlow-TCAwp, which might be explained by EBV infection to substantially promote NPC malignant cells apoptosis as reported previously[27]. Additionally, we found that EBVhigh-TCA-wP had a higher percentage and more robust apoptosis signature in the ICB responders than the non-responders, suggesting that the spatial proximity between plasma and EBVhigh malignant cells and antigen specificity are critical for the effectiveness of immunotherapy in NPC patients. Considering EBVhigh malignant cells with higher immunosuppressive capabilities and being enriched in the ICB non-responders, it is plausible that plasma cells secreting antibodies could reverse the immunosuppressive microenvironment formed by EBVhigh malignant cells, thereby leading to a response to immunotherapy for NPC. Thus, we suspect that a combination of EBV-specific antibodies and ICB might be a potentially effective therapeutic strategy for EBV-related tumours.

Our findings unveil a TLS-associated CAF cluster, CXCL13+ CAFs, colocalized with B cell aggregates in the TLS of NPC, sharing common features with immunofibroblasts that arise at the earliest stage during TLS neogenesis and support TLS formation in mouse[60,61]. These cells exhibit high expression of multiple chemokines (CXCL13, CCL19, and CCL21) and adhesion molecules (VCAM1 and ICAM1). Given that CXCL13, CCL19, and CCL21 are specifically involved in the recruitment and positioning of CXCR5+ B cells, CCR7+ T cells, and CCR7+ DC within

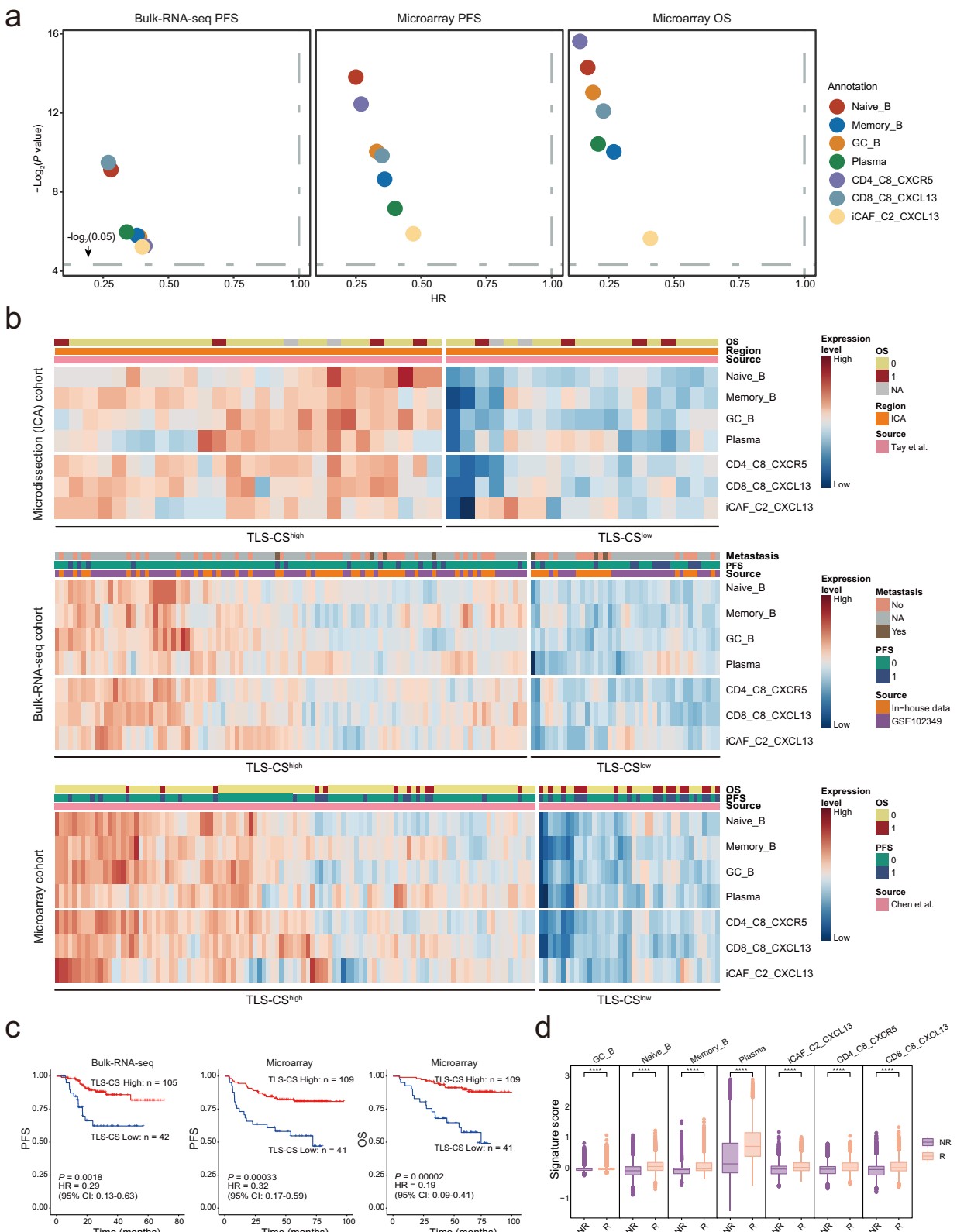

lymphoid follicles[18–20], CXCL13[+] CAFs might initiate TLS formation through recruitment of immune cells. Supportively, co-culture assays showed that CXCL13[+] CAFs could promote B cell adhesion and recruitment in NPC. Furthermore, co-culture assays also showed that CXCL13[+] CAFs promote IgG production of tumoral B cells via TNFSF13 B signalling, which has master regulatory function on B cell differentiation and antibody production[22,60]. Therefore, it is plausible that

better survival for NPC patients with high infiltration of CXCL13[+] CAFs and high levels of antibodies could be explained by the role of CXCL13[+] CAFs in aggregating B cells for TLS formation and antibody production to destroy malignant cells through ADCC and ADCP[62]. With the notion, induction and activation of CXCL13[+] CAFs with growth factors and activated cytokines might be a potential therapeutic strategy for cancer[61].

**Fig. 8 | Prognostic values of TLS components and signatures in NPC. a** Survival analysis of the signature scores for B lineage cells (Naïve_B, Memory_B, GC_B, and plasma cells), CD4_C8_CXCR5, CD8_C8_CXCL13, and iCAF_C2_CXCL13 in two NPC cohorts (Bulk-RNA-seq, *n* = 147; and Microarray, *n* = 150). *P* value (y-axis) and HR (x-axis) were calculated using two-sided cox tests and plotted in different colours for specific cell types annotated at the right. PFS, progression-free survival; OS, overall survival. **b** Heatmap showing the normalised signature scores for B lineage cells (Naïve_B, Memory_B, GC_B, and plasma cells), CD4_C8_CXCR5, CD8_C8_CXCL13, and iCAF_C2_CXCL13 (rows) in each sample (columns) from three NPC cohorts as indicated (source). Filled colours from blue to red represent scaled expression levels from low to high. Coloured bars on top of the heatmaps indicate metastasis, OS (0: dead; 1: alive) or PFS (0 or 1 represent with or without progression), and

sample source. Samples were grouped using two modules with low or high TLS cell signature (TLS-CS). **c** Kaplan-Meier survival curves of the two NPC cohorts (Bulk-RNA-seq, *n* = 147; Microarray, *n* = 150) with patients stratified by two modules of low or high TLS-CS. Survival duration and probability are indicated at the x- and y-axis, respectively. *P* value and HR were calculated using a two-sided cox test. **d** Box plots showing the signature scores for B lineage cells (Naïve_B, Memory_B, GC_B, and plasma cells), CD4_C8_CXCR5, CD8_C8_CXCL13, and iCAF_C2_CXCL13 in the responders (R) and non-responders (NR) to ICB (toripalimab plus chemotherapy) treatment from the Visium cohort of NPC (*n* = 12). Centre lines denote median values, and whiskers denote 1.5 × the interquartile range. *P* values are derived from two-sided student t-tests. **** $P < 2.22×10^{-16}$. Source data are provided as a Source Data file.

TLSs are critical TME facilities to provide effector memory T cells that recognize tumour antigens[33]. We identified the TLS-associated CD8_C8_CXCL13 cell cluster as stem-like CD8+ T cells with high expression of stem-like genes, including *TCF7* in NPC, which is a marker of stem-like CD8+ T cells reported in other tumours[63]. This stem-like CD8+ T cells are similar with CXCL13+TOX+CD8+ T cells reported recently in NPC[64]. Stem-like CD8+ T cells are present in tumours and are essential to promote tumour control in response to ICB[63,65]. Noteworthily, we revealed that CD8_C8_CXCL13 cells in TLS differentiate into CD8_C11_HAVCR2 cells in TCA. Both cell clusters share a high proportion of TCR clonotypes, suggesting that CD8_C8_CXCL13 T cells with higher TCR diversity in TLS are likely early precursor-exhausted T cells with the self-renewing capacity to replenish the effector T cell repertoire continuously. Cell-cell interaction analysis showed that B cells might interact with CD8_C8_CXCL13 cells and regulate their activation and cytotoxicity in TLS of NPC, consistent with findings in other cancers[45,46], suggesting that CD8_C8_CXCL13 T cells might be tumour reactive T cells in response to tumour antigens presented by B cells in TLS. Supportively, our study reveals that a higher proportion of CD8_C8_CXCL13 cells is a favourable indicator for better survival and responsiveness to ICB in NPC, as reported in other cancers[66-68]. These findings suggest CD8_C8_CXCL13 T cells in TLS to be an essential source of effector T cells for anti-tumour response in multiple cancers. The relatively low proportion of CD8_C8_CXCL13 T cells within NPC tumours (ranging from 0% to 11%) may potentially limit their anti-tumour activity. Therefore, we hypothesise that strategies to enrich CD8_C8_CXCL13 T cells and engineer TCRs from both CD8_C8_CXCL13 and CD8_C11_HAVCR2 T cells could enhance the efficacy of adoptive T cell therapy in cancer treatment.

In summary, our study uncovers the essential cellular composition and dynamic interaction of TLS and TCA along with tumour progression and immunotherapy response in NPC at single-cell and spatial-resolved resolutions, providing insights into understanding the crosstalk and immune mechanisms of TLS and TCA and developing therapeutic strategies for cancers. Furthermore, our study identifies TLS-associated cell types (B lineage cells, CD4_C8_CXCR5 and CD8_C8_CXCL13 T cells, and CXCL13+ CAFs) and TCA signatures (EBV^high-TCA-wP) as biomarkers for the prognosis and immunotherapy response of NPC, corroborating TLS's contribution to tumour development. Given that EBV is a type of DNA virus, our findings also provides insights into understanding TLS formation and functions orchestrated by B cells and pathogen infection in other DNA-virus-associated cancers, such as HPV+ head and neck cancers[8], HPV+ cervical squamous carcinoma[69], HBV+ liver cancer[70], etc., where TLS has been previously demonstrated. We acknowledge several limitations in our study. First, the molecular mechanisms behind the involvement of specific cells in TLS formation need further characterisations, and the tumour stratification modules need additional validation in larger, independent cohorts before clinical application. Second, our ST cohort for evaluating immunotherapy response in NPC is small and future research with larger cohorts is essential to confirm their clinical

significance. At technical aspects, the 10x Visium platform for FFPE sample relies on probe capture of RNA molecules and does not include probes for EBV genes. Although we employed cross-validation approaches using the stereo-seq platform and EBER staining to verify EBV infection status, there are inherent limitations in accurately defining the EBV infection status solely with the top 100 DEGs signature. For microfluidic-based single RNA sequencing technology, there is a potential capture bias in processing multiple cell types with variabilities, such as size, tractability, and fragility, resulting fewer tumour cells compared to immune and stromal cells.

## Methods

### Ethics statement
This study complies with all relevant ethical regulations. This study was approved by the Institutional Review Boards at the SYSUCC. Written informed consent was obtained from all participants.

### Study participants
For single-cell RNA sequencing, 56 NPC patients and 11 non-malignant individuals in this study were enrolled from five centres/laboratories previously[27,64,71-73], from whom 77 biopsy and peripheral blood samples were collected (Supplementary Data 1). For spatial transcriptomics (ST) analysis, we collected frozen specimens of three NPC and two EBV+ GaC patients and formalin-fixed paraffin-embedded (FFPE) specimens of 12 advanced NPC patients from Sun Yat-sen University Cancer Center (SYSUCC), Guangzhou, China (Supplementary Data 1). All NPC tumours were EBV positive, as confirmed using in situ hybridisation of EBV encoded small RNAs (EBERs) in tumour biopsy. We also included three independent NPC sample cohorts with bulk transcriptome data (Microdissection cohort[28], *n* = 189; Bulk-RNA-seq cohort[74,75], *n* = 147; and Microarray cohort[76], *n* = 150). For co-culture and multiplex IHC or immunofluorescent (IF) assays, fresh samples were additionally obtained from patients with NPC at the SYSUCC. All patients were histopathologically diagnosed with primary NPC by pathologists according to the World Health Organisation (WHO) classification. No history of cancer and any anti-tumour therapy prior to the primary diagnosis was self-reported before the first biopsy. The clinical staging of NPC was determined according to the eighth edition of the Union for International Cancer Control (UICC) and the American Joint Committee on Cancer (AJCC) staging system.

For FFPE samples included in the ST analysis, each patient received at least two courses of PD-1 blockade (toripalimab) treatment, and the mean treatment period was 7.91 months. Efficacy assessment for each patient was assessed by two experienced oncologists based on RECIST v.1.1[77]. We obtained five non-responders (four with progressive disease, PD; and one with stable disease, SD) and seven responders (five with complete remission, CR; and two with partial remission, PR). We also collected corresponding prognostic information for patients with bulk transcriptome data (progression-free survival for the Bulk-RNA-seq cohort; progression-free and overall survival for the Microarray cohort).

## Single-cell data processing

We processed single-cell RNA and VDJ sequencing data of the 77 samples from previous studies, where the data were generated using 10X Genomics GemCode platform (Supplementary Data 1)[27,64,71–73]. Among them, 20 samples with post-amplification cDNA were subjected to BCR fragment enrichment in this study. Sequencing libraries for BCR were separately constructed according to the 10X Genomics protocol CG000086. The average fragment size of a library was quantitated using Qseq100 (Bioptic, China). Each DNA library was loaded into a sequencing lane on a HiSeq X system (Illumina, USA) and sequenced with pair-end reads of 150 bp.

Raw sequencing data were aligned and quantified using the Cell Ranger Single-Cell Software Suit (version 3.0.1, 10X Genomics). Single-cell gene expression and immune cell receptor data (BCR and TCR) from the same cDNA library were processed using Cell Ranger count and Cell Ranger vdj implemented in the pipelines, respectively. The gene expression data was mapped to the human genome (GRCh38; http://cf.10Xgenomics.com/supp/cell-exp/refdata-cellranger-GRCh38-1.2.0.tar.gz) and EBV reference sequences (Akata; https://github.com/flemingtonlab/public/tree/master/annotation). The BCR and TCR enriched data were mapped to the VDJ reference sequence (http://cf.10Xgenomics.com/supp/cell-vdj/refdata-cellranger-vdj-GRCh38-alts-ensembl-2.0.0.tar.gz).

For the quality check procedure, we first identified doublets that were artefactual libraries generated from droplet encapsulation of more than two cells using R package "DoubletFinder" (version 2.0.3, https://github.com/chris-mcginnis-ucsf/DoubletFinder)[78] and additional examination of an abnormal cluster consisting of doublets with more than one cell type markers simultaneously. Doublets were removed for each sample, with an expected doublet rate of 0.05 and default parameters otherwise stated. Next, any cells were removed if had less than 1001 UMIs, expression of fewer than 501 genes, or over 25% UMIs linked to mitochondrial genes. Gene expression matrices for cells at each step were combined and converted to a Seurat object using the R package Seurat (version 4.0.0, https://satijalab.org/seurat), with log normalisation and linear regression using the NormalizeData and ScaleData functions, respectively.

## Annotation of cell type

First, we removed the potential batch effect that might be introduced because of independent sample processing and high-dimensional variables in single-cell sequencing using Harmony algorithm[79]. Next, cell clusters were identified using FindClusters function after reducing dimensionality after RunUMAP function implemented in Seurat, with a K parameter of 30 and default parameters otherwise stated. We annotated the clusters as different major cell types based on their average gene expression of well-known markers, including T (marker: *CD3D*), NK (*KLRF1*), myeloid (*AIF1*), malignant (*EPCAM*), B (*MS4A1*), plasma (*MZB1*), mast (*TPSB2*), pDC (*LILRA4*), endothelial (*VWF*), CAF (*COL1A1*), and cycling immune (*HMGB2*) cells.

By repeating the abovementioned steps (normalisation, dimensionality reduction, and clustering), we further identified sub-clusters and annotated them as different specific cell subtypes based on the average expression of respective gene sets in each major cell type. To identify marker genes for each sub-cluster within the major cell types (T, NK, myeloid, malignant, B, plasma, mast, pDC, endothelial, CAFs, and cycling immune cells), the expression profiles of the sub-cluster were contrasted with those of the other sub-clusters using Seurat FindAllMarkers function. Differential expression gene (DEG) analysis implemented in the function compared all the genes in the two data-sets using the default two-sided non-parametric Wilcoxon rank sum test. A significant DEG was determined if it had the Bonferroin-adjusted $P$ value lower than 0.05 and an average binary logarithm fold-change of expression of at least 0.25 for all clusters. Any cluster with multiple well-defined marker genes of different cell types and a high number of UMI was considered cell contamination and removed from downstream analysis. The selection criteria for the marker gene included 1) with top ranking at the DEG analysis for the corresponding cell cluster, 2) with strong specificity of gene expression meaning high expression ratio within the corresponding cell cluster but low in other clusters, and 3) with literature supports that it's either a marker gene or functional relevant to the cell type. For each cluster (like C1) of a major cell type (like CD4$^+$ T cells), we assigned a cluster identifier with a marker gene (like LEF1) as "CD4_C1_LEF1". For each sample, we assigned a sample identifier with the last name of the first author (like Liu), sample number (like 1802), and source of the sample (Tumour) marked in the corresponding study as "Liu_1802_Tumour".

## Sample processing and data generation for spatial transcriptomics

We performed high-resolution spatial transcriptomic (ST) assays using two platforms, including Stereo-seq (BGI-Shenzhen, China) and Visium Spatial for FFPE Gene Expression Kit (10X Genomics, USA) for NPC samples. The Stereo-seq FF platform using freshly frozen (FF) samples offers advantages of a wide-field view and high resolution for the detection of minute structures within tumours. The Visium FFPE platform provides feasibility to handle FFPE samples, allowing us to retrospectively assess the efficacy of immunotherapy in NPC patients.

We performed ST assays using Stereo-seq technology (BGI-Shenzhen, China) following the manufacturer's instructions. First, the tissue block was rinsed with cold PBS (Gibco™, cat. no. 10010023), immersed in a pre-cooled tissue storage solution (Miltenyi Biotec, Germany), and embedded with pre-cooled OCT (optimal cutting temperature compound; Sakura, Cat. no. 25608-930) in a −30 °C microtome (Thermo Fisher, USA) within 30 min after biopsy. Second, appropriate samples were selected based on qualified RNA integrity. In brief, total RNA was extracted from each frozen sample block with a RNeasy Mini Kit (Qiagen, Cat. no. 74104) following the manufacturer's instructions. The RNA quality of each sample was evaluated by 4200 Bioanalyzer (Agilent, USA), and samples with an RNA Integrity Number (RIN) greater than seven were selected. Third, a spatial transcriptomic RNA library was constructed for each sample. In brief, a cryosection of 10 μm thickness cut from OCT-embedded tissue was quickly placed on the chip, followed by incubation at 37 °C for 3 min, and then fixation in pre-cooled methanol at −20 °C for 40 min. The fixed tissue sections were stained with Qubit ssDNA dye (Thermo Fisher, USA) to check the tissue integrity before fluorescent imaging. The tissues were treated according to the manufacturer's recommendation, and the optimisation procedure showed an optimal permeabilization time of 18 min of digestion and release of RNA from the tissue slide. Subsequently, RNA released from the permeabilized tissue was reverse transcribed for 1 h at 42 °C. The tissue sections were digested with a tissue removal buffer at 42 °C for 30 min. The cDNA-containing chip was then subjected to cDNA-release enzyme treatment overnight at 55 °C. The released cDNA was further amplified using a cDNA HIFI PCR mix (MGI, China). Finally, approximately 20 ng of cDNA was fragmented to 400-600 bp, amplified for 13 cycles, and purified to generate a DNA nanoball library that was sequenced with the single-end 50 + 100 bp strategy on an MGI DNBSEQ sequencer (MGI, China).

We also performed ST assay for NPC FFPE samples using Visium Spatial for FFPE Gene Expression Kit (10X Genomics, USA) following the manufacturer's instructions. First, total RNA was extracted from each NPC FFPE sample block with a RNeasy FFPE kit (Qiagen, Cat. no. 73504) to evaluate the RNA quality of FFPE samples, and samples with a DV200 greater than 50% (percentage of RNA fragments longer than 200 nucleotides) were selected for Visium spatial transcriptomics assay. Subsequently, the paraffin-embedded sections (5 μm thickness) adhered on the Visium slides were deparaffinized using xylene and graded ethanol and Hematoxylin-Eosin staining according to the 10X Genomics protocol CG000409. After Hematoxylin-Eosin brightfield

images were acquired with microscopy (Aperio Versa 8, Leica), decrosslinking was performed to release RNA sequestered by the formalin fixation. Then, whole human transcriptome probe pairs were added to the slide for hybridisation to their complementary target RNA, followed by the addition of a ligase to seal the junction between the probe pairs hybridised to RNA. RNase treatment and permeabilization were conducted to release the single-stranded ligation products, which were then captured on the Visium slides. Finally, ligated probe products were released from the slide and carried forward for library construction according to the 10X Genomics protocol CG000407. Each DNA library was loaded into a sequencing lane on a NovaSeq 6000 system (Illumina, USA) and sequenced with pair-end reads of 150 bp.

We further performed ST assay for freshly frozen EBV⁺ GaC samples using the Visium Spatial for FF Gene Expression Kit (10X Genomics, USA) following the manufacturer's instructions. OCT-embedding and RNA quality control were performed as abovementioned for Stereo-seq. The 10-μm section was placed on the pre-chilled Optimisation slides (Visium, 10X Genomics, PN-1000193), and the optimal lysis time was determined. The tissues were treated following the manufacturer's recommendations, and the optimisation procedure showed an optimal permeabilization time of 18 min of digestion and release of RNA from the tissue slide. The frozen sections adhered on the Visium slides were performed with methanol fixation and Hematoxylin-Eosin staining according to the 10X Genomics protocol CG000160. After second strand synthesis and denaturation, we amplified cDNA for library construction according to the 10X Genomics protocol CG000239. Each DNA library was loaded into a sequencing lane on a NovaSeq 6000 system (Illumina, USA) and sequenced with pair-end reads of 150 bp.

### Spatial transcriptomic data processing and analysis
Raw data for Visium assay in FASTQ files with manually aligned histological images were aligned and quantified using the Space Ranger Software Suit (version 1.3.1, 10X Genomics). Spatial gene expression was processed using Space Ranger count implemented in the pipelines. The spatial data of Visium FF were mapped to the human genome (GRCh38; http://cf.10Xgenomics.com/supp/cell-exp/refdata-cellranger-GRCh38-1.2.0.tar.gz) and EBV reference sequences (Akata; https://github.com/flemingtonlab/public/tree/master/annotation), and the spatial data of Visium FFPE were mapped to Visium Human Transcriptome Probe Set v1.0 GRCh38-2020-A). Raw data for Stereo-seq assay in FASTQ files were automatically processed using the SAW software (https://github.com/BGIResearch/SAW), in which the reads were decoded, trimmed, deduplicated, and mapped to human and EBV reference genomes. Each capture spot in the Stereo-seq chips was 220 nm in diameter, with a centre-to-centre distance of 500 nm between two adjacent spots. By contrast, each capture spot in the Visium chips was 55 μm in diameter, with a centre-to-centre distance of 100 μm between two adjacent spots. To ensure sufficient number of genes for annotation and accurate clustering, we annotated the Stereo-seq data using a bin100 (containing $100 \times 100 = 10,000$ spots), which covered an area of approximately 49.72 μm×49.72 μm.

Count matrices were loaded into Seurat (version 4.0.0) and BayesSpace (version 1.0.0)[80] for all subsequent data filtering, normalisation, filtering, dimensional reduction, and visualisation. Data normalisation was performed on independent tissue sections using the variance-stabilising transformation method in the SCTransform function implemented in Seurat. To enhance the resolution of the clustering map, we segmented each spot into four and six sub-spots for Stereo-seq and Visium cohorts, respectively, and leveraged spatial information using the spatialEnhance function in BayesSpace.

To illustrate the cell-cell interaction in situ, we used SpaGene[81] to identify cell-cell communications mediated by colocalized ligand and receptor pairs according to the spatial transcriptomic data. For each ligand-receptor pair, SpaGene estimated the spatial connectivity of the subnetwork comprising only connections between spots with both high expression of the ligand and the receptor. SpaGene used the Earth's mover's distance based on the degree distribution of the subnetwork to quantify its spatial connectivity.

### Calculation of functional module scores
To evaluate the potential functions of a cell cluster of interest, we calculated the scores of functional modules for the cluster, using the AddModuleScore function in Seurat at the single-cell level. The average expression levels of the corresponding cluster were subtracted by the aggregated expression of control feature sets. All analysed genes were binned based on averaged expression, and the control features were randomly selected from each bin. The functional modules included exhausted, activation, and stem-like signature scores for CD8⁺ T cells and apoptosis signature (extrinsic apoptosis) score for spatial spots. To evaluate the immunosuppressive signatures, we calculated the immunosuppressive score according to the expression levels of immunosuppressive genes including *CD47*, *PVR*, *CD276*, *LGALS9*, *ADORA2B*, *ADAM10*, *HLA-G*, *CD274*, *FASLG*, *TGFB1*, and *IL10*. The genes for each module are listed in the Supplementary Data 4.

### Deconvolution of the relative cell composition in Bulk-RNA-seq and Microarray cohorts
To ascertain the proportionate composition of cells, CIBERSORTx[82] is utilised for each specimen in both Bulk-RNA-seq and Microarray cohorts, adhering to the software guidelines. The gene signature of each single-cell RNA cluster was employed as a reference in CIBERSORTx. To minimise the bias from variable cell-type capture rates, we chose 100 cells randomly from each cluster identified in our scRNA-seq analysis. Following the determination of each cluster's composition, we consolidated subclusters into their respective major cell types. This categorisation included grouping all tumour cell clusters into malignant cells, combining B and plasma cells into B cells, and aggregating CD4⁺ and CD8⁺ T cells into T cells. NK cells comprised δγ T cells, NK cells, and MAIT cells. Endothelial cells were a collection of lymph vessel and blood vessel cells, while CAF encompassed all fibroblast subclusters. Myeloid cells included monocytes, macrophages, dendritic cells, and mast cells. In our downstream analysis, we excluded samples without malignant cells, minimising the potential influence of tumour content due to sampling bias during NPC patient recruitment at various clinical stages (Supplementary Fig. 15a). A few single-cell RNA samples do not contain malignant cells and are excluded in the cell proportion analysis.

### Correlation and survival analyses
For Bulk-RNA-seq data, raw data in FASTQ files were processed using STAR (version 2.6.1) and further normalised as transcripts per million (TPM) to exclude potential bias using RSEM (version 1.3.3). For Microarray data, raw data was obtained using Command Console Software 4.0 (Affymetrix) with the default setting and were further processed and normalised using Transcriptome Analysis Console (version 4.0.1, ThermoFisher Scientific). For Microdissection data, TPM matrix was obtained from Tay et al.[28]. We deconvolved the cell composition of each sample for Bulk-RNA-seq, Microarray, and Microdissection cohorts using single-sample gene set enrichment analysis (ssGSEA, https://github.com/broadinstitute/ssGSEA2.0), which estimates the relative proportion of cell clusters from gene expression profile. Using the Pearson correlation test, ssGSEA scores were subsequently correlated within cell clusters. We defined signature genes for estimated cell clusters based on the top 100 DEGs among cell clusters (Supplementary Data 2) and the expression levels of pan-Ig using *IGHA1*, *IGHA2*, *IGHG1*, *IGHG2*, *IGHG3*, and *IGHG4*.

For survival analysis, Receiver operating characteristic (ROC) was used to determine the optimal cut-off value of gene expression for patient stratification. Kaplan-Meier analysis was conducted to reveal

the prognostic ability of the expression levels of genes or the fractions of cell clusters in NPC samples with survival data, and a two-sided cox test was performed to compare survival outcomes between patient groups with high and low expression levels of genes or fractions of cell clusters.

## Relative quantification of EBV DNA load in the Bulk-RNA-seq NPC cohort

To determine EBV DNA load in a sample, we performed quantitative PCR (qPCR) using EBV-encoded BALF5 as the marker of EBV genomic DNA and β-actin as the internal reference gene. In brief, genomic DNA was extracted from tumours, peripheral blood, or EBV-positive C666-1 cells, using a DNeasy Blood & Tissue Kit (Qiagen, Cat. no. 69504). 1 μg of tumour tissue or peripheral blood DNA and 60 ng of C666-1 cell line DNA were subjected to qPCR using SYBR Premix Ex Taq kit (Takata, Cat. no. RR390W) and their corresponding primer pairs (Supplementary Data 7), following the manufacturer's instructions. EBV DNA copy number in tumour tissue or peripheral blood was calculated using the Livak (2-$\Delta\Delta$Ct) method with the Ct values of the target gene BALF5 and the internal reference gene β-actin, relative to the positive control EBV-positive NPC cell line C666-1. According to the mean fold of difference values for EBV DNA copy number, the patients were divided into two groups (high/low EBV load).

## Definition of spatial regions

For ST cohorts, we defined tumour cell aggregates (TCA) using NPC malignant cell makers (*EPCAM*, *KRT13*, *KRT8*, and *KRT5*) with tissue morphology identified by pathologists. We also defined B and T cell aggregates as generalised TLS regions with specific expression of markers for B and T cells, respectively, according to DEG analysis (B cell marker: *CD19*, *MS4A1*, *CD79A*, and *CD79B*; T cell marker: *CD2*, *CD3D*, *CD3E*, *CD3G*, and *CD7*). We further defined the remaining regions as stromal regions. Moreover, we quantified the spatial distribution and proportion of the cell clusters using the AddModuleScore function in Seurat with the signature genes based on the top 100 DEGs among cell clusters in single-cell RNA data (Supplementary Data 2). When calculating the cell signature score for TLS regions, we included larger immune-infiltrated regions that have an expression profile similar to TLS. We defined EBV-related TCA in the FFPE NPC samples of the Visium cohort using the expression profile of the top 100 DEGs of EBV^high NPC malignant cells from our previous study[27], and then divided malignant spots into EBV^high-(>75%), EBV^mid-(>25% and <75%), and EBV^low-(<25%) TCA according to the expression quartiles. We also defined EBV^high-TCA in FF NPC samples of the Stereo-seq cohort and in FF GaC samples of the Visium cohort using EBV encoded genes (*RPMS1/A73*, *BARF0*, *BALF3/4*, *LMP-1/BNLF2a/b*, *LMP-2A/B*, and *EBNAs*), pan-Ig using *IGHA1*, *IGHA2*, *IGHG1*, *IGHG2*, *IGHG3*, and *IGHG4*, and plasma spot using *CD79A*, *CD79B*, *MZB1*, *XBP1*, *IGHA1*, *IGHA2*, *IGHG1*, *IGHG2*, *IGHG3*, and *IGHG4*.

For the Microdissection cohort, we followed the region definitions as previously reported[28], including dysplastic epithelium (DYS), normal nasopharyngeal epithelium (NAT), and normal tumour-adjacent epithelium (NOR) regions. To ensure comparability with the definitions of other regions in our paper, we redefined the tumour area as tumour cell aggregates (TCA) and the immune cell-rich area as immune cell aggregates (ICA)[28].

## BCR and TCR repertoire analyses

Raw sequencing reads of the BCR and TCR libraries were processed by Cell Ranger VDJ pipeline (version 3.0.1, 10X Genomics) with the default parameters to assemble contigs that represent the best estimate of transcript sequences in each cell.

All IgH sequences were annotated with AssignGenes.py[83] (ChangeO, version 1.0.0) and IgBLAST[84] (version 1.17.0). Clonally related

sequences were identified using DefineClones.py with the nearest neighbour distance threshold determined by distToNearest[83] (Shazam, version 1.0.2). Germline sequences were inferred using CreateGermlines.py, and SHM frequencies were calculated with observedMutations. SHM frequencies of >0.02 were annotated as 'high', 0 to 0.02 as 'low', and 0 as 'none'.

R package STARTRAC[85] (version 0.1.0) was used to assess the enrichment of BCR/TCR in various B/T cell clusters. In brief, the degree of cell linking between different clusters of B cells was determined by the transition-index score. TCR diversity (Shannon-index score) was calculated using "1-expansion-index score." For the detailed pipeline, please refer to the STARTRAC website (https://github.com/Japrin/STARTRAC).

## Pathway enrichment analysis

To explore the potential functions of the heterogenous B, T, and stromal cells, we performed gene set variation analysis (GSVA, version 1.34.0). Seurat output data was performed for GSVA, using the Molecular Signatures Database (MSigDB)[86], including hallmark, C1 positional, C2 curated, C3 motif, C4 computational, C5 GO, C6 oncogenic, and C7 immunologic gene sets with default parameters. To compare the different signalling pathway enrichment in plasma cells at early and advanced stages, we performed the gene set enrichment analysis (GSEA, version 3.0) using the MSigDB.

## Developmental trajectory analysis

To characterize the potential process of functional changes and determine the potential lineage differentiation among diverse B cells, we performed pseudotime trajectory analyses using diffusion map[87]. For dimension reduction, we calculated diffusion map using batch-corrected shared space output by Harmony as abovementioned. The diffusion map's first three principal components were shown in 3D plot. Naive B cells were randomly selected as the root cell for diffusion pseudotime computation.

For better illustrating GC B cell developmental process, GC B cells including B_C8_STMN1 and B_C9_RGS13 were extracted from B and plasma cells. We then, performed principal component analysis (PCA) using the mean-centered expression of 91 pre-selected genes[12] known to be involved in B cell differentiation. For better visualisation, PC1 x PC2 cartesian coordinates were converted into polar coordinates and a constant value was added to the radius to move cells out from the origin/pole. Dark zone and light zone scores were calculated using AddModuleScore function implemented in Seurat package with pre-defined gene signatures[31].

For CD8+ T cell clusters, we performed pseudotime analysis using Monocle3[88] (version 0.0.2; http://cole-trapnell-lab.github.io/monocle3/). Briefly, Seurat output data for specific clusters were fed directly into Monocle3, followed by the removal of batch effect using the align_cds function. Cell trajectory was calculated by using learn_graph function. Cell differentiation trajectory was inferred after dimension reduction and cell ordering with the default parameters implemented in Monocle3. Gene expression along pseudotime data was extracted from the output of plot_genes_in_pseudotime function and was used to plot genes along pseudotime of lineages using ggplot2 (version 3.3.3) in the R package.

## Cell-cell interaction analysis

To illustrate the interaction potential between two cells, we used CSOmap (version 1.0, https://github.com/lijxug/CSOmapR) to construct a 3D pseudo space and calculate the significant interaction for CD8+ T and malignant cells[89]. Then, we used a normalised connection to investigate the interaction potentials between CD8+ T and malignant cells. For a cluster pair, a normalised connection was calculated by dividing its corresponding connection value by the product of their

respective cell numbers. Meanwhile, to highlight the key ligand-receptor pairs function in the interaction, we also examined the contribution output by CSOmap.

To compare the differences in mutual regulatory networks among cells from different sources, we used CellChat[90] (version 1.0.0, https://github.com/sqjin/CellChat) with the normalised counts by Seurat and the standard pre-processing functions identifyOverExpressedGenes, identifyOverExpressedInteractions, and projectData. As for the reference database, we included the secreted signalling pathways and the precompiled human protein-protein and extracellular matrix (ECM)-receptor Interactions as a priori network information. The core functions computeCommunProb, computeCommunProbPathway and aggregateNet were applied with standard parameters and fixed randomisation seeds. The function netAnalysis_computeCentrality was applied on the netP data slot to compute the network centrality scores.

### Infer CNV analysis
To identify malignant epithelial cells, we utilized inferCNV (https://github.com/broadinstitute/inferCNV) to detect somatic alterations in large-scale chromosomal copy number variants, including gains or losses, at the single-cell level. The raw single-cell gene expression data was extracted from the Seurat object following the software recommendations. As a control reference, a public single-cell dataset derived from normal epithelial cells was incorporated (GEO accession number: GSE121600)[91]. The inferCNV analysis was conducted with default parameters.

### Multiplex IHC and IF staining assays
For the tissue samples kept in the formalin, dehydration and embedding in paraffin were performed according to routine methods. The paraffin-embedded sections (5 μm thickness) adhered on the glass slides were dewaxed, rehydrated, subjected to the blockade of endogenous peroxidase activity, and antigen retrieval at a high temperature. Subsequently, the sections were processed further for IHC or multiplex IHC staining assays.

The protein expression of IgG/IgA was examined using IHC staining of FFPE specimens from NPC patients. FFPE sections were incubated with the primary antibody against IgG/IgA (anti-IgG, ZSbio, Cat no. ZA-0448; anti-IgA, ZSbio, Cat no. ZA-0446) followed by the incubation with a secondary antibody (Zsbio, Cat no. PV-6001). Next, the sections were stained with 3,3′-diaminobenzidine (DAB, Zsbio, Cat no. ZLI-9017) and counterstained with hematoxylin (Beyotime, Cat no. C0107) for nuclei. IHC scores were determined by the percentages of stained cells and the intensities of colour reaction for the stained sections.

To determine TLS-associated cell clusters (B, CD4+ T, CD8+ T cells, and fibroblasts) and the spatial contacts of apoptotic malignant and plasma cells, we performed multiplex IHC staining assays using the PANO 7-plex IHC kit (Panovue, China) according to the manufacturer's instructions. Briefly, FFPE slides were incubated with blocking antibody diluent at room temperature for 10 min, then incubated overnight at 4 °C with primary antibodies. The slides were then incubated with the secondary antibody (HRP polymer, anti-mouse/rabbit IgG) at room temperature for 50 min. Subsequently, fluorophore (tyramide signal amplification or TSA plus working solution) was applied to the sections, followed by heat treatment with a microwave. The primary antibodies were applied sequentially, followed by incubation with the secondary antibody and TSA treatment. Nuclei were stained with DAPI after all the antigens had been labelled. Multispectral images for each stained slide were captured using a multispectral scanner (Akoya, USA). All experiments were performed in at least three biological replicates. Primary antibodies included anti-CD20 (Abcam, Cat. no. ab9475, 1:50), anti-CD4 (ZSbio, Cat. no. ZA-0519), anti-CD8A (ZSbio, Cat. no. TA802079), anti-FAP-α (Abcam, Cat. no. ab207178, 1:200), anti-CXCL13 (NOVUS, Cat. no. NBP2-16041, 1:100; R&D systems, Cat.

no. MAB8012, 1.7 μg/mL), anti-CASP3 (CST, Cat. no. 9664, 1:1000), anti-CD3 (ZSbio, Cat no. ZA-0503), anti-CD68 (ZSbio, Cat. no. TA802952), anti-CD56 (ZSbio, Cat no. ZM-0057), anti-PanCK (ZSbio, Cat. no. ZM-0069), anti-CXCR5 (Abcam, Cat. no. ab254415, 1:1000), anti-PD1 (ZSbio, Cat. no. ZM-0381), anti-TIM3 (CST, Cat. no. 45208, 1:400), anti-CD70 (CST, Cat. no. 69209, 1:500), and anti-CD27 (Proteintech, Cat. no. 66308-1-lg, 1:1000).

### Multiplex IHC analyses
Positive stain quantification and spatial cell analysis were performed using HALO 3.2 software. We classified the staining intensity of CASP3 as weak, moderate, and strong for calculating the IHC score. Proximity analysis was used to quantify the spatial relationship between any two cell or object populations. Once the cell data was plotted, proximity analysis was used to quantify total IgG+ plasma cell counts across the image and the number of CASP3+ tumour cells within the specified proximity distance of IgG+ plasma cells (20 μm). For immune cell infiltration scores, we defined 1, 2, 3, and 4 as representing 0 - 25%, 26 - 50%, 51 - 75%, and 76 - 100% of the tumour area infiltrated by immune cells, respectively.

### in situ hybridisation of EBERs
To determine the EBV infection status of NPC biopsy samples, we performed in situ hybridization assay of the EBV-encoded EBERs (ZSbio, Cat no. ISH-7001). In brief, paraffin sections were first deparaffinized, blocked, and enzyme-treated. Then, probes and HRP-labelled anti-digoxin antibodies were added, followed by DAB as a substrate-chromogen and nuclear staining with haematoxylin. Subsequently, the sections were dehydrated, cleared, and mounted. Finally, the stained images of each section were captured under a microscope, and positive staining was identified in the cell nucleus.

### Cell lines and lentivirus
Human lung fibroblast cells (MRC-5) were obtained from Zhejiang Meisen Cell Technology Co., Ltd. and were cultured in RPMI-1640 (Gibco™, Cat. no. C11875500BT) supplemented with 10% fetal bovine serum (FBS; Gibco™, Cat. no. 10099141), 1% Penn/Strep (Gibco™, Cat. no. 15070063). Human embryonic kidney 293 T cells were obtained from Cell Bank of Type Culture Collection of the Chinese Academy of Sciences, Shanghai Institute of Cell Biology, Chinese Academy of Sciences. Human NPC cell lines (HK-1 and S26) were kindly gifted by Professor Chaonan Qian. All cell lines were cultured in Dulbecco's Modified Eagle Medium (DMEM, Gibco™, Cat. no. C11995500BT) supplemented with 10% FBS, 1% Penn/Strep. All cell lines were incubated at 37 °C, 5% CO₂ humidified incubator. For over-expression of CXCL13 or TNFSF13B in MRC-5 cells, CXCL13 or TNFSF13B were cloned into pCDH-puro lentiviral vector, which was packaged and transduced into 293T cells to generate lentivirus. MRC-5 cells were infected with the lentivirus and selected against puromycin. Cell lines were authenticated by short tandem repeat (STR) analysis. No evidence of mycoplasma contamination was observed using mycoplasma detection kit (Vazyme, Nanjing, China).

### Co-culture assay for the assessment of cell-cell interactions
PBMCs were isolated using a leucocyte separation solution, following the manufacturer's instruction (HISTOPAQUE-1077; Sigma-Aldrich, Cat. no. 10771). Thereafter, the PBMC were washed and resuspended in RPMI-1640 (Gibco™, Cat. no. 11875500) supplemented with 10% FBS. B cell fraction was isolated from the PBMC using MojoSort Human B Cell (CD43⁻) Isolation Kit (Biolegend, Cat. no. 480061). Human CAFs were extracted from fresh NPC tissues according to previous study[92]. Briefly, fresh tissues were cut into pieces of 2 to 3 mm and cultured in DMEM medium with 10% FBS and 1% Penn/Strep over four weeks in a 6-well plate. Anti-FAP-α and anti-E-Cadherin were used to identify CAFs through western blotting assay. For all experiments, fibroblasts were

used between passages 2 and 5. All cells were cultured in a humidified incubator containing 5% $CO_2$ at 37 °C.

Co-culture assay was performed to measure the binding interaction between CAFs and B cells. Briefly, the abovementioned CAFs were further enriched for VCAM1⁺ or VCAM1⁻ CAFs using fluorescence-activated cell sorting (FACS) with VCAM1 staining (BioLegend, Cat. no. 305805). Then, VCAM1⁺ or VCAM1⁻ CAFs ($2 \times 10^4$ per well) were seeded on a 96-well plate for one day. On the second day, CAFs were incubated with the presence or absence of neutralising antibody anti-VCAM1 Antibody (30 μg/ml each, R&D systems, BBA5) for one hour prior to the adhesion assay. B cells ($6 \times 10^4$ per well) were added and incubated with VCAM1⁺ or VCAM1⁻ CAFs for 30 min. The 96-well plate was gently tapped twice, followed by removing the medium and unbound B cells. Cell density was measured by Cell Counting Kit-8 (CCK8; Dojindo, Cat. no. CK04) following the manufacturer's instructions. Optical density (OD) was measured at 450 nm. B cells bound to CAFs were calculated by the OD of co-cultured B cells and CAFs subtracting that of CAFs cultured alone.

Co-culture assay was also performed to assess the effect of CAFs on the IgG production of B cells. Briefly, peripheral blood B cells untreated or co-cultured with VCAM1⁺ or VCAM1⁻ CAFs were incubated in RPMI-1640 supplemented with 10% FBS, 1% Penn/Strep, CpG oligodeoxynucleotide (1 μg/ml; Invivogen, San Diego, CA), and anti-IgM (1 μg/ml; Jackson ImmunoResearch Laboratories, Bar Harbor, ME). Cultures were incubated at 37 °C, 5% $CO_2$ for the indicated times. After the replacement of half of the culture medium on day 3, IgG production in the supernatant was detected on day 6 with an enzyme linked immunosorbent assay (ELISA) kit (MultiSciences, Cat. no. 70-EK171-96).

To assess the effect of TNFSF13B expression in fibroblasts on the IgG production of B cells, the abovementioned MRC-5 with or without TNFSF13B overexpression ($3 \times 10^4$ per well) were seeded on a 96-well plate and allowed to adhere overnight. Cell supernatant was discarded and cells were washed with PBS. Then fresh RPMI-1640 containing 10% FBS and 1% Penn/Strep was added to continue culture. After 48 hours, cell supernatants were collected, clarified by centrifugation, and incubated with the presence or absence of neutralising antibody anti-TNFSF13B Antibody (20 μg/ml, R&D systems, AF124) for one hour. Then the supernatants were added to the original wells containing MRC-5 with or without TNFSF13B overexpression. B cells suspension of 100 μl ($1 \times 10^6$ per well) was added and incubated with MRC-5 cells in RPMI-1640 supplemented with 10% FBS, 1% Penn/Strep, 1 μg/ml CpG oligodeoxynucleotide, and 1 μg/ml anti-IgM. After the replacement of half of the culture medium in the presence or absence of the blocking antibody with a final concentration of 20 μg/ml on day 3, IgG production in the supernatant was detected as previously mentioned[93].

For the migration assay, B cells were firstly activated with in RPMI-1640 supplemented with 10% FBS, 1% Penn/Strep, 1 μg/ml CpG oligodeoxynucleotide, and 1 μg/ml anti-IgM for 48 hours before assay. MRC-5 cells with or without overexpression of CXCL13 ($1.5 \times 10^5$ per well) were seeded on a 24-well plate and allowed to adhere overnight. Cell supernatants were discarded and washed with PBS. Then fresh RPMI-1640 containing 10% FBS and 1% Penn/Strep was added to continue the culture for 24 hours, then cell supernatants were collected, clarified by centrifugation, and incubated in the presence or absence of neutralising anti-CXCL13 Antibody (4 μg/ml; Thermo, PA5-47035) for one hour prior to the migration assay. 600 μl of the supernatant was added to a bottom chamber of the 24-well transwell plate (Corning, 3421), whereas activated B cells ($1 \times 10^6$ per well) were added into the top chambers. Cells were kept for 2 hours at 37 °C, 5% $CO_2$ incubator to allow migration. Then number of B cells in the bottom chamber were counted, and divide by the total number of B cells to calculate the percentage of B cell migration.

All experiments were performed in at least three biological replicates.

## Western blotting assay

Western blotting assay was performed to determine protein expression using the conventional method. In brief, protein lysates were prepared in cell lysis buffer (CST, Cat. no. 9803S) with a protease inhibitor cocktail (Beyotime, Cat. no. P1005), followed by separation on sodium dodecyl sulphate-polyacrylamide gel electrophoresis (SDS-PAGE) and transfer to polyvinylidene difluoride (PVDF) membrane (Merck Millipore, USA) with wet tank blotting (Bio-Rad, USA). After incubations with primary and secondary antibodies, signals were detected by using ChemiDoc Touch Imaging System (Bio-Rad, USA). Primary antibodies included anti-FAP-α (Abcam, Cat. no. ab207178, 1:1000), anti-E-Cadherin (CST, Cat. no. 14472, 1:1000), and anti-GAPDH (ABclonal, Cat. no. AC002, 1:5000). Secondary antibodies included anti-rabbit IgG (CST, Cat. no. 7074, 1:2000) and anti-mouse IgG (CST, Cat. no. 7076, 1:2000).

## Quantitative PCR assay

Total RNA was extracted from samples using TRIzol reagent (Invitrogen, Cat. no. 15596026), and reverse transcription was performed using oligo (dT) primers and M-MLV Reverse Transcriptase (Promega, Cat. no. M1705) according to the manufacturers' instructions. Real-time quantitative reverse transcription PCR (qRT-PCR) was performed to determine the expression level of genes using the TB Green Premix Ex Taq kit (Takata, Cat. no. RR420A) and the corresponding primer pairs (Supplementary Data 7).

## Statistics and reproducibility

All statistical analyses were performed using R, including two-sided paired student t-test, Wilcoxon test, Pearson correlation test, Chi-square test, and Kruskal–Wallis test. $P < 0.05$ was considered as statistical significance. Multiplex IF and IHC staining assays were confirmed in at least three independent samples. Sample size for scRNA-seq and ST was determined by the availability of patient samples. No statistical tests were performed for sample size calculation but it was sufficient for this proof-of-concept study. The exact number of samples used per figure is informed in each figure. The patients with nasopharyngeal carcinoma and EBV⁺ gastric cancer were recruited randomly in this study. Investigators were blinded to patient origin.

## Reporting summary

Further information on research design is available in the Nature Portfolio Reporting Summary linked to this article.

# Data availability

The raw sequence data (BCR and spatial data) generated in this study have been deposited in the Genome Sequence Archive (GSA) of the National Genomics Data Center (NGDC), Beijing Institute of Genomics (China National Center for Bioinformation), Chinese Academy of Sciences, under accession number HRA006885. The raw data are available under controlled access due to data privacy laws related to patient consent for data sharing. The data should be used for research purposes only. According to the guidelines of GSA-human, all non-profit researchers are allowed access to the data, and the Principal Investigator of any research group can apply for the data following the guidelines at the GSA database portal (https://ngdc.cncb.ac.cn/gsa-human/). The response time for access requests is approximately 10 working days. Once access has been granted, the data will be available for download within one month. The user can also contact the corresponding author directly for enquiries. This study's processed single-cell VDJ-seq and spatial transcriptomic data can be obtained from Gene Expression Omnibus (GEO) with an accession number of GSE206245 and the Research Data Deposit (RDD; http://www.researchdata.org.cn/) with an accession number of RDDB2024955995. The NPC single-cell RNA and bulk RNA-seq publicly available data used in this study are available in the GEO database under accession code GSE162025[27],

GSE150825[64], GSE150430[71], GSE102349[74], GSE121600[91], GSA database under accession code HRA000087[73] and the URL https://www.science.org/doi/10.1126/sciadv.abh2445[28]. GaC Bulk mRNA-seq expression data (normalised) generated by The Cancer Genome Atlas (TCGA) on primary stomach adenocarcinoma were downloaded from NCI Cancer Genomic Data Commons (NCI-GDC: https://gdc.cancer.gov). The remaining data are available within the Article, Supplementary Information, or Source Data file. Source data are provided with this paper.

## Code availability
The codes used for all processing and analysis is available at https://github.com/yliuup/NPC-TLS (https://doi.org/10.5281/zenodo.12728508)[94].

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

## Acknowledgements

This study is supported by the National Natural Science Foundation of China and the Macao Science and Technology Development Fund (FDCT-NSFC; 82261160657), the National Key R&D Program of China (NKRDPC; 2022YFC3400901), the National Natural Science Foundation (82130078), the Sci-Tech Project Foundation of Guangzhou City (201707020039), the Guangdong Innovative and Entrepreneurial Research Team Program (2016ZT06S638), the Chang Jiang Scholars Program (J.-X.B.), and the Hong Kong Research Grant Council (RGC) Theme-based Research Scheme Funds (T12-703/22-R and T12-703/23-N). We thank all the participants in the study and staffs at the High-Throughput Analysis Platform (HTAP) of SYSUCC for data generation and processing.

## Author contributions

J.-X.B. conceived and supervised this study; R.-J.P. provided clinical samples and information; Y.L., S.-Y.Y., X.-Z.W., D.-M.C., S.H., D.M., R.-C.N., Z.-H.R., P.-P.W., J.Z., Q.C., M.-Y.C., J.-P.Y., J.C.D., H.-Q.M., and X.J.X. performed sample preparation, library construction, and data generation for scRNA-seq and ST; Y.L., S.H., D.-M.C., Y.Z., and G.-W.L. performed bioinformatics analyses; S.-Y.Y., Y.L., X.-Z.W., Y.-F.W., D.M., P.X., and C.-L.L. performed functional validations; Y.L. and J.-X.B. interpreted the results and wrote the manuscript. All authors read and approved the final manuscript.

## Competing interests

The authors declare no competing interests.

## Additional information

¹State Key Laboratory of Oncology in South China, Guangdong Key Laboratory of Nasopharyngeal Carcinoma Diagnosis and Therapy, Guangdong Provincial Clinical Research Center for Cancer, Sun Yat-sen University Cancer Center, Guangzhou 510060, P. R. China. ²The Eighth Affiliated Hospital, Sun Yat-sen University, Shenzhen 518033, P. R. China. ³Department of Anesthesiology, Sun Yat-sen University Cancer Center, Guangzhou 510060, P. R. China. ⁴Department of Radiation Oncology, Affiliated Cancer Hospital & Institute of Guangzhou Medical University, Guangzhou 510000, P. R. China. ⁵Department of Laboratory Medicine, Zhujiang Hospital, Southern Medical University, Guangzhou, Guangdong 510282, P. R. China. ⁶Department of Pathology, Guangdong Provincial People's Hospital (Guangdong Academy of Medical Sciences), Southern Medical University, Guangzhou, Guangdong 510080, P. R. China. ⁷Department of Immunology, Zhongshan School of Medicine, Sun Yat-sen University, Guangzhou, Guangdong 510080, P. R. China. ⁸Collaborative Innovation Center for Cancer Medicine, Nanjing Medical University, Nanjing, Jiangsu 211103, P. R. China. ⁹Division of Medical Oncology, National Cancer Centre Singapore, 30 Hospital Boulevard, 168583 Singapore, Singapore. ¹⁰These authors contributed equally: Yang Liu, Shuang-Yan Ye, Shuai He, Dong-Mei Chi, Xiu-Zhi Wang. ✉e-mail: beijx@sysucc.org.cn

