## [Peer Review File · Nature Communications]

Single-Cell and Spatial Transcriptome Analyses Reveal Tertiary Lymphoid Structures Linked to Tumour Progression and Immunotherapy Response in Nasopharyngeal CarcinomaREVIEWER COMMENTS

Reviewer #1, expertise in head and neck cancer and immunotherapy (Remarks to the Author):

This manuscript by Liu et al describes the single-cell and spatial transcriptomic analyses of tertiary lymphoid structures (TLS) in nasopharyngeal carcinoma (NPC), and associating these characteristics with clinical outcome, specifically with respect to immune checkpoint blockage.

General comments:

1. The clinicopathological characteristics of NPC patients who were evaluated in this study should be provided and summarized, not only their stage but also their prior treatment history. Based on Supplementary Table 1, there appears to be a mix of patients with early stage and late stage locally advanced disease, and also those treated with immune checkpoint blockade. Presumably the former group was treated with curative (chemo)radiotherapy, and the latter with systemic therapy. It is difficult to associate the findings in each analysis without clear articulation of the clinical status of the NPC patients being examined.

2. Figure 7 a-d suggest that TLS cell signature with the 7 cell clusters appear to be prognostic and predictive for immune checkpoint blockade. However, these are based on gene expression bulk RNAseq data and thus they are not able to be certain that these cell clusters are from TLS and not from other structures. How do the applicants confirm that these are from TLS?

Specific comments:

1. Line 94: LT_i should be spelled out

2. Lines 104-105: Please clarify in which cell populations BCL-2 (the anti-apoptotic molecule)

is upregulated to inhibit the growth of melanoma. This sentence is unclear.

3. Figure 2e: the expression of IgA and IgG in advanced NPC tumors shown in this figure seems non-specific, what EBV-related viral antigens are these antibodies directed against? The same question is for Figure 2d that demonstrates pan-immunoglobulin transcription levels being lower in advanced stages than at the early stages, how can the authors prove specificity of these antibodies against NPC antigens?

4. Figure 4g: please provide clinicopathological characteristics of these two independent cohorts (Bulk-RNA-seq and Microarray cohorts). How comparable are these cohorts to the NPC patient samples used in the current study?

5. What criteria were used to select specific cases for spatial transcriptomics? The details of the “microdissection cohort, n = 189” in Supplementary Figure 3 should be provided.

6. Supplemental Table 3-1: this table describing the 12 NPC patients treated with toripalimab and chemotherapy seems to be missing.

7. Figure 4d: please describe how the immunosuppressive scores were determined.

Reviewer #2, expertise in spatial transcriptomics, TME (Remarks to the Author):

Herein, the authors conducted integrative scRNA-seq and spatial transcriptomics analyses on various cohorts of nasopharyngeal carcinoma to elucidate the single-cell spatial organization of Tertiary Lymphoid Structures (TLS) and their relationship with tumor cell aggregates. The strength of this study lies in its utilization of different patient cohorts, including those who received immunotherapy, as well as its combined use of scRNA-seq and spatial transcriptomics, leading to the identification of cell subsets potentially influencing anti-tumor responses and immunotherapy outcomes. While I acknowledge these strengths, I would like to highlight both major and minor concerns regarding the study.

Major:

1- The novelty of some of the findings is not clearly established. For example, the role of CXCL13 produced by Cancer-Associated Fibroblasts (CAFs) in B cell recruitment and TLS formation has been previously reported (Rodriguez et al, Cell Rep). Similarly, the identification of CXCL13+CD8+ T cells within TLS and the Tumor Microenvironment (TME) has been observed in other cancer types (Workel et al, Cancer Immunol Res. 2019, Dai et al, J Immunother Cancer. 2021).

2- Although the integrative transcriptomics analysis has revealed intriguing findings, the study appears to lack essential validation and functional studies. For instance:

I-The study presented here demonstrates that plasma cells play a role in promoting the apoptosis of malignant cells in EBV-related tumors, likely through mechanisms such as ADCC and ADCP. These noteworthy findings have primarily emerged from multi-omics analysis at the transcriptomics level and certain correlation analyses. However, Figure 3 appears to lack in vitro and in vivo validation to confirm the killing effects of antibodies produced by B cells on EBV-positive TCA. Additionally, it remains unclear whether these killing effects are mediated by both NK cells and macrophages. Moreover, the underlying mechanisms responsible for the higher infiltration of plasma cells and IgG deposition within high EBV TCA have not been elucidated, and addressing these aspects would strengthen the study's conclusions.

II- In this study, CXCL13+ TLS-associated CAFs were observed to promote B cell adhesion and antibody production in NPC through multi-omics integrative analysis at the transcriptomics level, which was further validated with in vitro co-cultures. However, additional validation is required to solidify these findings. The cell-cell interaction analysis identified potential mechanisms for crosstalk between CXCL13+ TLS-associated CAFs and B cells, involving CXCL13-CXCR5, VCAM1-integrin, ICAM1-integrin, and TNFSF13B-related ligand-receptor pairs. However, in vitro blocking experiments are lacking to confirm the direct involvement of these mechanisms. The co-culture experiments (cell adhesion assay and co-culture assay) were performed on VCAM1+ versus VCAM1- CAFs, but they did not confirm the contribution of CXCL13-CXCR5 produced by CAFs in supporting B cell adhesion and antibody production. Furthermore, to provide direct evidence for the importance of CXCL13-CXCR5 in driving TLS

formation, B cell recruitment, and antibody production in NPC, it is necessary to conduct in vivo experiments in a mouse model lacking CXCR5. Additionally, performing IHC to show the expression of CXCR5 on B cells in proximity to CXCL13+ CAFs in the TLS would provide valuable insights.

III- Once again, validation regarding the presence of CXCL13+CD8+ T cells in TLS in NPC is lacking, with only transcriptomics analysis supporting their existence. IHC is essential to confirm the presence of this subset of T cells within the TLS of NPC and to investigate their phenotype, such as the expression of PD-1 and the absence of other exhaustion molecules. Moreover, the roles of this specific subset in TLS formation and anti-tumor immunity remain unclear. It is essential to conduct in vitro and in vivo experiments to confirm whether CD8_C8_CXCL13 cells indeed have the highest potential for recognizing and attacking malignant NPC cells, rather than solely relying on predictions based on bioinformatic analysis.

Additionally, the connection between CD8_C8_CXCL13 cells and malignant cells, which is predicted to occur through CXCR3-CCL20, CXCR3-CXCL10, and CXCR3-CCL19 interactions, as well as the co-stimulatory pathway (CD27-CD70), requires thorough validation. Inclusion of experimental evidence will significantly strengthen the findings and shed light on the functional relevance of CXCL13+CD8+ T cells within the TLS and their potential impact on anti-tumor immunity in NPC.

3- The manuscript lacks a solid transition between some of the data, creating a disconnect between key findings. For instance, the study has identified two critical populations, namely CXCL13+ CAFs and CD8_C8_CXCL13, within the TLS of NPC. However, there is no clear logical connection between these populations and their collaborative roles in TLS formation, B cell recruitment, and the regulation of anti-tumor immunity. Additionally, the relation between these subsets and the killing effects of plasma cells in EBVhigh TCAs is not well-established. As a result, the current format of the paper presents these findings as isolated islands in an ocean, rather than creating a cohesive and interconnected narrative.

Minor comments:

- 1- Some of the abbreviations are not expanded when they are first introduced in the paper (e.g., GC, LTi cells, among others).
- 2- Figure 5g is not clear; it does not indicate that the data is from different patients.
- 3- The significance of IGH3 being significantly higher in NPC was not mentioned in the paper (see Supplementary Figure 4A).
- 4- In the text, Figure 3D was mentioned earlier than Figures 3a-b.
- 5- The explanations for some figures are not sufficient and clear (e.g., Supplementary Figures 8b-d and Figure 1b).
- 6- CD4 is also expressed on other non-T cell populations, so CD3 staining should be included in Figure 1a.
- 7- There is no discussion explaining why the results in Figure 1e are not consistent between Visium and Stereo-seq for plasma.
- 8- The scale bar in Figure 1d should display actual values instead of "high" and "low."
- 9- There is no discussion explaining why the apoptosis signature is higher in EBVhigh-TCAwoP than in EBVlow-wp (Figures 3a-c).
- 10- The meaning of CR, PR, PD, and SD in Supplementary Figure 2 should be explained.
- 11- It is not clear if the statistical analysis was performed for comparison between all groups or not (Supplementary Figure 4a).

Reviewer #3, expertise in NPC omics, TME (Remarks to the Author):

In this manuscript, the authors identified several new interesting components in the tumor microenvironment and proposed the role of TLS in immune surveillance in NPC. The study also suggested the action of GC reaction in triggering plasma cell differentiation, and their impacts in producing antibodies targeting "EBV-high" NPC cells. Finally, they concluded the TLS signatures predict the prognosis and immunotherapy response of NPC. In addition to re-analysis of multiple cohorts of single-cell sequencing data, the group has reported the first set of spatial transcriptome findings in this unique EBV-associated cancer. While the five single-cell sequencing studies on NPC have previously been reported and multiple interesting features of NPC tumor microenvironment were identified (Liu et al., Jin et. al., Chen et al., Zhao et al, Gong et al, Supplementary Table 1), the authors of this study focused on the characterizing the TLS and their potential interactions with TCA. The findings further

enhance our understanding of the tumor microenvironment of this EBV-associated cancer. Despite of the interesting findings, several key points are needed to clarify to support their conclusion. Most importantly, the “EBV-high/-low” tumor cell classification is needed to be clearly defined.

Major comments:

- A high variety of the proportion of malignancy cells (~0% to >80%) in the NPC biopsies is a major concern for the interpretation of association of GC-B cells proportion with NPC status (early vs advanced, Figure 2b) in the single cells sequencing datasets, bulk and microarray cohorts. The malignancy cell proportion for each sample in these cohorts should be determined to correct the sampling bias during taking biopsies from NPC patients with different disease status. The B cell, T cell and malignant cell proportions in these tumor samples should be shown. The samples containing no malignant cells should be excluded from the study.

- Similar problem should also be considered for determining the prognostic values of TLS components and signatures in different NPC cohorts (Figure 7). The findings may be misleading since the high changes for successfully collection of more malignancy cells (less lymphoid cells) from the patients with advanced disease (with large and obvious tumors) during endoscopic examination. The prognostic roles of TLS components in NPC should be validated in a well-selected NPC cohort by multiple IHC analysis, ISH and histological investigation.

- The clinical and pathological parameter for the NPC samples included in the Stereo-seq and Visium spatial transcription studies have not been provided. As shown in Figure S1D, ST1 shows absence of TLC, what is the disease status of ST1, ST2 and ST3. It is also noted that there is no TLC in ST17, ST9, ST18, ST10, ST11 sections.

- The details of the methods for defining EBV-high and EBV-low TCA in two ST cohorts have not been described in either the results and methodology sections. In Figure 3a and 3b, the corresponding HE image of each sample should be shown. The sFigure 6 shows a poor-quality data and problems for determining EBV gene expression in their ST cohorts. While the abundant expression of EBERs is well known in EBV-associated cancers, either NPC and

EBV+GaC, the EBERs signals shown in the ST2, ST3, M39 and M55 are much lower or similar to that of RPMS1/A73, and BARF0, BALF3/4. In s Figure 6F, the detection of LF3, LF2, BALF3, BARF0 in the EBV-high TCA may indicate the activated EBV-lytic cells in these clusters or problems occurred in annotation of EBV genes in the spatial transcriptome studies.

- In sFigure 6F, multiplex IHC staining should also include the staining for B-cells (CD20) and malignant cells (pan-CK). The EBER signals for these regions can be detected in other sections of the samples.

- In Figure 3d, a high-power view should be shown to demonstrate the EBER signal in single cell level. HE images can help to identify the different cellular features of NPC cells. As shown in the figure, the difference between the two TCA may only be due to cell density, but not EBV signals while the heterogenous morphonology of NPC cells (e.g. spindle cells or cells with round-to-oval vesicular nuclei) is commonly observed in the tissue sections.

- The ST cohort for determination of immunotherapy response in NPC is small (n=12). The authors' conclusion is not convincing and the hypothesis is needed to confirm in a large cohort of NPC patients with ICB therapy.

- In Gong et al study (2021), the CXCL13 high T cells are correlated to better PFS in NPC patient and suggested to be expression in the exhausted T cells, particularly PD-1+ T cells. The findings should be validated in current study and discussed in the results and discussion sections.

Miner comments:

- Figure S1C, the gene names listed in x-axis are unable to read! The small characters in Figure S1D and S2 are too small to read. The H&E parts do not show the histological features of the sample clearly. Zoom-in figures should be included to illustrate the representative features of the malignant, B, and T cell regions.

Reviewer #4, expertise in head and neck cancer, scRNAseq, TME (Remarks to the Author):

The authors performed spatial transcriptomics on 15 nasopharyngeal cancer samples, analyzing spatially-resolved transcriptomes from >30,000 spots. They use this analysis to supplement their re-analysis of >300,000 cells from 77 previously-sequenced NPC samples. They identify immune and stromal cell subpopulations associated with immune checkpoint blockade response and prognosis in nasopharyngeal cancers. The paper is well written and presents interesting analysis of multi-omic data.

Major comments:

1. The authors should clearly define the term tumor cell aggregate and provide clear annotated histologic images highlighting tertiary lymphoid structures and tumor cell aggregates.
2. For scRNA-seq results, the authors describe numbers of B cells, CD4 T cells, CD8 T cells, and CAFs. Were cancer cells analyzed, and if so, how many? Beyond EPCAM expression, were any other techniques (e.g., inferred copy number variation) used to confirm the malignant identify of cells classified as malignant?
3. Throughout the results section, the authors make several statements that are not adequately described from a methodological standpoint or supported by statistical analyses. For example, the statement “co-localization of plasma cells with NK cells and macrophages in NPC tumors” (lines 219-220) should be supported by a description of the methodology and statistical analysis used to draw these conclusions even if these details are provided on the supplemental materials online.
4. In the spatial transcriptomic analyses, how do the authors define ICA regions? How do ICA regions differ from TLS?
5. Overall, there is a large amount of data presented, and it is not always clear how these data fit into the context of prior work. For example, the authors make the claim to identify novel clusters of TLS-associated CAFs (line 274) and TLS-associated CD8 T cells (line 320). The authors should clarify specifically what markers make these clusters novel relative to previously described clusters of the given cell type.

Minor comments:

1. Please define GC reaction (line 55) in the abstract and in the main text (line 83).

2. There are minor grammatical errors throughout the manuscript that need to be corrected prior to publication.

Reviewer #5, expertise in head and neck cancer, scRNAseq, TME (Remarks to the Author):

Liu et al. perform a comprehensive and extensive investigation of the NPC tumor microenvironment using a very broad set of wet lab and big data analytic techniques. The conclusions are highly interesting and, in most cases, compelling. In certain aspects, the analysis and comparison with published data remains superficial. The data transparency needs to be improved. Overall, the manuscripts provides a valuable addition to the current literature and has implications beyond NPC.

Major Revisions

Usage of published data and online resources

While it is perfectly appropriate to re-analyze already published data sets and cohorts, this needs to be clearer from the text and methods (see line 412: “multiple independent sample cohorts”). Given the highly diverse set of methods it would be good if the authors would provide a supplementary table either for each patient or at least each patient cohort that was (re-)used. This should state the patient characteristics, the methods/techniques that were used on that patient sample, for which figures the data was used in the present paper and if/where the data was previously published (analogous to sFig1). Thus far, this is only performed of the single cell data in Suppl. Table 1.

Relevance of differences in GC_B

The authors go into great detail exploring the relevance of differences between early and advanced cancer GC_B cell proportion. However, looking at Fig. 2b and sFig 4b, the actual quantitative difference seems to be minor. Please include the proportions and expand on why this difference is relevant.

Annotation of cell types

The annotation of single cell clusters by single makers should be avoided (e.g.,

CD8_C10_STMN1). Instead, one should try allocating each cluster or group of clusters into a known/published subtype or substate. If there is a “new” cell state or subtype, as might be in the present study for the CXCL13+ CD8 T cells, this needs to be compared with the existing scRNAseq annotation literature. Of note, there is data from non-NPC HNSCC where the expression of GZMK and CXCL13 in T cell substates has been described (Kürten et al. Nat. Comm., 2021). A broad review of single cell T cell states can be found in Andreatta et al. Nat. Communications 2021. Surely, within the T cell analysis there needs to be a distinction between early and late exhaustion (see Philip and Schietinger, Nat. Rev. Immun., 2021).

Underlying sample size (n = ?), proportion difference and p-value

Each Figure and Graph needs to state in the Figure legend how many patients, biological repeats or number of cells are depicted or used for the respective Figure. This is rarely performed throughout the manuscript, for example Fig 2a how many cells? 2b,c,d how many patients? Fig 2e how many slides? Etc., the same is true for the p-value, which is not reported consistently, see Fig. 2b, c, d. Also, for differences in proportions or means between groups the mean or median value should be stated (e.g., when describing sFig 5c or 2g)

Depth and breadth of discussion

Given the high amount of data (18 Figures) the discussion falls a bit short of putting all these aspects into context with the known literature. Especially the role of B cells/ Tfh/ TLS in HNSCC needs to be expanded on (Bruno Nature 2020, Ruffin Nat. Comm. 2021, Cillo Immunity 2020) as well as prognostic and therapeutic implications.

Minor revisions

Use of different spatial transcriptomics platforms

The authors should elaborate on why two different ST platform were used.

Further Implications

The authors should expand on the implications of their findings for other non-NPC but virus associated cancers (line 455) within and outside the head and neck region (cervix, liver,

Merkel cell etc.).

Pathological response data

Is there pathologic response data for the PD-1 treated patients (line 541). If so, that should be used rather than RECIST criteria for evaluating response to immunotherapy. Radiographic imaging is unreliable when it comes to evaluating immunotherapy response (see Forde et al. NEJM 2022).

Fig 2A

Move annotation next to figure

Fig 2f

Explain "NES" in Fig. legend

Fig 6a

Combine Fig. 6a with s9a to give comprehensive overview of marker expression.

sFig4

Explain why number of patients in the early cohort sFig 4 is so much lower than advanced stages.

sFig5 should read "plasma cells" not "plasma"

The grammar should be checked, especially plural forms such as "Tertiary lymphoid structureS (TLS) represent" or "A Tertiary lymphoid structure (TLS) represents" line 75 and "Germinal centreS (GC) are a" or "A Germinal centre (GC) is a" line 88

Line 54

Tfh is not described in study (line 54) beyond Fig 1b upper right panel. Maybe expand?

Line 60

The tone of the abstract needs to be more cautionary, e.g., line 60: this is not shown

specifically in the paper. Rather there are associations between the cell types that “could” mean this “may” be happening.

Naming of clusters

The authors should consider naming the cell clusters without using “_” for B cells, T cells, CAF etc.: “naïve B cells” or “CXCL13+ CAF” is easier to read than “Naïve_B” and “iCAF_C2_CXCL13 CAF”.

Line 266

Should “degree” read “ability”?

Reviewer #6, expertise in TLS structures (Remarks to the Author):

Through the analysis of single-cell and spatial-resolved transcriptomes, Liu and colleagues explored the composition, function, and clinical relevance of tertiary lymphoid structures (TLSs) in a large cohort of Nasopharyngeal Carcinoma (NPC) patients. The authors claim to have identified novel cell populations within tumor-associated TLSs, including GC B cells that support antibody production. Additionally, the authors claim that antibody-secreting plasma cells promote apoptosis of EBV+ tumor cells, probably by antibody-dependent cellular cytotoxicity or antibody-dependent (ADCC) or antibody-dependent cellular phagocytosis (ADCP). Also, CXCL13-expressing fibroblasts promote B cell adhesion and antibody production, whereas TLS-associated CXCL13-producing CD8 T cells differentiate towards TCA-associated exhausted CD8 T cells with decreased stemness and TCR diversity. Finally, all this TLS apparatus was associated with better prognosis and immunotherapy response in NPC patients. Although the manuscript has positive points, most of the conclusions claimed by the authors are not supported by the data presented, due to the highly descriptive nature and circular analyses of the manuscript. Therefore, many conclusions need to be toned down or further supported by in vitro experiments.

1. Germinal Centers are a very dynamic and specialised structure. Circular trajectories, as shown elsewhere (PMID 37230755), are definitely more appropriate to represent GC reactions in TLSs. In the introduction, authors claim that little is known about GC reactions within TLSs, and this reference must be cited.

2. If GC reaction within TLSs is critical to generate high affinity antibodies against EBV, virus antigens are the trigger that generate TLS formation in NPC patients. This should be addressed.
3. GC B cells are quite abundant in normal tissues (Fig. 2b and Suppl. Fig. 4b) – it is probably more abundant in normal tissues than in the other sites. Why?
4. If plasma cells derive from GC B cells in a linear direction, it seems contradictory to have high abundance of GC B cells and low abundance of plasma cells in early stages, and low abundance of GC B cells and high abundance of plasma cells in late stages of NPC. What is the physiological advantage of having high abundance of GC B cells in early stages of NPC if these cells are not being converted in highly effective plasma cells?
5. If the abundance of antibody secreting plasma cells is similar across NPC samples at different stages (Fig. 2b), this should not be associated with tumor progression.
6. The authors seem confused about the differences between GC B cells and plasma cells.
7. Hard to understand whether the apoptotic signature is coming from tumor cells. Single cell transcriptomics data should be used to address this and better correlate with plasma cells.
8. The authors claim that novel cell populations were identified within TLSs. All populations shown here have been shown elsewhere (several papers).
9. Apart from correlative data, no data prove that i) antibody-secreting plasma cells promote apoptosis of EBV+ tumor cells, ii) CXCL13-expressing fibroblasts promote B cell adhesion and antibody production, and iii) TLS-associated CXCL13-producing CD8 T cells differentiate towards TCA-associated exhausted CD8 T cells with decreased stemness and TCR diversity. All conclusions need to be toned down or supported by extra experiments.
10. Please specify what the immunosuppressive signature means (Fig. 4d). This result is out of context.
11. Fig. 8 needs to be removed, as it is based on lots of assumptions. The data from Liu and colleagues do not support the model proposed.

Minor comments:

- Title seems inaccurate.
- Antiviral responses (against EBV) are not introduced throughout the manuscript.
- Why did the authors use two protocols of spatial transcriptomics?

- For the ST analyses, anything that was not TLS or TCA was defined as stromal regions. This reviewer was wondering if immune cells sparsely distributed across the TME could interfere in such analyses, as they were included as stromal components.
- Other TLS examples of multiplex IHC should be added as a Suppl. Figure.
- What is the biological tissue sample used for stereo-seq?
- According to the tissue of origin, please show the UMAP plots (Suppl. Fig. 1b) distributed into 3 groups only: blood samples, tumor tissue samples, and non-tumor tissue samples.
- It will be great to see the distribution of non-TLS associated cell populations (such as CXCL13- T cells, CXCL13- CAFs etc.) as negative controls in Fig. 1e and Suppl. Fig. 3e.
- Correlations between NK and macrophage abundances with plasma cell abundances are weak.
- Correlations between CXCL13+ fibroblast abundance with plasma cell abundance is also weak.
- Expression of FAP-alpha by CAFs is weak for CAF-1 and CAF-2 (Suppl. Fig. 8b)
- Sentences that are not clear and need to be rephrased: (line 266: ICB therapy response might be mediated through the degree of ICB to reverse the immunosuppression of EBVhigh malignant cells, line 272: no canonical marker (CR2) expression was detected in the fibroblasts in NPC samples).
- Suppl. Fig 1a has a typo (Please replace immunotheray by immunotherapy).

Overall, this reviewer found that the manuscript is very descriptive and fails to propose a mechanism that is not supported by the data shown by Liu and colleagues.

Reviewer #1, expertise in head and neck cancer, immunotherapy (Remarks to the Author):

This manuscript by Liu et al describes the single-cell and spatial transcriptomic analyses of tertiary lymphoid structures (TLS) in nasopharyngeal carcinoma (NPC), and associating these characteristics with clinical outcome, specifically with respect to immune checkpoint blockage.

General comments:

1. The clinicopathological characteristics of NPC patients who were evaluated in this study should be provided and summarized, not only their stage but also their prior treatment history. Based on Supplementary Table 1, there appears to be a mix of patients with early stage and late stage locally advanced disease, and also those treated with immune checkpoint blockade. Presumably the former group was treated with curative (chemo)radiotherapy, and the latter with systemic therapy. It is difficult to associate the findings in each analysis without clear articulation of the clinical status of the NPC patients being examined.

Response: Thanks for your thoughtful comments. In this revision, we have provided the clinicopathological characteristics, stage, and prior treatment history for our patient cohort. For the previously published data, we have also integrated and summarized the clinical information (provided in the respective papers) into previous **Supplementary Table 1**. Patient information from the spatial transcriptome cohort who underwent immunotherapy was initially provided in previous **Supplementary Table 3**. For the convenience of reference, in this revised version, we have consolidated all patient-related information into new **Supplementary Table 1**. Moreover, we summarised the clinical status (gender, age, and stage) of the NPC patients for our significant findings on TLS, and we found no significant differences between the TLS-CS-high and TLS-CS-low groups in three independent cohorts (Bulk RNA-seq, Microarray, and Microdissection cohorts; **Response Figures 1a-c**). Please kindly review in Results section in Page 17, Lines 18-19 and **Supplementary Figures 15c-e**.

Response Figure 1: Bar plots showing the sample proportions for different age (left), gender (middle), and clinical stage (right) categories between the TLS-CS-high and TLS-CS-low groups for the Bulk-RNA-seq, Microarray, and Microdissection cohorts. Chi-square test was performed to evaluate the significant difference. NS: not significant (P value > 0.05).

2. Figure 7 a-d suggest that TLS cell signature with the 7 cell clusters appear to be prognostic and predictive for immune checkpoint blockade. However, these are based on gene expression bulk RNAseq data and thus they are not able to be certain that these cell clusters are from TLS and not from other structures. How do the applicants confirm that these are from TLS?

Response: Thanks for your comments. Using an approach integrating single-cell and spatial transcriptomics, we were able to specify TLS subclusters and their spatial locations, following a similar strategy applied in other studies (Qi, J et al, Nat Commun, 2022; Cable, D.M et al, Nat Biotechnol, 2022). To further evaluate the clinical relevance of our findings in larger sample collections, we utilized Bulk-RNA-seq, Microarray, and Microdissection cohorts' data to explore the relationship between seven TLS-related cell clusters and patient survival. We deconvolved the cell composition of each sample for above cohorts using single-sample gene set enrichment analysis (ssGSEA), which estimates the relative proportion of cell clusters based on gene expression profile. Using the Pearson correlation test, ssGSEA scores were subsequently correlated within cell clusters. This method was widely used for estimating cell compositions and testing their clinical relevance in samples with bulk RNA-seq data (Chen, Y.P et al, Cell Res, 2020). Importantly, the specificity of cell annotation and spatial location in NPC TLS for these TLS-related cell clusters has been independently validated using multiple immunohistochemistry (IHC) assays (main **Figure 1a, 5d, and 7b**). Additionally, previous study has reported the presence of B cells and Tfh (T follicular helper cells) within tumour-associated TLS (Li, J.P et al, J Immunother Cancer, 2021). Together, these findings assure the presence of seven TLS-related cell clusters within the TLS of NPC.

Specific comments:

1. Line 94: LTi should be spelled out

Response: Thanks. As suggested, we fully spell out lymphoid tissue inducer cells at the first appearance in this Revision. Please kindly review this in Introduction section in Page 4, Line 26 in this revision.

2. Lines 104-105: Please clarify in which cell populations BCL-2 (the anti-apoptotic molecule) is upregulated to inhibit the growth of melanoma. This sentence is unclear.

Response: Thanks. Please note that the previous study reported BCL-2 upregulation in T cells in or close proximity to TLSs in melanoma based on bulk RNA-seq data and did not specify exact T cell subpopulation responsible for the upregulation (Cabrita, R et al, Nature, 2020). We have clarified this point in this revision. Please kindly review in Introduction section in Page 5, Lines 7-8 in this revision.

3. Figure 2e: the expression of IgA and IgG in advanced NPC tumors shown in this figure seems non-specific, what EBV-related viral antigens are these antibodies directed against? The same question is for Figure 2d that demonstrates pan-immunoglobulin transcription levels being lower in advanced stages than at the early stages, how can the authors prove specificity of these antibodies against NPC antigens?

Response: Thanks for the thoughtful comments. To address the comments, we provided high-resolution, magnified staining results for IgA and IgG, demonstrating specific staining on plasma cells (**Response Figure 2a**). However, we could not infer the specific EBV-

related antigens targeted by these antibodies. The EBV genome is relatively large, approximately 172 kilobase pairs in size, encoding approximately 80 to 86 proteins with more than 3,000 different variants (Santpere, G et al., Genome Biol Evol, 2014; Vita et al, Nucleic Acids Res 2015). It is challenging to identify specific antigens and corresponding antibodies at present. Instead, through multiplex IHC staining assay, we found enhanced activity of antibody-dependent cellular cytotoxicity (ADCC) or antibody-dependent cellular phagocytosis (ADCP) in the EBV^{high}-TCA, along with increased apoptosis of tumour cells (**Response Figure 2b**). These findings suggest the presence of antibodies recognizing viral and/or tumoral antigens due to EBV infection and tumorigenesis in the EBV^{high}-TCA of NPC. Please kindly review in the Results section in Page 9, Lines 3-5 and Page 10, Lines 27-29 and main **Figure 2e** and **3c**.

Response Figure 2: EBERs staining and multiplex IHC staining of molecules related to ADCC and ADCP of NPC tissue biopsy. a. Representative images of IgA and IgG staining in NPC tumour samples. Images are representative of three biological replicates. Scale bars are 50µm. b. Representative image of EBERs staining and multiplex IHC staining of molecules related to ADCC and ADCP at different regions of NPC tissue biopsy. Cells are coloured according to their staining with EBERs (brown), IgG (green), CD20 PanCK (cyan), CD56 (yellow), CD68 (purple), and CASP3 (red) proteins as indicated on top. Images are representative of three biological replicates. Scale bars are 30µm.

4. Figure 4g: please provide clinicopathological characteristics of these two independent cohorts (Bulk-RNA-seq and Microarray cohorts). How comparable are these cohorts to the NPC patient samples used in the current study?

Response: Thanks. As suggested, we have included the clinicopathological characteristics of these two independent cohorts in **Supplementary Table 1** in this revision. These samples share the similar pathological diagnosis, with NPC originated from South China. Additionally, we compared the differences in gender, age, and stage between these two independent cohorts (Bulk-RNA-seq and Microarray cohorts) and NPC patient samples used in the current study, and observed no significant difference (**Response Figure 3**). Please kindly review in Results section in Page 17, Lines 6-7 and **Supplementary Figure 15b**.

Response Figure 3: Bar plots showing the proportion of different age (left), gender (middle), and stage (right) groups in Bulk RNA-seq, Microarray, and Single-cell RNA-seq cohorts. Chi-square test was performed to evaluate the significant difference. NS: not significant (P value > 0.05).

5. What criteria were used to select specific cases for spatial transcriptomics? The details of the “microdissection cohort, $n = 189$ ” in Supplementary Figure 3 should be provided.

Response: Thanks. For spatial transcriptomics (ST) analysis, we collected fresh-frozen specimens of three NPC and two EBV⁺ GaC patients and formalin-fixed paraffin-embedded (FFPE) specimens of 12 advanced NPC patients from Sun Yat-sen University Cancer Center (SYSUCC), Guangzhou, China (**Supplementary Table 1**). Among these, the 12 advanced NPC patients each received at least two courses of PD-1 blockade (toripalimab) treatment, facilitating us to investigate the key spatial characteristic associated with immune response. The immunotherapy responses for the 12 patients should be fully accessed. To validate our findings, we performed high-resolution spatial transcriptomic (ST) assays for the three fresh-frozen and treatment-naive NPC samples using Stereo-seq technology (BGI-Shenzhen, China). We also collected the two treatment-naive EBV⁺ GaC patients for extending our findings in EBV positive solid tumour. In this revision, we have provided detailed information regarding the inclusion criteria. Please kindly review in **Supplementary Table 1**.

For NPC samples from the microdissection cohort by Tay et al. (Sci Adv, 2022), we had included the basal clinical information in the Methods section in the previous version. In this revision, we have updated and provided a more detailed clinical information for each enrolled patient in **Supplementary Table 1**.

6. Supplemental Table 3-1: this table describing the 12 NPC patients treated with toripalimab and chemotherapy seems to be missing.

Response: Thanks. We have added the treatment information for each patient in the **Supplementary Table 1**, which is reproduced as below (**Response Table 1**).

Supplementary Table 1-5 Spatial transcriptomics summary of NPC and GaC									
Sample	TNM stage	Number of Spots Under Tissue	Median Genes per Spot	Mean Reads per Spot	ICB	Platform	Therapy	Chemotherapy	
ST1	II	6,053	4,643	NA	NA	Stereo-seq	treatment-naive	treatment-naive	
ST2	IVa	1,722	3,058	NA	NA	Stereo-seq	treatment-naive	treatment-naive	
ST3	I	1,788	2,226	NA	NA	Stereo-seq	treatment-naive	treatment-naive	
ST5	IVa	2,021	5,100	114,480	CR	Visium	toripalimab plus chemotherapy	Capecitabine	
ST6	IVb	2,614	1,859	59,441	CR	Visium	toripalimab plus chemotherapy	Paclitaxel + Cisplatin + 5-Fluorouracil	
ST7	IVa	1,939	5,826	91,934	PD	Visium	toripalimab plus chemotherapy	Gemcitabine + Anlotinib	
ST8	IVb	1,209	2,854	151,387	CR	Visium	toripalimab plus chemotherapy	Paclitaxel + Cisplatin + 5-Fluorouracil	
ST9	III	2,802	1,118	74,173	PD	Visium	toripalimab plus chemotherapy	Tegafur, Gimeracil and Oteracil Potassium	
ST10	IVa	1,309	2,625	91,293	PR	Visium	toripalimab plus chemotherapy	Gemcitabine + Cisplatin	
ST11	IVb	1,272	2,500	90,144	CR	Visium	toripalimab plus chemotherapy	Gemcitabine + Cisplatin	
ST12	IVa	3,669	6,947	76,081	CR	Visium	toripalimab plus chemotherapy	Paclitaxel + Tegafur, Gimeracil and Oteracil Potassium	
ST16	IVb	1,556	8,502	104,781	PD	Visium	toripalimab plus chemotherapy	Paclitaxel + Carboplatin	
ST17	IVb	2,067	2,171	82,985	SD	Visium	toripalimab plus chemotherapy	Tegafur, Gimeracil and Oteracil Potassium	
ST18	III	1,977	1,096	87,060	PR	Visium	toripalimab plus chemotherapy	Docetaxel	
ST19	IVb	1086	728	128997	PD	Visium	toripalimab plus chemotherapy	Gemcitabine	
M39	IVb	3280	1812	156942	NA	Visium	treatment-naive	treatment-naive	
M55	III	4,605	2,637	71,276	NA	Visium	treatment-naive	treatment-naive	

*Pathological diagnosis and TNM stage of NPC were determined according to the 8th edition of the International Union against Cancer (UICC) and American Joint Committee on Cancer (AJCC) staging system.

Response Table 1: Spatial transcriptomics summary and treatment information of NPC and EBV positive GaC samples. GaC, Gastric adenocarcinoma; NA, Not available; CR, Complete response; PD, Progressive disease; PR, Partial response; SD, Stable disease.

7. Figure 4d: please describe how the immunosuppressive scores were determined.

Response: Thanks. As suggested, we provided a description for the definition of immunosuppressive scores in main **Figure 4d**, as read "Immunosuppressive score was determined according to the expression levels of immunosuppressive genes including CD47, PVR, CD276, LGALS9, ADORA2B, ADAM10, HLA-G, CD274, FASLG, TGFB1, and IL10." Please kindly review in the Methods section in Page 30, Lines 11-15.

Reviewer #2, expertise in spatial transcriptomics, TME (Remarks to the Author):

Herein, the authors conducted integrative scRNA-seq and spatial transcriptomics analyses on various cohorts of nasopharyngeal carcinoma to elucidate the single-cell spatial organization of Tertiary Lymphoid Structures (TLS) and their relationship with tumor cell aggregates. The strength of this study lies in its utilization of different patient cohorts, including those who received immunotherapy, as well as its combined use of scRNA-seq and spatial transcriptomics, leading to the identification of cell subsets potentially influencing anti-tumor responses and immunotherapy outcomes. While I acknowledge these strengths, I would like to highlight both major and minor concerns regarding the study.

Response: Thanks for the favourable comments. Following the Reviewer's suggestions, we have provided additional experimental assays and data analyses to improve our manuscript. We believe that we have addressed the concerns raised by the Reviewer in this revised manuscript. Please kindly review our responses below.

Major:

1- The novelty of some of the findings is not clearly established. For example, the role of CXCL13 produced by Cancer-Associated Fibroblasts (CAFs) in B cell recruitment and TLS formation has been previously reported (Rodriguez et al, Cell Rep). Similarly, the identification of CXCL13+CD8+ T cells within TLS and the Tumor Microenvironment (TME) has been observed in other cancer types (Workel et al, Cancer Immunol Res. 2019, Dai et al, J Immunother Cancer. 2021).

Response: Thanks for the comments. Indeed, we identified CXCL13+ CAFs as a novel cell cluster based on their unique gene signatures (**Response Figure 4a**), which are distinct from the CXCL13+ CAFs identified by Rodriguez et al and follicular dendritic cells (FDC) in lymphoid tissue. In our study, CXCL13+ CAFs exhibited sparse CR2 expression (**Response Figure 4b**), a marker associated with FDCs, indicating that they were not well-known FDC cells. This cell cluster also demonstrated high expression of CXCL10 and CCL19, which are associated with B cell chemotaxis but were not found in the study by Rodriguez et al. Moreover, we have supplemented additional experiments to show CXCL13+ CAFs' functions in promoting B cell antibody production via TNFSF13B, which is the first report in human tumours. Although both cell clusters share common high expression of some markers (CXCL13, VCAM1, and ICAM1; **Response Figure 4a**) and play a role in B cell recruitment and TLS formation, their distinct gene signatures suggest divergence in precise functions. Please kindly review in Results section in Page 12, Lines 9-17 and main **Figure 5a** and **Supplementary Figure 11a**.

Regarding CXCL13+CD8+ T cells, we identified their high expression of activated (TNFRSF18) maker and low expression of exhausted (HAVCR2) and effector (GZMB) makers (**Response Figure 4c**). By contrast, Workel et al and Dai et al found CXCL13+CD8+ T cells highly expressing effector (GZMB), exhausted (HAVCR2), and activated (TNFRSF18) markers in tumours. These markers indicate distinct signatures for CXCL13+CD8+ T cells between our present study and the other two studies. Moreover, we

uncovered novel functions and states of CXCL13⁺CD8⁺ T cells through multi-omics analyses. For instance, TCR transition and trajectory analyses suggest that CXCL13⁺CD8⁺ T cells may undergo a pre-exhausted state in NPC. Additionally, spatial transcriptomics analysis and multiplex IHC assay solidified the specific localization of CXCL13⁺CD8⁺ T cells within the TLS of NPC. Although Gong et al also reported a subset of CXCL13⁺CD8⁺ T cells in NPC (Gong, L et al, Nat Commun, 2021), we note that our characterization of the functional state of CXCL13⁺CD8⁺ T cells, including stemness, activation, and their location within TLS, further deepens our understanding of their role in anti-tumour immunity. Please kindly review in Results section in Page 14, Lines 18-24 and main **Figure 7a**.

Response Figure 4: The expression levels of cluster markers for CAFs and CD8⁺ T cells in NPC. a. Heatmap showing the normalized mean expression of signature genes (rows) for each CAF cluster (columns). b. Violin plot showing the expression levels of CR2 in different CAF clusters. c. Heatmap showing the normalized mean expression of signature genes (rows) for each CD8⁺ T cell cluster (columns). For heatmap, filled colours from blue to red represent scaled expression levels from low to high.

2- Although the integrative transcriptomics analysis has revealed intriguing findings, the study appears to lack essential validation and functional studies. For instance:

I-The study presented here demonstrates that plasma cells play a role in promoting the apoptosis of malignant cells in EBV-related tumors, likely through mechanisms such as ADCC and ADCP. These noteworthy findings have primarily emerged from multi-omics analysis at the transcriptomics level and certain correlation analyses. However, Figure 3 appears to lack in vitro and in vivo validation to confirm the killing effects of antibodies produced by B cells on EBV-positive TCA. Additionally, it remains unclear whether these killing effects are mediated by both NK cells and macrophages.

Response: We appreciate the reviewers' insightful comments. Our study identifies the significant presence of B and plasma cells in NPC patients, indicative of a favourable prognosis and their potential antitumor role. Spatial transcriptomic analysis demonstrates that CASP3⁺ malignant cells in EBV^{high}-TCA are surrounded by a higher density of IgG⁺ plasma cells compared to EBV^{low}-TCA. This suggests a correlation between apoptosis induction in tumour cells and IgG⁺ plasma cells, especially in those tumour cells with EBV infection (**Figure 3a-b**). In this revision, we conducted additional IHC assays for IgG, CD56, CD68, PanCK, and CASP3 on NPC tissue biopsy section. These assays revealed a close spatial proximity between apoptotic tumour cells (PanCK⁺CASP3⁺) and natural NK cells (CD56⁺), macrophages (CD68⁺), and plasma cells (IgG⁺) predominantly in EBV^{high}-TCA, contrasting with EBV^{low}-TCA (above **Response Figure 2**). This observation supports the role of antibody-dependent cellular cytotoxicity (ADCC) or antibody-dependent cellular phagocytosis (ADCP) in mediating tumour cell apoptosis in these regions. Additionally, this mechanism has been reported in renal tumours (Meylan, M et al, Immunity, 2022), suggesting a common mechanism of ADCC or ADCP mediating tumour cell apoptosis across various tumour types with enriched TLS. We have added these results in the revised manuscript. Please kindly review in the Results section in Page 10, Lines 27-29 and main **Figure 3c**.

Moreover, the underlying mechanisms responsible for the higher infiltration of plasma cells and IgG deposition within high EBV TCA have not been elucidated, and addressing these aspects would strengthen the study's conclusions.

Response: By addressing the comment, we delved into the interactions between EBV⁺ tumour cells with various subsets of B and plasma cells (below **Response Figure 5a**). Notably, we identified specific interaction pairs CCL2-CCR2 between plasma cells and EBV⁺ (LMP1) malignant cells in NPC. Given the known functions of this ligand and receptor in recruiting different immune cells to TME (Hao, Q et al, Cell Commun Signal, 2020), this interaction is likely critical for the recruitment and infiltration of plasma cells and thereby IgG production in EBV⁺ TCA. Additionally, our analysis revealed a positive correlation solely between the plasma cell signature and the average expression of CCL2-CCR2 ligand/receptor pairs in a bulk RNA-seq cohort (**Response Figure 5b**), corroborating that EBV⁺ malignant cells may actively recruit plasma cells through the CCL2-CCR2 axis.

However, we acknowledge that further experiment is warranted to validate this assumption. In this revision, we have updated the context and corresponding supplementary figures. Please kindly review in Results section in Page 11, Lines 8-12 and **Supplementary Figure 10c-d**.

Response Figure 5: Cell-cell interactions between EBV⁺ (LMP1) malignant epithelial cells and B cells and correlation analysis between CCL2-CCR2 and plasma proportion in NPC. a. Significant ligand-receptor pairs contributing to the signalling from EBV⁺ malignant cell to four B cell clusters. The dot colour and size represent the calculated communication probability and p-values, respectively. *P* values are derived from one-sided permutation test. Colour from blue to red indicates communication probability from low to high. b. Scatter plot showing the correlation between the expression level of CCL2-CCR2 and the plasma cell signature. Each data point represents an individual sample. Pearson correlation coefficient was used to assess the relationship, and the confidence interval of the correlation is depicted in grey.

II- In this study, CXCL13⁺ TLS-associated CAFs were observed to promote B cell adhesion and antibody production in NPC through multi-omics integrative analysis at the transcriptomics level, which was further validated with in vitro co-cultures. However, additional validation is required to solidify these findings. The cell-cell interaction analysis identified potential mechanisms for crosstalk between CXCL13⁺ TLS-associated CAFs and B cells, involving CXCL13-CXCR5, VCAM1-integrin, ICAM1-integrin, and TNFSF13B-related ligand-receptor pairs. However, in vitro blocking experiments are lacking to confirm the direct involvement of these mechanisms. The co-culture experiments (cell adhesion assay and co-culture assay) were performed on VCAM1⁺ versus VCAM1⁻ CAFs, but they did not confirm the contribution of CXCL13-CXCR5 produced by CAFs in supporting B cell adhesion and antibody production.

Response: We are grateful to the thoughtful suggestions. In this revision, we have conducted a series of coculture and blockage experiments to elucidate the role of CXCL13⁺ TLS-associated CAFs in recruiting B cells and facilitating antibody production. Initially, we conducted co-culture experiments using VCAM1⁺ or VCAM1⁻ CAFs and B cells from NPC tissues and or healthy peripheral blood samples. CCK8 assays demonstrated a significantly stronger binding of VCAM1⁺ CAFs with B cells than VCAM1⁻ CAFs. Furthermore, addition of VCAM1 neutralizing antibody notably reduced B cells adhesion to VCAM1⁺ CAFs (below **Response Figure 6a**), underscoring the critical role of VCAM1 in B cell adhesion.

Subsequently, to determine whether CXCL13⁺ TLS-associated CAFs enhance B cell antibody production via TNFSF13B signalling, we carried out similar co-culture experiments with additional TNFSF13B overexpression or blockade in fibroblasts. The co-culture assays revealed significant increase in IgG production of B cells co-cultured with CAFs overexpressing TNFSF13B compared to those with control fibroblasts (**Response Figure 6b**). By contrast, the increased IgG production of B cells due to TNFSF13B overexpression was significantly abolished with addition of a TNFSF13B neutralizing antibody to the co-culture setting. These findings corroborate that CXCL13⁺ TLS-associated CAFs promote antibody production through TNFSF13B signalling.

To further verify the role of CXCL13 in CXCL13⁺ TLS-associated CAFs to recruit B cells, we also conducted co-culture assays for fibroblasts and B cells, with or without CXCL13 overexpression or blockade. These assays revealed that CXCL13 overexpression significantly enhanced B cell migration, which was impeded by the addition of CXCL13 neutralizing antibody (**Response Figure 6c**). This observation supports the critical role of CXCL13 in B cell recruitment for TLS-associated CAFs.

Collectively, these experiments substantiate the critical role of CXCL13⁺ TLS-associated CAFs in the B cell recruitment and antibody production within the NPC TME. In this revision, we have updated the results and corresponding figures. Please kindly review in Results section in Page 13, Lines 14-29 and Page 14, Lines 1-10 and main **Figure 6**.

Response Figure 6: Co-culture assays of CAFs and B cells. a. Bar plots showing the capability of B cell adhesion to CAFs. B cells were cocultured with either VCAM1⁺ CAFs, VCAM1⁻ CAFs, or VCAM1⁺ CAFs in combination with a VCAM1 neutralizing antibody. The

relative number (OD value) of B cells bound to CAFs were quantified using CCK8 assays. b. Bar plot showing the relative expression level of TNFSF13B in fibroblasts (left panel). Fibroblasts were transduced with an empty vector (Con_EV) or TNFSF13B plasmid. Bar plots showing the IgG production of B cells co-cultured with fibroblasts overexpressing either an empty vector (Con_EV), CXCL13 alone (OE_CXCL13), or CXCL13 in combination with a CXCL13-specific neutralizing antibody (right panel). IgG level in supernatant was determined using ELISA. c. Bar plot showing the relative expression level of CXCL13 in fibroblasts (left panel). Fibroblasts were transduced with an empty vector or CXCL13 plasmid. Bar plot illustrates the migration capabilities of B cells under various co-culture conditions with fibroblasts (right panel). The B cells were co-cultured with fibroblasts overexpressing either an empty vector (Con_EV), TNFSF13B alone (OE_TNFSF13B), or TNFSF13B in combination with a TNFSF13B-specific neutralizing antibody. *P* values are derived from two-sided student t-tests. ns *P* > 0.05, **P* < 0.05, ***P* < 0.01, ****P* < 0.001. The data is presented as mean ± SD. OE: overexpression; Neu: neutralizing antibody; Con: control; EV: empty vector.

Furthermore, to provide direct evidence for the importance of CXCL13-CXCR5 in driving TLS formation, B cell recruitment, and antibody production in NPC, it is necessary to conduct in vivo experiments in a mouse model lacking CXCR5. Additionally, performing IHC to show the expression of CXCR5 on B cells in proximity to CXCL13+ CAFs in the TLS would provide valuable insights.

Response: Following the Reviewer's suggestion, we performed IHC staining assays for CXCR5, CXCL13, and PDPN on NPC biopsy samples (**Response Figure 7**). The IHC assays revealed CXCL13+ CAFs in proximity to CXCR5+ B cells within TLS, providing spatial contact supporting the notion that CXCL13+ CAFs recruit B cells through CXCR5 involvement. Moreover, a previous study has demonstrated the essential role of CXCR5 in TLS formation within the TME using a mouse model lacking CXCR5 (Rodriguez, A.B, et al, Cell Rep, 2021). This finding further corroborates the critical role of CXCL13-CXCR5 axis in driving TLS formation in TME. We acknowledge that in vivo experiments utilizing a mouse model deficient in CXCR5 would provide more definitive evidence for the necessity of CXCR5 in TLS formation. However, it is still a challenge to establish a suitable spontaneous tumour mouse model for NPC. In this revision, we have added the IHC staining results and acknowledged limitation. Please kindly review in Results section in Page 12, Lines 27-28 and main **Figure 5d**.

Response Figure 7: Multiplex IHC staining of CXCL13⁺ CAFs and CXCR5⁺ B cells in NPC tissue biopsy. Cells were coloured according to their staining with CXCR5 (green), CD20 (cyan), FAP- α (yellow), and CXCL13 (red) proteins as indicated on top. The green, cyan, yellow, and red arrows indicated positive cells with the expression of CXCR5, CD20, FAP- α , and CXCL13 proteins in NPC tissue, respectively. Images are representative of three biological replicates. Scale bars are 20 μ m or 50 μ m, respectively.

III- Once again, validation regarding the presence of CXCL13⁺CD8⁺ T cells in TLS in NPC is lacking, with only transcriptomics analysis supporting their existence. IHC is essential to confirm the presence of this subset of T cells within the TLS of NPC and to investigate their phenotype, such as the expression of PD-1 and the absence of other exhaustion molecules.

Response: Thanks for the suggestion. In our study, we identified CXCL13⁺CD8⁺ T cells in NPC with distinctive molecular signatures, characterized by elevated expression of TCF7 (a marker of T cell stem-like properties), CXCL13, and BCL6. These cells also exhibited low expression of naïve markers such as SELL and CCR7, and were lack of exhaustion markers including LAG3, CTLA4, ENTPD, and HAVCR2 (TIM-3). These features make them different from naïve and exhausted T cell populations. Following the Reviewers' suggestion, we performed multiplex IHC to simultaneously stain CXCL13, CD8, CD3, PDCD1, and HAVCR2 in NPC biopsy samples (**Response Figure 8**). We observed predominant location of CXCL13⁺CD8⁺ T cells around B cells. Notably, these cells do not express the exhaustion marker HAVCR2. These observations verify the robustness of our findings based on transcriptomics analysis and further elucidate the unique characteristics of CXCL13⁺CD8⁺ T cells in the NPC TME. These results have been added in this revision. Please kindly review in Results section in Page 14, Lines 25-27 and main **Figure 7b**.

Response Figure 8: Multiplex IHC staining of CXCL13⁺CD8⁺ T cells in NPC tissue biopsy. Cells were coloured according to their staining with PD1 (green), HAVCR2 (red), CD20 (cyan), CD8 (yellow), and CXCL13 (orange) proteins as indicated on top. The green, red, cyan, yellow, and orange arrows indicated positive cells with the expression of PD1, HAVCR2, CD20, CD8, and CXCL13 proteins in NPC tissue, respectively. Images are representative of three biological replicates. Scale bars are 20µm.

Moreover, the roles of this specific subset in TLS formation and anti-tumor immunity remain unclear. It is essential to conduct *in vitro* and *in vivo* experiments to confirm whether CD8⁺ CXCL13⁺ T cells indeed have the highest potential for recognizing and attacking malignant NPC cells, rather than solely relying on predictions based on bioinformatic analysis.

Response: We agree with the Reviewer that experimental validation of the findings based on bioinformatic analysis is important to confirm the potential of CXCL13⁺CD8⁺ T cells in recognizing and attacking NPC cells. However, this endeavour faces significant challenges. The small size of NPC biopsy samples allows only collection of very limited number of cells, and CXCL13⁺CD8⁺ T cells constitute a minor fraction of the overall CD8⁺ T cell population (cell proportion ranges from 0% to 9.89%, with a median of 1.42%). These result in difficulty in isolating sufficient CXCL13⁺CD8⁺ T cells for in-depth analysis. Additionally, the absence of an appropriate spontaneous tumour mouse model for NPC further impedes *in vivo* experiments.

Despite these challenges in directly studying the functions of CXCL13⁺CD8⁺ T cells in NPC TME, previous study has demonstrated that CXCL13⁺CD8⁺ T cells are tumour antigen-specific and associated with improved responses to immunotherapy in other tumour types (Liu, B et al, Nature Cancer, 2021). These cells are characterized by high CXCL13 and TCF7 expression, and low or absent expression of exhaustion markers such as ENTPD1, LAYN, HAVCR2, and LAG3. Remarkably, these features closely align with the CXCL13⁺CD8⁺ T cells identified within the NPC TME in our study (**Response Figure 4c**). This underscores a critical role of CXCL13⁺CD8⁺ T cells in the antitumor immunity in

NPC. We have added this comparison in this revision. Please kindly review in Discussion section in Page 21, lines 4-5.

Additionally, the connection between CD8_C8_CXCL13 cells and malignant cells, which is predicted to occur through CXCR3-CCL20, CXCR3-CXCL10, and CXCR3-CCL19 interactions, as well as the co-stimulatory pathway (CD27-CD70), requires thorough validation. Inclusion of experimental evidence will significantly strengthen the findings and shed light on the functional relevance of CXCL13+CD8+ T cells within the TLS and their potential impact on anti-tumor immunity in NPC.

Response: As abovementioned, functional characterization of CXCL13+CD8+ T cells in NPC remains challenging. Cell-cell communication analysis revealed the interactions of CXCL13+CD8+ T cells and malignant cells, via CXCR3-CCL20, CXCR3-CXCL10, CXCR3-CCL19, and CD27-CD70 interactions. By contrast, these ligand-receptor interaction pairs are weaker between HAVCR2+CD8+ T cells and malignant cells. Moreover, both CXCR3-receptors and CD27-CD70 pairs have been involved in the activation of CD8+ T cells (Ferreira, C. P et al, PLoS Negl Trop Dis, 2020; Song, D. G et al, Blood, 2012), highlighting the potential of these axes in the chemotaxis and activation of CXCL13+CD8+ T cells within the context of NPC TME.

3- The manuscript lacks a solid transition between some of the data, creating a disconnect between key findings. For instance, the study has identified two critical populations, namely CXCL13+ CAFs and CD8_C8_CXCL13, within the TLS of NPC. However, there is no clear logical connection between these populations and their collaborative roles in TLS formation, B cell recruitment, and the regulation of anti-tumor immunity. Additionally, the relation between these subsets and the killing effects of plasma cells in EBV^{high} TCAs is not well-established. As a result, the current format of the paper presents these findings as isolated islands in an ocean, rather than creating a cohesive and interconnected narrative.

Response: Thanks for the insightful comments. In our study, we would like to emphasize that all the TLS-related clusters, including CXCL13+ CAFs, CXCL13+CD8+ T cells, Tfh, and B cells, exhibit intrinsic colocalizations and interactions in NPC TLS. In this revised manuscript, we have restructured the content to highlight the central role of B cell as a pivotal hub connecting various components, including CXCL13+ CAFs, CXCL13+CD8+ T cells, and EBV^{high}-TCA. Firstly, leveraging single-cell RNA sequencing, we unravelled the heterogeneity of B cell clusters and delineated the trajectory of GC B cell differentiation into plasma cells. Secondly, further investigations using multi-platform spatial transcriptomics and multiplex IHC staining assays revealed that plasma cells drive apoptosis of EBV^{high}-TCA through antibody-dependent cellular cytotoxicity (ADCC) and antibody-dependent cellular phagocytosis (ADCP). Additionally, the apoptosis of EBV^{high}-TCA correlates with the immunotherapy response in NPC patients. Third, we have conducted additional experiments to confirm the spatial co-localization of CXCL13+ CAFs and B cells, as well as the role of CXCL13+ CAFs in facilitating B cell adhesion through VCAM1, chemoattraction via CXCL13, and enhancement of B cell antibody secretion via

TNFSF13B. Furthermore, multiplex IHC staining assays corroborate the spatial co-localization of CXCL13⁺CD8⁺ T and B cells. Additionally, cell-cell interaction revealed that B cells could function as antigen-presenting cells, activating CD8⁺ T cells through MHC-CD8, ICOSLG-ICOS, CD86-CD28, and IL7-IL7R in NPC. In conclusion, our findings, with each section seemingly highlighting the characteristics of distinct cell clusters, collectively elucidate the B cell-centric anti-tumour immune response within NPC.

Minor comments:

1- Some of the abbreviations are not expanded when they are first introduced in the paper (e.g., GC, LTi cells, among others).

Response: Thanks. In this revision, we have provided full spelling for all abbreviations at their first appearance in the text.

2- Figure 5g is not clear; it does not indicate that the data is from different patients.

Response: Thanks. In this revision, we have updated the axis labelling to distinctly indicate the patient ID and CAF. Please kindly review at main **Figure 6a**.

3- The significance of IGH3 being significantly higher in NPC was not mentioned in the paper (see Supplementary Figure 4A).

Response: Thanks. In the initial manuscript, we had stated that "Higher somatic hypermutation (SHM) frequencies of IgH, indicating increased affinity of BCRs associated with antigen recognition and activation." We referred to all heavy chain variants including IGH3 in this statement. To provide better clarity, we have revised the description to explicitly highlight the elevated frequencies of IGHA1, IGHG1, and IGHG3. Please kindly review in Results section in Page 8, Line 1.

4- In the text, Figure 3D was mentioned earlier than Figures 3a-b.

Response: Thanks. In this revision, we have revised the manuscript to rearrange the figures in accordance with their order appearing in the text.

5- The explanations for some figures are not sufficient and clear (e.g., Supplementary Figures 8b-d and Figure 1b).

Response: Thanks. Comprehensive analytic methods and experimental protocols had been provided in the Methods section of our initial manuscript. In this revision, we have supplemented more detailed information for these figures in our manuscript to improve clarity, which are reproduced as below:

Pervious Figure 1b (now Figure 1b): UMAP plots of cell clusters for B cells, CD4⁺ T cells, CD8⁺ T cells, and CAFs. Individual cell types were segregated from the major cell clusters identified in Supplementary Fig 1c and subsequently clustered independently.

Each dot represents a single cell, with colour coded according to the cell types or clusters as indicated in the legend to the right (comprising five, 16, 11, and five clusters for 97,360 B cells, 88,009 CD4⁺ T cells, 95,686 CD8⁺ T cells, and 1,652 CAFs, respectively).

Pervious Supplementary Fig 8b (now Supplementary Fig 12b): Western blotting assay showing the protein expression of GAPDH, E-cadherin (epithelial cell marker), and FAP- α (fibroblast marker) in NPC cell line (HK1 and S26) and CAFs (patient 1, patient 2, and patient 3) derived from NPC tissues.

Pervious Supplementary Fig 8c (now Supplementary Fig 12c): Representative image of flow cytometry analysis for CAFs in NPC biopsy tissue samples. Cells were stained with a VCAM1 antibody.

Pervious Supplementary Fig 8d (now Supplementary Fig 12d): Bar plots showing the relative expression levels of VCAM1, ICAM1, TNFSF13B, and C3 in VCAM1⁺ and VCAM1⁻ CAFs (n = 3). *P* values are derived from two-sided student t-tests. NS: *P* > 0.05, **P* < 0.05, ***P* < 0.01, ****P* < 0.001. The data is presented as mean \pm SD.

6- CD4 is also expressed on other non-T cell populations, so CD3 staining should be included in Figure 1a.

Response: Thanks. In this revision, we have incorporated immunohistochemical staining for CD3, CD4, CD8, and CD20 for NPC biopsy sections (below **Response Figure 9**), providing a comprehensive overview of the TLS architecture. The corresponding description has been updated in the revision. Please kindly review in Results section in Page 6, Lines 6-8 and main **Figure 1a** and **Supplementary Figure 1a**.

Response Figure 9: Multiplex IHC staining of TLS in NPC tissue biopsy. Cells were coloured according to their staining with CD20 (cyan), CD3 (green), CD4 (purple), CD8 (yellow), and FAP- α (red) proteins as indicated on top. The cyan, green, purple, yellow, and red arrows indicated positive cells with the expression of CD20, CD3, CD4, CD8, and FAP- α proteins in NPC tissue, respectively. Images are representative of three biological replicates. Scale bars are 50 μ m or 30 μ m as indicated.

7- There is no discussion explaining why the results in Figure 1e are not consistent between Visium and Stereo-seq for plasma.

Response: Thanks. In this revision, we have incorporated an explanation for the differences in plasma cells between the Visium and Stereo-seq platforms, among these cell types, plasma cells demonstrated a variable distribution between the two cohorts, with the highest proportion in the stromal region of the Stereo-seq cohort (**Figure 1e**), which is

attributed to a high infiltration of plasma cells in one of the three samples examined (**Response Figure 10**). Please kindly review in the Results section in Page 7, Lines 11-14 in this revision.

Response Figure 10: Box plots showing the signature scores of plasma cells from different regions in the Stereo-seq cohort. In box plots, centre lines denote median values, and whiskers denote $1.5 \times$ the interquartile range. *P* values are derived from two-sided student t-tests; ns: $P > 0.05$, **** $P < 0.0001$.

8- The scale bar in Figure 1d should display actual values instead of "high" and "low."

Response: Thanks. We have revised the presentation following the Reviewer's suggestion.

9- There is no discussion explaining why the apoptosis signature is higher in EBVhigh-TCAwoP than in EBVlow-wp (Figures 3a-c).

Response: Thanks. In this revision, we have provided an explanation for the observation of higher apoptosis signature in EBV^{high}-TCAwoP than in EBV^{low}-TCAwoP, as read "Additionally, we also observed a higher apoptosis signature in EBV^{high}-TCAwoP than in EBV^{low}-TCAwoP, which might be explained by the EBV's role in promoting the apoptosis of NPC cells as reported previously". Please kindly review in the Discussion section in Page 19, Lines 26-28 in this revision.

10- The meaning of CR, PR, PD, and SD in Supplementary Figure 2 should be explained.

Response: Thanks. In this revision we have fully spelled out these abbreviations. Please note that we also provided annotations for these terms in **Supplementary Figure 2** for

improved clarity, although these have been described in the Methods and Materials section in the main text.

11- It is not clear if the statistical analysis was performed for comparison between all groups or not (Supplementary Figure 4a).

Response: Thanks. Yes, the statistical analysis was conducted for the cell proportions across all groups. In this revision, we have specified this in the legend.

Reviewer #3, expertise in NPC omics (Remarks to the Author):

In this manuscript, the authors identified several new interesting components in the tumor microenvironment and proposed the role of TLS in immune surveillance in NPC. The study also suggested the action of GC reaction in triggering plasma cell differentiation, and their impacts in producing antibodies targeting “EBV-high” NPC cells. Finally, they concluded the TLS signatures predict the prognosis and immunotherapy response of NPC. In addition to re-analysis of multiple cohorts of single-cell sequencing data, the group has reported the first set of spatial transcriptome findings in this unique EBV-associated cancer. While the five single-cell sequencing studies on NPC have previously been reported and multiple interesting features of NPC tumor microenvironment were identified (Liu et al., Jin et. al., Chen et al., Zhao et al, Gong et al, Supplementary Table 1), the authors of this study focused on the characterizing the TLS and their potential interactions with TCA. The findings further enhance our understanding of the tumor microenvironment of this EBV-associated cancer. Despite of the interesting findings, several key points are needed to clarify to support their conclusion. Most importantly, the “EBV-high/-low” tumor cell classification is needed to be clearly defined.

Major comments:

- A high variety of the proportion of malignancy cells (~0% to >80%) in the NPC biopsies is a major concern for the interpretation of association of GC-B cells proportion with NPC status (early vs advanced, Figure 2b) in the single cells sequencing datasets, bulk and microarray cohorts. The malignancy cell proportion for each sample in these cohorts should be determined to correct the sampling bias during taking biopsies from NPC patients with different disease status. The B cell, T cell and malignant cell proportions in these tumor samples should be shown. The samples containing no malignant cells should be excluded from the study.

Response: Thanks for the thoughtful suggestions. It's true that tumour biopsies generally contain varied proportion of malignant cells. As suggested, we determined the cell proportions for malignant, B, and T cells for each sample (**Response Figure 11**), using CIBERSORTx (Newman, A.M et al, Nat Biotechnol, 2019) for Bulk-RNA-seq and Microarray cohorts. All Bulk-RNA-seq and Microarray samples in this study contain malignant cells, with proportions ranging from 5.63% to 67.29%. In previous version, a few single-cell RNA samples do not contain malignant cells and are excluded in the association analysis between the GC-B cell proportion with NPC status (early vs. advanced, main **Figure 2b**). In this revision, we emphasize the details of cell proportion analysis in the method, please kindly review the content at Methods section in Page 31, Lines 3-4 and **Supplementary Figure 15a**.

Response Figure 11: Bar plots showing the cell proportions (y-axis) of each cell types for each sample (x-axis) in Bulk RNA-seq (top), Microarray (middle), and Single-cell RNA-seq (bottom) cohorts. Cell types are colour-coded and indicated on the right.

- Similar problem should also be considered for determining the prognostic values of TLS components and signatures in different NPC cohorts (Figure 7). The findings may be

misleading since the high changes for successfully collection of more malignancy cells (less lymphoid cells) from the patients with advanced disease (with large and obvious tumors) during endoscopic examination.

Response: Thanks for the suggestion. We agree that it might be possible to collect more malignant cells from the advanced samples. To further support our previous finding, we have included an additional NPC patient cohort with paraffin-embedded tissue sections for TLS staining (CD20; **Response Figure 12a**) to assess the prognostic values of TLS for NPC. The results indicate that NPC patients with TLS had better overall survival (OS) compared to those without TLS (**Response Figure 12b**), consistent with our previous finding, suggesting that the prognostic value of TLS is likely independent of tumour cell content. Please kindly review this updated content in Results section in Page 17, Lines 19-21 and **Supplementary Figure 15f**.

Response Figure 12: Prognostic values of TLS in NPC. a. IHC staining of CD20 in NPC tumour samples. CD20 is considered a marker of TLS. Scale bars are 200µm. b. Kaplan-Meier survival curves of NPC cohorts with patients stratified by presence of TLS in tumour section. Survival duration and probability are indicated at the x- and y-axis, respectively. P value and HR were calculated using a two-sided cox test. OS, overall survival.

The prognostic roles of TLS components in NPC should be validated in a well-selected NPC cohort by multiple IHC analysis, ISH and histological investigation.

Response: Thanks for the suggestion. In this revision, we have included an additional NPC patient cohort with paraffin-embedded tissue sections for TLS staining (CD20; **Response Figure 12a**). These patients were histologically diagnosed at SYSUCC. Survival analysis revealed that NPC patients with TLS had better overall survival (OS) compared to those without TLS (**Response Figure 12b**). Please kindly review this update in Results section Page 17, Lines 19-21.

- The clinical and pathological parameter for the NPC samples included in the Stereo-seq and Visium spatial transcription studies have not been provided. As shown in Figure S1D, ST1 shows absence of TLC, what is the disease status of ST1, ST2 and ST3. It is also noted that there is no TLC in ST17, ST9, ST18, ST10, ST11 sections.

Response: Thanks for the comments. In this revision, we have provided clinical and pathological information for the NPC samples used in the spatial transcriptomics analysis in Supplementary Table 1. ST1, ST2, and ST3 are treatment-naive NPC samples, with TMN stages 2, 4, and 1, respectively. Please note that ST1 sample has TLS, although the area is relatively small (arrowed in below **Response Figure 13a**). The samples ST17, ST9, ST18, ST10, and ST11 did not exhibit TLS regions in our tissue sections. Moreover, we compared the proportions of NPC patients in gender, age, stage, and immunotherapy response groups between their presence of TLS in tumours or not, revealing no significant difference (**Response Figure 13b**). Please kindly review the figure at **Supplementary Figure 2c**.

Response Figure 13: Features of NPC tumour in the spatial transcriptome cohort. **a.** Feature plots showing the presence and locations of sample and region in three NPC tumour biopsies for spatial transcriptome analysis using Stereo-seq (Stereo-seq cohort). Sample IDs are shown as ST1, ST2, and ST3. TLS, TCA, and Stromal regions are defined. **b.** Bar plots showing the proportions of NPC patients with distinct age, gender, stage, and immunotherapy response categories in the with TLS or without TLS group for the ST cohort.

Chip-square test was performed to evaluate the significant difference. NS: not significant (P value > 0.05). R, response; NR, non-response.

- The details of the methods for defining EBV-high and EBV-low TCA in two ST cohorts have not been described in either the results and methodology sections.

Response: Thanks. We indeed had described the definition of EBV-high and EBV-low TCA in the Methods section in our previous submission. To improve clarity in this revision, we have indicated the reference to Methods section in the Results section. Additionally, we have edited the description in Methods section, as read “We defined EBV-related TCA in the FFPE NPC samples of the Visium cohort by utilizing the expression profile of the top 100 DEGs in EBV^{high} NPC malignant cells from our previous study (Liu, Y et al, Nat Commun, 2021). Subsequently, we categorized malignant spots into three TCA groups based on the expression quartiles, including EBV^{high}-(>75%), EBV^{mid}-(>25% and <75%), and EBV^{low}-(<25%) TCA. For fresh-frozen (FF) NPC samples of the Stereo-seq cohort and FF GaC samples of the Visium cohort, we defined EBV-related TCA based on the expression quartiles of EBV encoded genes (RPMS1/A73, BARF0, BALF3/4, LMP-1/BNLF2a/b, LMP-2A/B, EBNA_s, EBER1, and EBER2), including EBV^{high}-(>75%), EBV^{mid}-(>25% and <75%), and EBV^{low}-(<25%) TCA.” Please kindly review the content at in Methods section in Page 32, Lines 17-28, Page 33, Lines 1-6 in the revision.

In Figure 3a and 3b, the corresponding HE image of each sample should be shown. The sFigure 6 shows a poor-quality data and problems for determining EBV gene expression in their ST cohorts. While the abundant expression of EBERs is well known in EBV-associated cancers, either NPC and EBV+GaC, the EBERs signals shown in the ST2, ST3, M39 and M55 are much lower or similar to that of RPMS1/A73, and BARF0, BALF3/4. In s Figure 6F, the detection of LF3, LF2, BALF3, BARF0 in the EBV-high TCA may indicate the activated EBV-lytic cells in these clusters or problems occurred in annotation of EBV genes in the spatial transcriptome studies.

Response: Thanks for the comments. Please note that the corresponding HE images of each sample in Figure 3a and 3b were shown in Supplementary Figure 1d and 2 in our previous submission. We agree with the Reviewer that the abundant expression of EBERs is well known in EBV-associated cancers, and commonly used for NPC diagnosis. RNA-seq transcriptome analysis derived from RNA amplification using random primers reveals much higher EBERs expression in NPC tissues than any other EBV genes (**Response Figure 14**). However, the 10x FF Visium platform captures only Poly (A) tailed RNAs. EBERs contain no Poly (A) tail, while have RPMS1/A73, BARF0, and BALF3/4 (Majerciak, V et al, J Virol, 2019). In this context, we obtained higher expression levels of RPMS1/A73, BARF0, and BALF3/4 compared to EBERs, which is due to unexpected capture. We also revealed this phenomenon in our previous studies (Liu, Y et al, Nat Commun, 2021; Li, Y.Q et al, Adv Sci (Weinh), 2023). Therefore, this is a result due to the inherent design of the 10x FF Visium technology, but not problem in annotation of EBV genes.

Response Figure 14: Bar plot showing the expression levels of EBV molecules in NPC tissues using RNA-seq data. The expression levels of EBV molecules ($\log_{10}(\text{TPM}+1)$; TPM, transcripts per million) and EBV genes are indicated at the x- and y-axis, respectively.

- In sFigure 6F, multiplex IHC staining should also include the staining for B-cells (CD20) and malignant cells (pan-CK). The EBER signals for these regions can be detected in other sections of the samples.

Response: Thanks for the suggestion. We have added the staining results for CD20 and PanCK. We found that EBER signals are present only in NPC malignant cells (PanCK) (**Response Figure 15**). Please kindly review the results in **Supplementary Figure 8d**.

Response Figure 15: Results of in-situ hybridization (top) and multiplex IHC staining (bottom) in NPC biopsy sections. Brown staining on the top panels indicates scattered EBER-positive malignant cells in NPC tissue. Cells in the bottom panels were coloured according to their staining with PanCK (red) and CD20 (green) proteins as indicated. Images are representative of three biological replicates. Scale bars are 500µm and 100µm as indicated.

- In Figure 3d, a high-power view should be shown to demonstrate the EBER signal in single cell level. HE images can help to identify the different cellular features of NPC cells. As shown in the figure, the difference between the two TCA may only be due to cell density, but not EBV signals while the heterogenous morphonology of NPC cells (e.g. spindle cells or cells with round-to-oval vesicular nuclei) is commonly observed in the tissue sections.

Response: Thanks for the suggestion. In this revision, we have provided high-power view of EBER staining with magnified details (**Response Figure 16a**), enabling quantification of EBERs' signals into strong, moderate, and weak levels in individual NPC cells (**Response Figure 16b**). To eliminate the potential influence of cell density, we conducted a statistical analysis of EBV^{high}-TCA and EBV^{low}-TCA, considering the cell proportions with strong, moderate, and weak EBER signals in the two groups. This analysis revealed significantly more cells with strong EBER staining in EBV^{high}-TCA compared to EBV^{low}-TCA (**Response Figure 16c**). This excludes the likelihood that the difference between the two

TCA may be due to cell density. We have added the results in this revision. Please kindly review the Results section in Page 10, Lines 11-14 and **Supplementary Figures 8b-c**.

Response Figure 16: EBERs staining for NPC tissues. a. Representative image of the in-situ hybridization of EBERs in NPC biopsy section. Brown staining indicates scattered EBER-positive malignant cells in NPC tissue. Images are representative of three biological replicates. Scale bars for the left and right panels are 150 μ m and 100 μ m, respectively. b. Representative images of EBERs staining with different intensity (strong, moderate, and weak) in NPC biopsy tissue samples. c. Bar plot showing the percentage of different EBERs staining cells in EBV^{low}- and EBV^{high}-TCA. *P* values are derived from two-sided student t-tests; NS *P* > 0.05, **P* < 0.05, *****P* < 0.0001. The data is presented as mean \pm SD.

- The ST cohort for determination of immunotherapy response in NPC is small (n=12). The authors' conclusion is not convincing and the hypothesis is needed to confirm in a large cohort of NPC patients with ICB therapy.

Response: Thanks for the comments. As the first study of this kind, we aim to dissect the potential mechanisms underlying NPC immunotherapy response and resistance at the spatial transcriptomic level. We plan to expand the sample size in subsequent investigations and validate our findings in additional cohorts, enhancing their clinical

relevance. In this revision, we acknowledge the limitation of sample size. Please kindly review the content in Discussion section in Page 22, Lines 7-10 in this revision.

- In Gong et al study (2021), the CXCL13 high T cells are correlated to better PFS in NPC patient and suggested to be expression in the exhausted T cells, particularly PD-1+ T cells. The findings should be validated in current study and discussed in the results and discussion sections.

Response: Thanks for the suggestion. In this revision, we have compared the CD8⁺CXCL13⁺ T cells identified in our study and the CXCL13⁺TOX⁺CD8⁺ T cells reported by Gong et al (Gong, L et al, Nat Commun, 2021). Both cell clusters exhibit high expression levels of CXCL13, TOX, and HIF1A (**Response Figure 4c**). Furthermore, both studies found that CXCL13 high T cells are correlated with better progression-free survival (PFS) in NPC patients. Therefore, we believe that both cell clusters belong to an identical subset of CXCL13⁺CD8⁺ T cells in NPC, providing in-depth understanding of their functions in NPC development. Additionally, we have further elucidated some novel functions, states, and relationships of CXCL13⁺CD8⁺ T cells in our present study. For example, they represent a subset of pre-exhausted CD8⁺ T cells that are spatially distributed within TLS and associated with the immune therapy response in NPC patients. Please kindly review the content in main **Figure 8a** and Discussion section in Page 21, Lines 2-3.

Minor comments:

- Figure S1C, the gene names listed in x-axis are unable to read! The small characters in Figure S1D and S2 are too small to read. The H&E parts do not show the histological features of the sample clearly. Zoom-in figures should be included to illustrate the representative features of the malignant, B, and T cell regions.

Response: Thanks for the suggestions. We have revised all these relevant figures to improve clarity and presentation.

Reviewer #4, expertise in head and neck cancer, scRNAseq, TME (Remarks to the Author):

The authors performed spatial transcriptomics on 15 nasopharyngeal cancer samples, analyzing spatially-resolved transcriptomes from >30,000 spots. They use this analysis to supplement their re-analysis of >300,000 cells from 77 previously-sequenced NPC samples. They identify immune and stromal cell subpopulations associated with immune checkpoint blockade response and prognosis in nasopharyngeal cancers. The paper is well written and presents interesting analysis of multi-omic data.

Response: We are grateful to the Reviewer for the favourable comments on our work.

Major comments:

1. The authors should clearly define the term tumor cell aggregate and provide clear annotated histologic images highlighting tertiary lymphoid structures and tumor cell aggregates.

Response: Thanks for the suggestion. In this revision, we have provided the definition of tumour cell aggregate (TCA) and annotations of histologic images more clearly (**Response Figure 17**). We defined TCA using NPC malignant cell makers (EPCAM, KRT13, KRT8, and KRT5) and tissue morphology reviewed by pathologists. Moreover, we defined B and T cell aggregates as generalized TLS regions with specific expression of markers for B and T cells, respectively, according to DEG analysis (B cell marker: CD19, MS4A1, CD79A, and CD79B; T cell marker: CD2, CD3D, CD3E, CD3G, and CD7). Additionally, we outlined the boundaries of the TLS and TCA regions using different colours for all ST samples in this revision. Please kindly review the content in Methods section in Page 32, Lines 17-28 and Page 33, Lines 1-6 and **Supplementary Figures 3 and 4** in this revision.

Response Figure 17: Histologic images of NPC samples for spatial transcriptomics assay. Morphological regions were annotated by pathologists into three distinct categories: stromal tissue (purple), TLS (pink), and TCA (red). Scale bars are 1mm.

2. For scRNA-seq results, the authors describe numbers of B cells, CD4 T cells, CD8 T

cells, and CAFs. Were cancer cells analyzed, and if so, how many? Beyond EPCAM expression, were any other techniques (e.g., inferred copy number variation) used to confirm the malignant identify of cells classified as malignant?

Response: Thanks for the comments. In this revision, we have supplemented the analysis of tumour cells. In total, we obtained approximately 20,000 epithelial cells, which were further classified into 14 cell clusters (**Response Figure 18a**). Through inferCNV analysis, we found that all epithelial cells exhibit large-scale copy number variations, indicating their malignant phenotype (**Response Figure 18b**). Moreover, we have also added differential expression gene analysis to define the features of these malignant cell clusters. Please kindly review the content in Results section in Page 6, Line 20 and **Supplementary Figure 1g**.

Response Figure 17: Clustering and annotation of epithelial cells in NPC tissue. a. UMAP plots of cell clusters for NPC malignant cells. Each dot represents a cell, coloured according to the cell clusters indicated in the right legend. b. Heatmap showing the large-scale CNVs for epithelial cells (rows along y-axis) from NPC tumours. CNVs were inferred according to the average expression of 100 genes spanning each chromosomal position (x-axis). Red: gains; blue: losses. Malignant NPC cells from different patients (rows) and the range of different chromosomes are indicated as different colour bars on the top to the heatmap.

3. Throughout the results section, the authors make several statements that are not adequately described from a methodological standpoint or supported by statistical analyses. For example, the statement “co-localization of plasma cells with NK cells and macrophages in NPC tumors” (lines 219-220) should be supported by a description of the methodology and statistical analysis used to draw these conclusions even if these details are provided on the supplemental materials online.

Response: Thanks for the suggestion. In this revision, we have improved the expression of relevant statements by adding methodological evidence. For instance, we have revised the abovementioned statement as read "Furthermore, we observed a significant positive correlation between the proportions of plasma cells and that of macrophages or NK cells in most NPC ST samples (Stereo-seq cohort: 2/3; Visium cohort: 10/12; **Supplementary Figure 7c**), suggesting a widespread colocalization of plasma cells with NK cells and macrophages in NPC tumours." Please kindly review in the Results section in Page 10, Lines 2-4 in this revision. Additionally, we have included the sample numbers for all the

statistical plots in the figure legend, as illustrated in **Figure 2e, 3e, 4a-c, 6a-d** and **Supplementary Figure 5c, 6a, 6g, 6h, 6k, 8a, 8b, 8f, 10d, 13d**, etc.

4. In the spatial transcriptomic analyses, how do the authors define ICA regions? How do ICA regions differ from TLS?

Response: Thanks. We adopted the definition of ICA or Immune Cell Aggregates from a previous study by Tay et al. (Tay, J.K et al, Sci Adv, 2022). ICA encompasses not only TLS (T and B cell aggregates) but also other regions with immune cells aggregate. In this revision, we have provided the definition of ICA in Methods section. Please kindly review at Methods section in Page 33, Lines 10-12.

5. Overall, there is a large amount of data presented, and it is not always clear how these data fit into the context of prior work. For example, the authors make the claim to identify novel clusters of TLS-associated CAFs (line 274) and TLS-associated CD8 T cells (line 320). The authors should clarify specifically what markers make these clusters novel relative to previously described clusters of the given cell type.

Response: Thanks for the suggestion. In this revision, we have provided some comparisons of the novel CAFs and CD8⁺ T cells with previous reported counterparts. For TLS-associated CAFs, we identified novel markers, including CXCL13, PTGDS, ICAM1, and VCAM1, compared to previous reports on NPC CAFs (Jin, S et al, Cell Res, 2020). These markers suggest the function of this novel CAFs in recruiting (CXCL13) and adhering (VCAM1 and ICAM1) CXCR5⁺ B cells (Kazanietz, M.G et al, Front Endocrinol (Lausanne), 2019; Carrasco, Y.R et al, EMBO J, 2006) (**Response Figure 4a**). Regarding TLS-associated CXCL13⁺CD8⁺ T cells, we identified the presence of BCL6 and CXCR5 (as TLS-related markers; Zhao, H et al, Cancer Biol Med, 2021), as well as CD28, CD27, and TCF7 (as stem-like markers; Krishna, S et al, Science, 2020) beyond their common signatures reported previous (Gong, L et al, Nat Commun, 2021) (**Response Figure 4c**). These markers suggest the spatial location in TLS and stem-like state of this novel CXCL13⁺CD8⁺ T cells. In this revision, we have added these comparison results. Please kindly review in Results section in Page 12, Lines 9-17 and Page 14, Lines 18-25.

Minor comments:

1. Please define GC reaction (line 55) in the abstract and in the main text (line 83).

Response: Thanks. We have added the definition of GC reaction in the abstract and in the main text in this revision.

2. There are minor grammatical errors throughout the manuscript that need to be corrected prior to publication.

Response: Thanks. We have double checked the grammar and spelling throughout the manuscript and made corrections.

Reviewer #5, expertise in head and neck cancer, scRNAseq, TME (Remarks to the Author):

Liu et al. perform a comprehensive and extensive investigation of the NPC tumor microenvironment using a very broad set of wet lab and big data analytic techniques. The conclusions are highly interesting and, in most cases, compelling. In certain aspects, the analysis and comparison with published data remains superficial. The data transparency needs to be improved. Overall, the manuscripts provides a valuable addition to the current literature and has implications beyond NPC.

Response: We are grateful to the Reviewer for the favourable comments.

Major Revisions

Usage of published data and online resources

While it is perfectly appropriate to re-analyze already published data sets and cohorts, this needs to be clearer from the text and methods (see line 412: “multiple independent sample cohorts”). Given the highly diverse set of methods it would be good if the authors would provide a supplementary table either for each patient or at least each patient cohort that was (re-)used. This should state the patient characteristics, the methods/techniques that were used on that patient sample, for which figures the data was used in the present paper and if/where the data was previously published (analogous to sFig1). Thus far, this is only performed of the single cell data in Suppl. Table 1.

Response: Thanks for the thoughtful suggestion. In this revision, we have supplemented more detailed information in a new **Supplementary Table 1**, merging with patient characteristics, methods/techniques, and references to published dataset. Please kindly review the **Supplementary Table 1**, of which the header and the top several rows are reproduced as below (**Response Table 2**).

Num	New SampleID	New PatientID	Old PatientID	Clinical stage*	Sample type	Disease condition	scRNA sequencing Technology	Author (PMID)	Therapy
1	Liu_1802_PBMC	Liu_1802	P02	III	PBMC	NPC	10X 5'	Liu et al (33531485)	treatment-naive
2	Liu_1802_Tumour	Liu_1802	P02	III	Nasopharyngeal Tumour tissue	NPC	10X 5'	Liu et al (33531485)	treatment-naive
3	Liu_1805_PBMC	Liu_1805	P05	IVb	PBMC	NPC	10X 5'	Liu et al (33531485)	treatment-naive
4	Liu_1805_Tumour	Liu_1805	P05	IVb	Nasopharyngeal Tumour tissue	NPC	10X 5'	Liu et al (33531485)	treatment-naive
5	Liu_1806_PBMC	Liu_1806	P06	IVa	PBMC	NPC	10X 5'	Liu et al (33531485)	treatment-naive
6	Liu_1806_Tumour	Liu_1806	P06	IVa	Nasopharyngeal Tumour tissue	NPC	10X 5'	Liu et al (33531485)	treatment-naive
7	Liu_1807_PBMC	Liu_1807	P07	III	PBMC	NPC	10X 5'	Liu et al (33531485)	treatment-naive
8	Liu_1807_Tumour	Liu_1807	P07	III	Nasopharyngeal Tumour tissue	NPC	10X 5'	Liu et al (33531485)	treatment-naive
9	Liu_1808_PBMC	Liu_1808	P08	IVa	PBMC	NPC	10X 5'	Liu et al (33531485)	treatment-naive
10	Liu_1808_Tumour	Liu_1808	P08	IVa	Nasopharyngeal Tumour tissue	NPC	10X 5'	Liu et al (33531485)	treatment-naive
11	Liu_1810_PBMC	Liu_1810	P10	III	PBMC	NPC	10X 5'	Liu et al (33531485)	treatment-naive
12	Liu_1810_Tumour	Liu_1810	P10	III	Nasopharyngeal Tumour tissue	NPC	10X 5'	Liu et al (33531485)	treatment-naive
13	Liu_1811_PBMC	Liu_1811	P11	III	PBMC	NPC	10X 5'	Liu et al (33531485)	treatment-naive
14	Liu_1811_Tumour	Liu_1811	P11	III	Nasopharyngeal Tumour tissue	NPC	10X 5'	Liu et al (33531485)	treatment-naive
15	Liu_1813_PBMC	Liu_1813	P13	III	PBMC	NPC	10X 5'	Liu et al (33531485)	treatment-naive
16	Liu_1813_Tumour	Liu_1813	P13	III	Nasopharyngeal Tumour tissue	NPC	10X 5'	Liu et al (33531485)	treatment-naive
17	Liu_1815_PBMC	Liu_1815	P15	II	PBMC	NPC	10X 5'	Liu et al (33531485)	treatment-naive
18	Liu_1815_Tumour	Liu_1815	P15	II	Nasopharyngeal Tumour tissue	NPC	10X 5'	Liu et al (33531485)	treatment-naive
19	Liu_1816_PBMC	Liu_1816	P16	IVb	PBMC	NPC	10X 5'	Liu et al (33531485)	treatment-naive
20	Liu_1816_Tumour	Liu_1816	P16	IVb	Nasopharyngeal Tumour tissue	NPC	10X 5'	Liu et al (33531485)	treatment-naive
21	Jin_33P_Tumour	Jin_33P	NPC33	IVc	Nasopharyngeal Tumour tissue	NPC	10X 3'	Jin et al (32901110)	treatment-naive
22	Jin_36P_Tumour	Jin_36P	NPC36	IVb	Nasopharyngeal Tumour tissue	NPC	10X 3'	Jin et al (32901110)	treatment-naive

Response Table 2: clinical characteristics for NPC cohorts.

Relevance of differences in GC_B

The authors go into great detail exploring the relevance of differences between early and advanced cancer GC_B cell proportion. However, looking at Fig. 2b and sFig 4b, the actual

quantitative difference seems to be minor. Please include the proportions and expand on why this difference is relevant.

Response: Thanks for the suggestions. In the revised Figures, we have indicated the proportions of GC_B cells in early (3.3%) and late (1.0%) stages of NPC. Although the difference in the absolute numerical cell proportion is relatively small, the proportion has significantly decreased by 70% from early stage to late-stage NPC. We note that the abundance of GC cells is indeed low in pan-cancer context (Ruffin, A.T et al, Nat Commun, 2021; Hao, D et al, Cancer Discov, 2022). Despite their low numbers, GC cells exhibit a robust proliferative capacity, generating a substantial quantity of memory B cells and plasma cells (Akkaya, M et al, Nat Rev Immunol, 2020). These cells play a role in influencing the tumour microenvironment through the production of antibodies, cytokines, and other factors (Laumont, C.M et al, Nat Rev Cancer, 2022). Additionally, a small number of plasma cells can effectively contribute to producing a large quantity of antibodies (Sharonov, G.V et al, 2020). These findings suggest that small number of cells do matter in biological function and the dramatical change in proportion of GC_B cells is likely functionally relevant. Please kindly review the expanded comparison in Page 8, Lines 19-20 in the Result section and Page 18, Line 16 in Discussion section.

Annotation of cell types

The annotation of single cell clusters by single makers should be avoided (e.g., CD8_C10_STMN1). Instead, one should try allocating each cluster or group of clusters into a known/published subtype or substate. If there is a “new” cell state or subtype, as might be in the present study for the CXCL13+ CD8 T cells, this needs to be compared with the existing scRNAseq annotation literature. Of note, there is data from non-NPC HNSCC where the expression of GZMK and CXCL13 in T cell substates has been described (Kürten et al. Nat. Comm., 2021). A broad review of single cell T cell states can be found in Andreatta et al. Nat. Communications 2021. Surely, within the T cell analysis there needs to be a distinction between early and late exhaustion (see Philip and Schietinger, Nat. Rev. Immun., 2021).

Response: Thanks for the insightful suggestions. We agree with the Reviewer that multiple makers may be more precise in defining a cell cluster. Indeed, we annotated CD8_C10_STMN1 cluster using both STMN1 and markers for CD8⁺ T cells. To determine whether a cell cluster is new or novel, we usually compare its expression profile with the existing literature with scRNA-seq data, as suggested by the Reviewer. For CD8_C10_STMN1, we had conducted a correlation analysis of CD8⁺ T cell clusters identified in this study and our previous study (Liu, Y et al, Nat Commun, 2021; GEO database: GSE162025). We observed a high correlation of this cluster and the previous highly proliferative CD8⁺ T cell (CD8_C10_MKI67; **Response Figure 19**), suggesting that they may represent the same cluster. Given the higher expression level and broader expression of STMN1 in this cluster compared to MKI67, we annotated the cluster as CD8_C10_STMN1.

For CD8_C8_CXCL13, we did not observe its high correlation with any other T cell clusters (**Response Figure 19**), indicating that CD8_C8_CXCL13 may represent a novel

cell cluster. Furthermore, DEG analysis also revealed high expression of both GZMK and CXCL13. This is different from the GZMK-high/CXCL13-low and GZMK-low/CXCL13-high CD8⁺ T cell subpopulations reported previously by Kürten et al. (Kurten, C.H.L et al, Nat Commun, 2021). Andreatta et al.'s study (Andreatta, M et al, Nat Commun, 2021) primarily includes T cell data from tumour infiltrations in melanoma and colon adenocarcinoma, as well as murine tumour data. Given the potential differences in T cell states among different tumours, we conducted a comparison with another published research on T cells within NPC (Gong, L et al, Nat Commun, 2021). Interestingly, we found that both CXCL13⁺CD8⁺ T cell groups in two studies exhibit elevated expression of TOX and HIF1A (**Response Figure 4c**). Moreover, combining our pseudotime development analysis and TCR transition analysis, we revealed that CD8_C8_CXCL13 represents a precursor state of terminally exhausted CD8⁺ T cells in NPC. These cells exhibit similarities to the progenitor exhausted T cells described by Philip et al. (Philip, M et al, Nat Rev Immunol, 2022), expressing CXCR5, PD1, TOX, and TCF7 (**Response Figure 4c**). Taken together, we have identified CD8_C8_CXCL13 as a subgroup of pre-exhausted CD8⁺ T cells in NPC. Please kindly review the content in the Results section in Page 14, Lines 18-29 and Page 15, Lines 1-24.

Response Figure 19: Heatmap showing the correlation coefficient for CD8⁺ T cell clusters between this study (rows) and previous study (GSE162025; columns). Filled colours from blue to red represent scaled expression levels from low to high.

Underlying sample size (n = ?), proportion difference and p-value

Each Figure and Graph needs to state in the Figure legend how many patients, biological repeats or number of cells are depicted or used for the respective Figure. This is rarely performed throughout the manuscript, for example Fig 2a how many cells? 2b,c,d how many patients? Fig 2e how many slides? Etc., the same is true for the p-value, which is not reported consistently, see Fig. 2b, c, d. Also, for differences in proportions or means

between groups the mean or median value should be stated (e.g., when describing sFig 5c or 2g)

Response: Thanks for the suggestions. In this revision, we have added the sample size, proportion difference, and p-value for each figure.

Depth and breadth of discussion

Given the high amount of data (18 Figures) the discussion falls a bit short of putting all these aspects into context with the known literature. Especially the role of B cells/ Tfh/ TLS in HNSCC needs to be expanded on (Bruno Nature 2020, Ruffin Nat. Comm. 2021, Cillo Immunity 2020) as well as prognostic and therapeutic implications.

Response: Thanks for the suggestion. In this revision, we delve into the prognostic and therapeutic implications of B cells, Tfh, and TLS in HNSCC in the discussion section. Please kindly review in Discussion section in Page 22, Lines 1-4.

Minor revisions

Use of different spatial transcriptomics platforms

The authors should elaborate on why two different ST platform were used.

Response: Thanks. Initially, we used the Visium FFPE platform because of its feasibility to handle FFPE samples, allowing us to retrospectively assess the efficacy of immunotherapy in NPC patients. We further included the Stereo-seq FF platform, which offers advantages of a wide-field view and high resolution for the detection of minute structures within tumours. Combining both technologies not only enables us in dissecting the cellular composition and expression profiles of TLS but also facilitates our exploration of the relationship and impact of TLS on immunotherapy. We have elaborated this in the revised manuscript. Please kindly review in Methods section in Page 26, Lines 15-21 in this revision.

Further Implications

The authors should expand on the implications of their findings for other non-NPC but virus associated cancers (line 455) within and outside the head and neck region (cervix, liver, Merkel cell etc.).

Response: Thanks for the suggestion. In the discussion section, we delve into the implications of TLS-related cell clusters for other non-NPC but virus-associated cancers, as read “Additionally, our study provides insights into understanding the TLS formation and functions orchestrating by B cells and pathogen infection in other virus-associated cancers such as HPV⁺ head and neck cancer (Ruffin, A.T et al, Nat Commun, 2021), HPV⁺ cervical squamous carcinoma (Qiu, J et al, Adv Sci (Weinh), 2023), HBV⁺ liver cancer (Zhang, C et al, Gut, 2023), etc, where TLS has been demonstrated (Qiu, J et al, Adv Sci (Weinh), 2023; Zhang, C et al, Gut, 2023). Please kindly review the content in Discussion section in Page 21, Line 29 and Page 22, Lines 1-4 in this revision.

Pathological response data

Is there pathologic response data for the PD-1 treated patients (line 541). If so, that should be used rather than RECIST criteria for evaluating response to immunotherapy. Radiographic imaging is unreliable when it comes to evaluating immunotherapy response (see Forde et al. NEJM 2022).

Response: Thanks. We don't have pathologic response data for the PD-1 treated patients. Thus, we used RECIST criteria as an alternative way to evaluate response to immunotherapy in NPC, following previous studies in NPC by Chen et al. (Chen, Y.P et al, Mol Cancer, 2021) and Li et al. (Yuan, L et al, Nat Commun, 2023).

Fig 2A

Move annotation next to figure

Response: Done.

Fig 2f

Explain "NES" in Fig. legend

Response: Done. NES represents normalized enrichment score.

Fig 6a

Combine Fig. 6a with s9a to give comprehensive overview of marker expression.

Response: Thanks for the suggestion. We have combined the two figures into a new **Figure 7a**.

sFig4

Explain why number of patients in the early cohort sFig 4 is so much lower than advanced stages.

Response: NPC patients often have a hidden onset symptom, and by the time they seek medical attention, most of them are already in the advanced stages (Chen, Y.P et al, Lancet, 2019). We recruited fresh samples for scRNAseq without prior knowing the staging criteria, so it is likely that we obtained more samples from advanced stages.

sFig5 should read "plasma cells" not "plasma"

Response: Thanks. We have made the correction. Please kindly review in **Supplementary figure 7c**.

The grammar should be checked, especially plural forms such as "Tertiary lymphoid structureS (TLS) represent" or "A Tertiary lymphoid structure (TLS) represents" line 75 and "Germinal centreS (GC) are a" or "A Germinal centre (GC) is a" line 88

Response: Thanks. In this revision, we have carefully checked the grammar to improve the readability.

Line 54

Tfh is not described in study (line 54) beyond Fig 1b upper right panel. Maybe expand?

Response: Thanks. As suggested, we have expanded a bit for the analysis of Tfh, including differentially expressed genes (**Supplementary table 2**), cell-cell interaction (Page 16, Lines 19-21), and survival analyses (Page 17, Lines 10-12) in the revised manuscript.

Line 60

The tone of the abstract needs to be more cautionary, e.g., line 60: this is not shown specifically in the paper. Rather there are associations between the cell types that “could” mean this “may” be happening.

Response: Thanks for the suggestion. In this revision, we have toned down this statement in the abstract, as read “B cells may activate CXCL13⁺CD8⁺ T cells in TLS”. Please note that we have provided additional assays to corroborate the interaction between CXCL13⁺ CAFs and B cells.

Naming of clusters

The authors should consider naming the cell clusters without using “_” for B cells, T cells, CAF etc.: “naïve B cells” or “CXCL13⁺ CAF” is easier to read than “Naïve_B” and “iCAF_C2_CXCL13 CAF”.

Response: Thanks for the suggestion. For some well-known clusters, such as naïve B cells, we use names as suggested in the manuscript. For other clusters, we would like to keep the current nomenclature of cell subpopulations, considering its consistency with our previously published papers and easier identifiers for numerous cell subpopulations or states. To enhance the distinction between these subpopulations, we have assigned numerical identifiers to them. For instance, “Naïve_B” is modified to “naïve B cell” and “iCAF_C2_CXCL13 CAF” is modified to “CXCL13⁺ CAF”.

Line 266

Should “degree” read “ability”?

Response: Thanks. We have changed the term "degree" to "ability", now in Results section in Page 12, Line 4.

Reviewer #6, expertise in TLS structures (Remarks to the Author):

Through the analysis of single-cell and spatial-resolved transcriptomes, Liu and colleagues explored the composition, function, and clinical relevance of tertiary lymphoid structures (TLSs) in a large cohort of Nasopharyngeal Carcinoma (NPC) patients. The authors claim to have identified novel cell populations within tumor-associated TLSs, including GC B cells that support antibody production. Additionally, the authors claim that antibody-secreting plasma cells promote apoptosis of EBV+ tumor cells, probably by antibody-dependent cellular cytotoxicity or antibody-dependent (ADCC) or antibody-dependent cellular phagocytosis (ADCP). Also, CXCL13-expressing fibroblasts promote B cell adhesion and antibody production, whereas TLS-associated CXCL13-producing CD8 T cells differentiate towards TCA-associated exhausted CD8 T cells with decreased stemness and TCR diversity. Finally, all this TLS apparatus was associated with better prognosis and immunotherapy response in NPC patients. Although the manuscript has positive points, most of the conclusions claimed by the authors are not supported by the data presented, due to the highly descriptive nature and circular analyses of the manuscript. Therefore, many conclusions need to be toned down or further supported by *in vitro* experiments.

Response: Thanks for the comments and suggestions. In this study, we aim to delineate the components of TLS in NPC, identifying novel cell clusters essential for B cell function and TLS formation. Following the suggestions from this Reviewer, we have supplemented more bioinformatic analyses and *in vitro* experiments to support our findings in this revision. Additionally, we have toned down some conclusions with less experimental validations. We believe that our revised manuscript has been significantly improved by addressing all comments from the Reviewers.

1. Germinal Centers are a very dynamic and specialised structure. Circular trajectories, as shown elsewhere (PMID 37230755), are definitely more appropriate to represent GC reactions in TLSs. In the introduction, authors claim that little is known about GC reactions within TLSs, and this reference must be cited.

Response: Thanks for the suggestions. For a better representing GC reaction in our data, we performed circular trajectories analysis of GC B cells in NPC. We observed a characteristic continuum of gene expression states spanning the dark-light zone axis (Gabriel D et al, Blood, 2012) (**Response Figure 20a**). Additionally, dark zone B cells exhibit upregulated expression of AICDA, while light zone B cells showed elevated CD83 expression (**Response Figure 20b**), consistent with previous findings (Milpied, P et al, Nat Immunol, 2018; Kinker GS, et al, Gut, 2023). These data suggest a common pattern of GC reactions in TLSs. We have added the results in this revision. Please kindly review in Results section in Page 8, Lines 10-15 and **Supplementary Figures 6c-e**.

In this revised manuscript, we also cited the finding by Kinker et al. in the introduction section, as read “A recent study has demonstrated GC B cells characterized by a continuum of gene expression states spanning the dark-light zone axis”. Please kindly review in Introduction section in Page 4, Lines 23-25.

Response Figure 20: Circular trajectory analysis of GC B cells in NPC TLS. a. PCA analysis of GC B cells showing PC1xPC2 coordinates with cells coloured by the score of GC light and dark zones (left panel). b. Radial projection of PC1xPC2 cartesian coordinates highlights a circular trajectory coloured by cell cluster (left panel), and the expression levels of light zone (CD83; middle panel), and dark zone (AICDA; right panel) markers are shown. Filled colours from blue to red represent scaled expression levels from low to high.

2. If GC reaction within TLSs is critical to generate high affinity antibodies against EBV, virus antigens are the trigger that generate TLS formation in NPC patients. This should be addressed.

Response: Thanks for the comment. Indeed, our study reveals that GC reaction is essential for the maturation of plasma cells from TLS to tumour cell aggregate (TCA). Furthermore, our study also reveals higher apoptosis status at the TCA with plasma cells, especially in the EBV^{high}-TCA. These findings suggest that the matured plasma cells generate antibodies against EBV-related tumour cells in NPC. Additionally, we observed a higher content of GC B cells in EBV-positive gastric cancer (GaC) compared to EBV-negative GaC (**Response Figure 21**). This is consistent with the finding in human papillomavirus (HPV)-related head and neck cancers, with a higher proportion of GC B cells in HPV-positive cases compared to HPV-negative cases (Ruffin, A.T et al, Nat Commun, 2021). Together, these findings suggest that virus and/or tumour associated antigens due to viral infection may act as triggers for the formation of TLS in cancer patients. In this revision, we have incorporated this content in the Discussion section. Please kindly review in Discussion section in Page 9, Lines 19-22 and **Supplementary Figure 6k**.

Response Figure 21: Box plots showing the GC B cell signature (a) and proportion (b) in samples from the Bulk-RNA-seq (left) and scRNA-seq (right) GaC cohorts, respectively. Centre lines denote median values, and whiskers denote 1.5 × the interquartile range. *P* values are derived from two-sided student t-tests. The GC B cell signature and EBV+/- groups are indicated at the y- and x-axis, respectively.

3. GC B cells are quite abundant in normal tissues (Fig. 2b and Suppl. Fig. 4b) – it is probably more abundant in normal tissues than in the other sites. Why?

Response: Thanks for the comment. Normal nasopharynx contains nasopharynx-associated lymphoid tissue (NALT), which is part of the nasal immune system and includes many T and B cells (Wu, H.Y et al, Immunology, 1996). Therefore, this may explain the abundant GC B cells in the nasopharynx observed in our study. Supportively, a previous paper has demonstrated that the proportion of GC B cells in normal tissue is significantly higher than in tumour tissue in head and neck cancer (Ruffin, A.T et al, Nat Commun, 2021).

4. If plasma cells derive from GC B cells in a linear direction, it seems contradictory to have high abundance of GC B cells and low abundance of plasma cells in early stages, and low abundance of GC B cells and high abundance of plasma cells in late stages of NPC. What is the physiological advantage of having high abundance of GC B cells in early stages of NPC if these cells are not being converted in highly effective plasma cells?

Response: Thanks for the insightful comment. We don't think that there is a contradiction in cell proportions for GC B and plasma cells in NPC. BCR transition analysis reveals a significantly higher BCR sharing between GC B cells and plasma cells in NPC at the early stage compared to the advanced stage, suggesting a linear relationship between GC B cells and neonatal plasma cells, but not all plasma cells (**Response Figure 22a**). Moreover, cell proportion analysis indicates a significant difference in the GC B cell proportion, rather than all plasma cells, between samples at early and advanced stages (**Response Figure 22b**). Additionally, we observed higher antibody levels in NPC samples with early stage compared to advanced stage (**Response Figure 22c**). Taken together, these data suggest a linear conversion of GC B cells to neonatal plasma cells, generating higher antibody levels at the early stage compared to advanced stage in NPC. To help better convey this message, we have added the content in the Results section. Please kindly review in Page 8, Lines 25-29 and Page 9, Lines 1-2.

Response Figure 22: The BCR transition, cell proportion, and expression profiles of different B cell clusters. a. Bar plot showing BCR transition scores among B cell clusters derived from different sources and stages of NPC. b. Box plots showing the expression levels of antibodies in samples from different sources and stages of NPC. c. Box plots showing the expression levels of antibodies in samples from different sources and stages of NPC. For box plots, centre lines denote median values, and whiskers denote $1.5 \times$ the interquartile range. P values are derived from two-sided student t-tests; NS, $P > 0.05$; $*P < 0.05$; $****P < 0.0001$.

5. If the abundance of antibody secreting plasma cells in similar across NPC samples at different stages (Fig. 2b), this should not be associated with tumor progression.

Response: Thanks for the comment. As mentioned above, we want to emphasize the linear relationship between GC B cells and neonatal plasma cells. We did observe a decrease in the proportion of neonatal plasma cells produced from GCs during tumour progression, comparing samples at early stage to advanced stage. In this revision, we have indicated more precisely the link between GC B cells and neonatal plasma cells, rather than all plasma cells. Please kindly review the content in the Results section in Page 8, Lines 25-29.

6. The authors seem confused about the differences between GC B cells and plasma cells.

Response: We apologize for any unclear descriptions regarding the definitions for the two cell types. In our study, we fully understand the differences between GC B cells and plasma cells. In our previous submission, we were aware of the distinct transcriptomic profiles between GC B cells and plasma cells using straightforward single-cell transcriptomics analysis. We also listed out the differential genes for these two cell clusters, among other B cell clusters, in **Supplementary Table 2**. Considering that plasma cells generated through GC reactions (Playoust, E et al, Cell Mol Immunol, 2023), we also utilized trajectory

analysis and BCR transition to delineate the differentiation trajectory from GC B cells to neonatal plasma cells in NPC (main **Figure 2a** and **2c**).

In this revised manuscript, we have compared the expression levels of antibody heavy chain-related genes (IGH), plasma cell-related transcription factors (MZB1 and PRDM1), proliferation genes (MKI67), and germinal centre marker genes (CXCR5, BCL6, AICDA, RGS13, and MME; Kim, W et al, Nature, 2022; Holmes, A.B et al, J Exp Med, 2020) for GC B cells and plasma cells using a heatmap (**Response Figure 23**). Please kindly review in Results section in Page 8, Line 5 and **Supplementary Figure 6b**.

Response Figure 23: Heatmap showing the normalized mean expression of B cell-related genes (rows) in different B cell clusters (columns). Filled colours from blue to red represent scaled expression levels from low to high.

7. Hard to understand whether the apoptotic signature is coming from tumor cells. Single cell transcriptomics data should be used to address this and better correlate with plasma cells.

Response: Thanks for the suggestion and apology for the unclear description. The apoptotic signature is estimated for tumour cells. As suggested, we have provided correlation analysis between plasma cell proportion and tumour cell apoptosis at the single-cell level (**Response Figure 24**). This analysis reveals a significant positive correlation between the proportion of plasma cells and apoptotic signature score in tumour cells. We have added this content in the revised manuscript. Please kindly review in Results section in Page 10, Lines 17-19 and **Supplementary Figure 8f**.

Response Figure 24: Scatter plots showing the pair-wise correlation of tumour apoptotic signature (x-axis) and the proportion of plasma cells (y-axis) in the NPC scRNA-seq cohort. 32 NPC samples were included, shown as dots. R value represents Pearson's correlation. P value is calculated by two-sided Pearson correlation analysis.

8. The authors claim that novel cell populations were identified within TLSs. All populations shown here have been shown elsewhere (several papers).

Response: Thanks for the comment. Recent studies have demonstrated key TLS-related cell populations in tumours, using both clinical human samples and mouse models. For our finding of a novel CXCL13⁺ CAFs in NPC TLS, we note its first discovery in human tumours. Moreover, we have thoroughly characterized the expression profile and functions of CXCL13⁺ CAFs in clinical human samples and validated these findings using in vitro assays. By contrast, Rodriguez et al reported CAF expressing CXCL13 in mice (Rodriguez et al, Cell Rep, 2021). For more details, please refer to our response to the Reviewer #2, points 1 and 2.

Our study also uncovers the transcriptional features, differentiation trajectory, and spatial location of CXCL13⁺CD8⁺ T cells, as well as its correlation with immunotherapy response in NPC. Although CXCL13⁺CD8⁺ T cell clusters have been reported human tumours (Liu, B et al, Nat Cancer, 2022), they are basically different from each other in terms of expression profile, functional states, and differentiation relationships. For more details, please refer to our response to the Reviewer #2, points 1 and 3.

In our study, we aim to discover the TLS at a higher resolution and spatial dimension level, by using single-cell and spatial transcriptomics in NPC. Our study reveals both novel cell clusters and known cell subpopulations in TLS. Our study further highlights the crucial interactions among different TLS-related cell subpopulations and their contribution to NPC prognosis and immunotherapy responses. These findings provide new insights into

understanding the contribution of TLS components to NPC development and developing treatment strategies targeting TLS for NPC.

9. Apart from correlative data, no data prove that i) antibody-secreting plasma cells promote apoptosis of EBV+ tumor cells, ii) CXCL13-expressing fibroblasts promote B cell adhesion and antibody production, and iii) TLS-associated CXCL13-producing CD8 T cells differentiate towards TCA-associated exhausted CD8 T cells with decreased stemness and TCR diversity. All conclusions need to be toned down or supported by extra experiments.

Response: Thanks for the suggestions. In this revision, we have incorporated additional experiments to further support our findings that i) antibody-secreting plasma cells promote apoptosis of EBV^{high} malignant cells and ii) CXCL13⁺ CAFs promote B cell adhesion and antibody production. For more details, please refer to our response to the Reviewer 2, point 2. We have included these findings in our revised manuscript. Please kindly review in Results section in Pages 11, Lines 17-28 and Pages 12-14 in this revision.

Regarding iii) the differentiation of CXCL13⁺CD8⁺ T cells to exhausted CD8⁺ T cells (CD8_C11_HAVCR2), it is still challenging to capture the process of dynamic migration of immune cells within human tumours. Additionally, establishing an appropriate spontaneous mouse tumour model is still a desirable goal in NPC research community. Although our bioinformatics analysis results offer valuable insights for the migration and differentiation of CXCL13⁺CD8⁺ T cells in NPC, we have toned down our conclusion, as read “Together, these observations might suggest a development trajectory from CD8_C8_CXCL13 cells in TLS to exhausted CD8_C11_HAVCR2 cells widespread into TCA along with decreased stemness and TCR diversity in NPC.”. Please kindly review in Results section in Pages 14-16.

10. Please specify what the immunosuppressive signature means (Fig. 4d). This result is out of context.

Response: Thanks. The immunosuppressive signature was defined as the expression level of genes related to known immunosuppressive functions (Sun, Y et al, Cell, 2021), including CD47, PVR, CD276, LGALS9, ADORA2B, ADAM10, HLA-G, CD274, FASLG, TGFB1, and IL10. We used the immunosuppressive signature to assess the immunosuppressive function of malignant cells. In this revision, we have updated the definition and purpose. Please kindly review in Methods section in Page 30, Lines 11-15 in this revision.

11. Fig. 8 needs to be removed, as it is based on lots of assumptions. The data from Liu and colleagues do not support the model proposed.

Response: Thanks. We have moved **Figure 8** to the **Supplementary Figure 16** in this revision.

Minor comments:

- Title seems inaccurate.

Response: Thanks. We have revised the title as read “Single-Cell and Spatial Transcriptome Analyses Reveal Tertiary Lymphoid Structures Linked to Tumour Progression and Immunotherapy Response in Nasopharyngeal Carcinoma”, where transcriptome is added to reflect the main data we used for discovery.

- Antiviral responses (against EBV) are not introduced throughout the manuscript.

Response: Thanks. We have supplemented current understanding on antiviral responses (against EBV) in the Discussion section. Please kindly review in the Discussion section in Page 19, Lines 18-22.

- Why did the authors use two protocols of spatial transcriptomics?

Response: Thanks. We used these two platforms for different sample types and purposes. For more details, please refer to our response to the Reviewer 5, Minor Revisions, Point 1.

- For the ST analyses, anything that was not TLS or TCA was defined as stromal regions. This reviewer was wondering if immune cells sparsely distributed across the TME could interfere in such analyses, as they were included as stromal components.

Response: Thanks for the comment. We defined TLS as areas encompassing both T cell and B cell clusters, and TCA as tumour cell aggregate, along with morphological features. Furthermore, our analyses focused primarily on significant outcomes related to TLS and TCA regions, without considering the stromal region. Therefore, the sparsely distributed immune cells across the TME will not interfere with our analyses for TLS and TCA.

- Other TLS examples of multiplex IHC should be added as a Suppl. Figure.

Response: Thanks. In this revision, we have supplemented additional multiplex IHC data for other TLS samples. Please kindly review the **Supplementary Figure 1a**, which is reproduced as above **Response Figure 9**.

- What is the biological tissue sample used for stereo-seq?

Response: Thanks. In the revision, we have added the information that fresh-frozen NPC biopsy samples were used for Stereo-seq.

- According to the tissue of origin, please show the UMAP plots (Suppl. Fig. 1b) distributed into 3 groups only: blood samples, tumor tissue samples, and non-tumor tissue samples.

Response: Thanks for the suggestion. We have supplemented UMAP plots to illustrate the origin of each cell (**Response Figure 25**). Please kindly review the **Supplementary Figure 1c**.

Response Figure 25: UMAP plot illustrates single cells identified in this study (n=343,829). Each dot represents one cell, coloured according to their origins from tumour, normal tissue, or PBMC.

- It will be great to see the distribution of non-TLS associated cell populations (such as CXCL13⁻ T cells, CXCL13⁻ CAFs etc.) as negative controls in Fig. 1e and Suppl. Fig. 3e.

Response: Thanks for the suggestion. We have previously supplemented the distribution of non-TLS-associated CXCL13⁻ T cells in previous **Figure 6f** (new **Figure 7f**). In this revision, we have added the distribution of non-TLS associated CXCL13⁻ CAFs as negative controls in the **Supplementary Figure 11c** and reproduced below as **Response Figure 26**.

Response Figure 26: Box plots showing the signature scores of CXCL13⁺ CAFs in different tumour regions from NPC patients of the Stereo-seq (left) and the Visium cohorts (right). Centre lines denote median values, and whiskers denote 1.5 × the interquartile range. *P* values are derived from two-sided student t-tests. *****P* < 0.0001.

- Correlations between NK and macrophage abundances with plasma cell abundances are weak.

Response: Thanks. Our multiplex IHC staining assay revealed co-localizations between NK cells and macrophage with plasma cells, providing spatial contact for interactions (**Response Figure 27**). The results have been updated in the **Supplementary Figure 7c**. Our analysis reveals significant correlations between NK and macrophage abundances with plasma cell abundance, despite the relatively small correlation coefficient (*R*). This might be attributed to the pluripotency of plasma cells interacting with other cell types besides NK cells and macrophage, such as CXCL13⁺ CAFs observed in our study. Indeed, from the multiplex IHC assay, we observed more plasma cells colocalize with other cells far away from NK cells and macrophages.

CD68 CD56 IgG

Response Figure 27: Representative image of multiplex IHC staining of NPC tissue biopsy. Cells were coloured according to their staining with IgG (green), CD56 (yellow), and CD68 (red) proteins as indicated on top. The green, yellow, and red arrows indicated positive cells with the expression of IgG, CD56, and CD68 proteins in NPC tissue, respectively. Images are representative of three biological replicates. Scale bar is 50µm as indicated.

- Correlations between CXCL13+ fibroblast abundance with plasma cell abundance is also weak.

Response: Thanks. Likewise, our multiplex IHC staining assay revealed co-localizations between CXCL13+ CAFs with plasma cells, providing spatial contact for interactions (**Response Figure 28**). The results have been updated in the **Supplementary Figure 11f**. The weak correlation, though statistically significant, might be explained by the pluripotency of plasma cells interacting with many other cell types including NK cells and macrophage.

Response Figure 28: Representative image of multiplex IHC staining of CXCL13⁺ CAFs in NPC tissue biopsy. Cells were coloured according to their staining with CXCL13 (green), IgG (yellow), CD20 (purple), and FAP-α (red) proteins as indicated on top. The green, yellow, purple, and red arrows indicated positive cells with the expression of CXCL13, IgG, CD20, and FAP-α proteins in NPC tissue, respectively. Images are representative of three biological replicates. Scale bar is 10µm as indicated.

- Expression of FAP-alpha by CAFs is weak for CAF-1 and CAF-2 (Suppl. Fig. 8b)

Response: Thanks. In this revision, we repeated the western blotting assay and observed a strong expression of FAP-alpha in three CAFs. The results have been updated in the **Supplementary Figure 12b** and reproduced as below (**Response Figure 29**).

Response Figure 29: Western blotting assay showing the protein expression of E-cadherin (epithelial cell marker) and FAP-α (fibroblast marker) in CAFs derived from NPC patients (patient 1, 2, and 3). NPC cell lines HK-1 and S26 of epithelial origin were used as control.

- Sentences that are not clear and need to be rephrased: (line 266: ICB therapy response might be mediated through the degree of ICB to reverse the immunosuppression of EBV^{high} malignant cells, line 272: no canonical marker (CR2) expression was detected in the fibroblasts in NPC samples).

Response: Thanks. In this revision, we have rephrased the two sentences as below.

Page 12, Lines 4-5 (previously line 266): ICB therapy response might be mediated through the ability of ICB to reverse the immunosuppression of EBV^{high} malignant cells.

Page 12, Lines 10-11 (previously line 272): The expression of their canonical marker (CR2) is scarcely detected in CAFs in NPC samples.

- Suppl. Fig 1a has a typo (Please replace immunotheray by immunotherapy).

Response: Thanks. We have corrected the typo in this revision.

Overall, this reviewer found that the manuscript is very descriptive and fails to propose a mechanism that is not supported by the data shown by Liu and colleagues.

Response: Thanks. By addressing the comments from all the six reviewers, we have conducted substantially additional experiments and data analyses to support our previous findings. Cross-validation across multiple omics datasets, utilizing widely adopted analytic pipeline and functional assays, ensures the reliability and authenticity of our findings. We believe that our revised manuscript has been significantly improved.

In this revision, we have reorganized the manuscript to elucidate the anti-tumour immune response orchestrated by B cells and TLS in NPC. First, single-cell RNA-seq and multiplex IHC staining assays reveal TLS-related cell components in NPC, including B cells, Tfh, CXCL13⁺ CAFs, and CXCL13⁺CD8⁺ T cells. Second, we focus on B cells, revealing their differentiation trajectories and clinical relevance in NPC. Third, along with the identification of novel cell clusters (CXCL13⁺ CAFs and CXCL13⁺CD8⁺ T cell), we examine their interactions with B cells. Additionally, co-culture assays and antibody neutralization experiments corroborate the genuine interaction of CXCL13⁺ CAFs with B cells through CXCL13 and TNFSF13B, contributing to the adhesion, recruitment, and promotion of B cells. Lastly, the application of the TLS-related cell cluster signature to clinical contexts, validated across multiple cohorts, has demonstrated its capacity to predict the survival and immunotherapeutic outcomes of NPC patients.

REVIEWER COMMENTS

Reviewer #1 (Remarks to the Author):

In this revised manuscript, Liu et al. clarified many of the reviewers' questions. There are some minor outstanding comments:

1. The authors used the terms TLS (tertiary lymphoid structures) and TCA (tumor cell aggregates) without defining them upfront, a task which should be done.
2. For the 12 NPC patients whose FFPE samples were tested with Visium (Supplemental Table 1), it is unclear whether they were all primary diagnostic samples (before any treatment) versus recurrent/metastatic tumor samples obtained immediately pre toripalimab plus chemotherapy treatment? This group appears to be quite different from the scRNA-seq cohort as the latter were all treatment naïve. The time interval between the FFPE samples being tested with Visium and the immunotherapy-based treatment for these 12 patients would be relevant to clarify.
3. Page 9, lines 19-21, Suppl Figure 6K, Suppl Figure 9c-f: There were only 2 cases of gastric cancers (so presumably one is EBV+ and the other is EBV-), thus any observations made based on these 2 cases must be cautioned.
4. Page 17, lines 14-18, Figure 8C: The survival analyses based on high versus low TLS-CS should be adjusted by disease stage, as that can be a confounding factor.

Reviewer #2 (Remarks to the Author):

The authors have conducted considerable revisions in response to the reviewers' comments. However, there are still some concerns that need to be addressed:

- 1-In the Response Figure 9, the IHC staining for markers is not perfect. For instance, there is significant overlap between CD3 and CD20, which are typically expressed on two different cell types.

2-While it is understood that functional characterization of CXCL13+CD8+ T cells in NPC remains challenging, IHC corroborating cell-cell communication analysis is necessary to validate the interactions of CXCL13+CD8+ T cells and malignant cells, via CXCR3-CCL20, CXCR3-CXCL10, CXCR3-CCL19, and CD27-CD70 interactions.

3-The staining for CXCL13 in Figure 7B is not convincing. There appears to be a high background, making it very difficult to identify real positive cells. Additionally, quantification of images for all examined patients is necessary to provide the reader with a better understanding of their frequency and potential importance.

4-It is advisable to avoid using “novel” for cell populations such as CXCL13+CD8+ T cells and CXCL13+ CAFs, as they have already been described in other publications referenced in the manuscript. While their transcriptome profile might differ in this study, it is not sufficient reason to classify them as novel.

5-The authors mentioned that CXCL13+CD8+ T cells constitute a minor fraction of the overall CD8+ T cell population (ranging from 0% to 9.89%, with a median of 1.42%). This information should be included in the manuscript along with appropriate supporting data, and their importance despite their low frequency in the TME should be discussed.

Reviewer #3 (Remarks to the Author):

In the revised manuscript, the authors have partially addressed my comments.

Nevertheless, they should further clarify the following points:

1. The authors should address the problems of heterogenous tumor contents in the Bulk RNA-seq, microarray and Single-cell RNA-seq data in the determination of association of GC-B cells proportion with NPC status (Figure 2b). They should show whether there are significant differences in the content of tumor cells, T-lymphocytes and total B-lymphocytes between the early and advanced NPC groups. If yes, they should tone down their conclusion of the specific association of GC-B proportion with NPC status. In Figure S15A, the status of NPC (early vs advance) in each tumor should be shown. For the single-cell RNA-seq, it is

noted that more than 1/3 of the cases are no tumor cells and 1/3 of the cases contain few tumor cells. Nevertheless, the high variety of the tumor quality might affect the interpretation of the results in Figure 2b. For supplementary Figure S15f, the authors should also conduct the CD3 and panCK staining/EBERs assay to determine and indicate the portion of the infiltrating lymphocytes and tumor cells and their correlation with the presence of TLS, tumor status (early vs advance) and OS.

2. For the FFPE NPC samples of Visium cohort (ST5-19), the EBV-high and -low status were defined indirectly by the 100DEGs genes, the accuracy of the method is questionable as they are not directly detected the EBV gene expression or the EBV copy number. They should clearly describe the limitations before they presented the results. For the fresh-frozen (FF) NPC samples of the stereo-seq cohort (ST2, ST3), the results of Figure 3 and in Figure S9a are not consistent. The regions/spots with high RPMS1/A73 in ST3/ST2 were also defined as EBV-low? The data of the Samples ST1 should also be shown. If the probe for detection of EBER2(or EBER1) is poly-A dependent, the data of EBER2 (or EBER1) should be removed. Why relatively high lytic genes (e.g. BARF0 and LF2) were detected in the NPC samples? It is possible that there are problems in EBV gene annotation as RPMS1/A73 are overlapped with the BARF0 and LF2 genome regions, but in different DNA stands. In FigureS9c, the author used the similar Yellow color to show the EBV-high-TCA-wP and EBV-high-TCA-woP group in the figure, the relationship between these two groups cannot be recognized! In Figure9e, it is noted that the pattern of expression of all EBV lytic (BARF0, BALF3/4) and latent genes (EBER1, EBER2, EBNA's?, LMP-1/BNLF2a/b) were almost identical and with the similar expression level in the case M39, M55. The results suggest the occurrence of errors during analysis. Why there are high expression of BALF3/4 and BARF0 lytic genes in the latent EBV-infected GC cells? EBVaGC should express abundant RPMS1 and A73, why these major EBV latent gene products were not shown in the figures. Again, EBER1/EBER2 detection in the samples M39 and M55 is poly-A dependent, please remove it from the analysis since EBERs are non-poly-A RNAs. Similar problems also occur in the analysis of the microdissection cohort in figure S9g and h. The authors did not address the problems in the detection of abundant lytic genes in the NPC samples. Overall, in this study, EBV-status (high vs low) were not accurately defined in the NPC/GC samples using the stereo and visium methods.

3. In all information analysis of single cell sequencing, bulk-RNA sequencing, Stereo-seq and Visium cohorts using the poly-A captured methods, the data of EBER1/EBER2 should be removed from the calculation since these EBV-encoded lncRNAs are lack of poly-A tails.

4. For the validation of EBV-high and -low samples, the authors should conduct the IHC of BZLF1 and EA-D to confirm the present of NPC cells with lytic induction. If possible, they should examine both EBER and RPMS1/A73 in the FFPE samples by in-situ hybridization analysis.

Reviewer #5 (Remarks to the Author):

All my concerns were addressed to a reasonable degree.

To better understand the rationale behind the relevance of small quantitative differences between GC B cells, please include a summary of explanation given in this rebuttal in the discussion of the main manuscript.

Reviewer #6 (Remarks to the Author):

The authors have provided detailed answers to my questions and improved their manuscript by adding new experiments and analysis. The additional data make the paper and the findings stronger.

Minor comments:

Please rephrase the following highlights of the manuscript:

- GC reaction is LINKED TO the differentiation of plasma cells producing antitumour antibodies from TLS to TCA with heterogeneous juxtapositions in NPC
- The antibodies ARE LINKED TO the apoptosis of EBVhigh malignant cells in EBV-related epithelial tumours and are associated with immunotherapy response in NPC
- In TLS CXCL13+ CAFs promote the adhesion and antibody production of B cells, which ARE ASSOCIATED WITH the stem-like CXCL13+CD8+ T cells

Please also tone down these conclusions throughout the manuscript.

Please tone down the following sentence: "These data suggest that EBV+

13 NPC malignant cells may actively recruit plasma cells through the CCL2-CCR2 axis,
14 forming a niche with rich plasma cells in EBV+ TCA."

Regarding Fig. 5D, please provide the H&E staining. It is hard to see if this is a TLS or a blood vessel.

Regarding Fig. 7B, please provide the H&E staining (I am not sure if this is a TLS) and use the name of the protein (TIM3), instead of the gene (HAVCR2).

Reviewer #1 (Remarks to the Author):

In this revised manuscript, Liu et al. clarified many of the reviewers' questions. There are some minor outstanding comments:

1. The authors used the terms TLS (tertiary lymphoid structures) and TCA (tumor cell aggregates) without defining them upfront, a task which should be done.

Response: Thanks for the suggestion. In this Revision, we have added the definitions of TLS (tertiary lymphoid structures) and TCA (tumour cell aggregates) at their first appearance in each relevant section in the main text. Please kindly review these on Page 4, Line 5 and Page 7, Line 5.

2. For the 12 NPC patients whose FFPE samples were tested with Visium (Supplemental Table 1), it is unclear whether they were all primary diagnostic samples (before any treatment) versus recurrent/metastatic tumor samples obtained immediately pre toripalimab plus chemotherapy treatment? This group appears to be quite different from the scRNA-seq cohort as the latter were all treatment naïve. The time interval between the FFPE samples being tested with Visium and the immunotherapy-based treatment for these 12 patients would be relevant to clarify.

Response: Thanks for the suggestion. The NPC patients whose FFPE samples tested with Visium assay were all primary diagnostic or treatment naïve samples, similar to those used for the Stereo-seq platform. To clarify better in this Revision, we have provided additional details such as the disease status at sampling, subsequent therapeutic strategy, and chemotherapy regimens in the **Supplementary Table 1** (reproduced as below **Response Table 1**).

Tissue	TNM stage*	Number of Spots per Tissue	Median Genes per Spot	Mean Reads per Spot	ICB	Platform	Disease status at sampling	Therapeutic strategy	Chemotherapy regimens
NPC	II	6,053	4,643	NA	NA	Stereo-seq	treatment-naïve	NA	NA
NPC	IVa	1,722	3,058	NA	NA	Stereo-seq	treatment-naïve	NA	NA
NPC	I	1,788	2,226	NA	NA	Stereo-seq	treatment-naïve	NA	NA
NPC	IVa	2,021	5,100	114,480	CR	Visium	treatment-naïve	toripalimab plus chemotherapy	Capecitabine
NPC	IVb	2,614	1,859	59,441	CR	Visium	treatment-naïve	toripalimab plus chemotherapy	Paclitaxel + Cisplatin + 5-Fluorouracil
NPC	IVa	1,939	5,626	91,934	PD	Visium	treatment-naïve	toripalimab plus chemotherapy	Genecitabine + Arlotinib
NPC	IVb	1,209	2,854	151,387	CR	Visium	treatment-naïve	toripalimab plus chemotherapy	Paclitaxel + Cisplatin + 5-Fluorouracil
NPC	III	2,602	1,118	74,173	PD	Visium	treatment-naïve	toripalimab plus chemotherapy	Tegafur, Gimeracil and Oteracil Potassium
NPC	IVa	1,309	2,625	91,293	PR	Visium	treatment-naïve	toripalimab plus chemotherapy	Genecitabine + Cisplatin
NPC	IVb	1,272	2,500	90,144	CR	Visium	treatment-naïve	toripalimab plus chemotherapy	Genecitabine + Cisplatin
NPC	IVa	3,869	6,947	76,081	CR	Visium	treatment-naïve	toripalimab plus chemotherapy	Paclitaxel + Tegafur, Gimeracil and Oteracil Potassium
NPC	IVb	1,556	8,502	104,781	PD	Visium	treatment-naïve	toripalimab plus chemotherapy	Paclitaxel + Carboplatin
NPC	IVb	2,067	2,171	82,985	SD	Visium	treatment-naïve	toripalimab plus chemotherapy	Tegafur, Gimeracil and Oteracil Potassium
NPC	II	1,977	1,695	97,960	PR	Visium	treatment-naïve	toripalimab plus chemotherapy	Docetaxel
NPC	IVb	1,086	7,288	129,897	PD	Visium	treatment-naïve	toripalimab plus chemotherapy	Genecitabine
GaC	IVb	3,280	1,812	156,942	NA	Visium	treatment-naïve	NA	NA
GaC	II	4,805	2,637	71,276	NA	Visium	treatment-naïve	NA	NA

Response Table 1: Spatial transcriptomics summary and treatment information of NPC and EBV positive GaC samples. GaC, Gastric adenocarcinoma; NA, Not available; CR, Complete response; PD, Progressive disease; PR, Partial response; SD, Stable disease.

3. Page 9, lines 19-21, Suppl Figure 6K, Suppl Figure 9c-f: There were only 2 cases of gastric cancers (so presumably one is EBV+ and the other is EBV-), thus any observations made based on these 2 cases must be cautioned.

Response: Thanks for the comment. In our study, we included 277 TCGA gastric cancer (GaC) samples with bulk-seq data and 20 in-house GaC samples with scRNA-seq data, containing both EBV+ and EBV- samples. Deconvolution analysis of the bulk-RNA-seq data and scRNA-seq analysis indicate a higher proportion of GC B cells in EBV+ GaC compared to EBV- GaC samples. We have clarified this point in the main text (Page 9, Lines 20-22) and the figure legend for the **Supplementary Figure 6k**.

Supplementary Figures 9c-f present the spatial transcriptomic features for two GaC

cases that were both EBV-positive as indicated in the **Supplementary Table 1**. These data demonstrate that the spatial interactions between malignant cells and plasma cells in GaC are consistent with those observed in NPC, although the sample size is limited.

4. Page 17, lines 14-18, Figure 8C: The survival analyses based on high versus low TLS-CS should be adjusted by disease stage, as that can be a confounding factor.

Response: Thanks for the suggestion. Using the microarray and bulk RNA-seq cohort data, which include stage information, we first conducted a univariate Cox regression analysis, revealing that TLS is associated with OS and PFS (**Response Table 2**). We further performed multivariate Cox regression analysis, adjusting for disease stage as suggested. This analysis reveals that TLS signature is an independent prognostic indicator for NPC survival (**Response Table 2**). In this Revision, we have included these results in **Supplementary Table 6**. Please kindly review these on Page 17, Lines 28-29.

Variable	Univariate analysis		Multivariate analysis		Survival	Number of samples	Cohort
	Hazard Ratio (95% CI)	P value	Hazard Ratio (95% CI)	P value			
TLS	0.19 (0.09-0.41)	0.00002	0.19 (0.088-0.4)	0.000018	OS	150	Microarray
TLS	0.32 (0.17-0.59)	0.00033	0.19 (0.09-0.41)	0.000321	PFS	150	Microarray
TLS	0.34 (0.099-1.2)	0.092	0.28 (0.078-1)	0.05	PFS	59	Bulk-RNA-seq

Response Table 2: Prognostic value of TLS in the microarray and Bulk-RNA-seq NPC sample collections. The table showing hazard ratio and confidence interval derived from Cox regression survival analyses for progression-free survival (PFS) and overall survival (OS) with either univariate model or multivariable model adjusted for disease stage; the corresponding Cox regression P values are also shown.

Reviewer #2 (Remarks to the Author):

The authors have conducted considerable revisions in response to the reviewers' comments. However, there are still some concerns that need to be addressed:

1-In the Response Figure 9, the IHC staining for markers is not perfect. For instance, there is significant overlap between CD3 and CD20, which are typically expressed on two different cell types.

Response: Thanks for the comment. We fully agree with the Reviewer's point that CD3 and CD20 are markers specifically expressed on T cells and B cells, respectively. When reviewing the previous images, we realized that the high fluorescence intensity resulted in a seeming overlap. In this Revision, we have adjusted the intensity settings (**Response Figure 1**) and updated the main **Figure 1a** and **Supplementary Figure 1a**.

Response Figure 1: Multiplex IHC staining of T and B cells in NPC tissue biopsy. a and b are two different samples. Cells were coloured according to their staining with CD3 (green), CD20 (red), and DAPI (blue) proteins as indicated on top. Images are representative of three biological replicates. Scale bars are 30µm or 50µm, respectively.

2-While it is understood that functional characterization of CXCL13+CD8+ T cells in NPC remains challenging, IHC corroborating cell-cell communication analysis is necessary to validate the interactions of CXCL13+CD8+ T cells and malignant cells, via CXCR3-CCL20, CXCR3-CXCL10, CXCR3-CCL19, and CD27-CD70 interactions.

Response: Thanks for the suggestion. Given the well-established interactions between CXCR3 on CD8+ T cells and CXCL10, CCL19, and CCL20 on epithelial cells in tumour microenvironment^{1, 2}, we performed multiplex IHC staining assay to validate the interaction between CXCL13+CD8+T cells and malignant cells via CD27-CD70 in additional NPC tissue

biopsies. These assays demonstrated the juxtaposition of CD70-expressing malignant cells (CD70+PanCK+) and CD27-expressing CXCL13+CD8+ T cells (CD27+CXCL13+CD8+) in NPC tissue samples (**Response Figure 2**), indicating their spatial interaction. Please kindly review these updated details in the Results section on Page 16, Lines 11-16 and in the new **Supplementary Figure 13g**.

Response Figure 2: Representative images of multiplex IHC staining for the juxtaposition of CD70-expressing malignant cells (PanCK+) and CD27-expressing CXCL13+CD8+ T cells (CXCL13+CD8+) in NPC tissue samples. Proteins detected using respective antibodies are indicated on top. The purple, red, green, yellow, and white arrows indicated positive cells with the expression of CD8, CXCL13, CD27, CD70, and PanCK proteins in NPC tissue, respectively (bottom panel). Images are representative of three biological replicates. Scale bars are 20µm or 40µm.

3-The staining for CXCL13 in Figure 7B is not convincing. There appears to be a high background, making it very difficult to identify real positive cells. Additionally, quantification of images for all examined patients is necessary to provide the reader with a better understanding of their frequency and potential importance.

Response: Thanks for the comments. In this Revision, we have conducted additional IHC staining with new antibodies, which resulted in more specific staining for CXCL13+CD8+ T cells and reduced background noise (**Response Figure 3**). Furthermore, we have conducted quantitative analysis on the staining results. The proportion of CXCL13+CD8+ T cells ranges from 0.5% to 11%, which is consistent with the results from our scRNA-seq analysis. We have included this new information in the main text. Please kindly review these updates in the Results section on Page 16, Lines 15-18 and in the **Supplementary Figure 13h** (reproduced as below **Response Figure 3**).

Response Figure 3: a. Multiplex IHC staining of CXCL13 in NPC tissue biopsy. Cells were coloured according to their staining with CXCL13 (red) protein. Images are representative of three biological replicates. Scale bar is 50µm. b. Representative images of multiplex immunofluorescence staining of CXCL13+CD8+ T cells in NPC tissues. The yellow, red, and orange arrows indicated the representative cells positive for CD8, CXCL13, and co-expression of CD8 and CXCL13 proteins in T cells, respectively. Scale bars, 50µm. c. Bar plot showing the percentage of CXCL13+CD8+ T cells for multiplex IHC cohort.

4-It is advisable to avoid using “novel” for cell populations such as CXCL13+CD8+ T cells and CXCL13+ CAFs, as they have already been described in other publications referenced in the manuscript. While their transcriptome profile might differ in this study, it is not sufficient reason to classify them as novel.

Response: Thanks for the suggestion. In this revised manuscript, we have toned down the descriptions regarding the novel discovery of CXCL13+CD8+ T cells and CXCL13+ CAFs in the main text.

5-The authors mentioned that CXCL13+CD8+ T cells constitute a minor fraction of the overall CD8+ T cell population (ranging from 0% to 9.89%, with a median of 1.42%). This information should be included in the manuscript along with appropriate supporting data, and their importance despite their low frequency in the TME should be discussed.

Response: Thanks for the suggestion. In this Revision, we have included the proportion data for CXCL13+CD8+T cells in the Results section and the **Supplementary Figure 13h**

(reproduced as below **Response Figure 4**). Please kindly review the updated details in the Results section on Page 16, Lines 15-18. Additionally, we also discussed the importance of this T cell subset in the TME in the Discussion section. Please kindly review this content in the Discussion section on Page 22, Lines 2-25.

Response Figure 4: Bar plot showing the percentage of CXCL13+CD8+ T cells in the scRNA-seq NPC samples.

Reviewer #3 (Remarks to the Author):

In the revised manuscript, the authors have partially addressed my comments. Nevertheless, they should further clarify the following points:

1. The authors should address the problems of heterogenous tumor contents in the Bulk RNA-seq, microarray and Single-cell RNA-seq data in the determination of association of GC-B cells proportion with NPC status (Figure 2b). They should show whether there are significant differences in the content of tumor cells, T-lymphocytes and total B-lymphocytes between the early and advanced NPC groups. If yes, they should tone down their conclusion of the specific association of GC-B proportion with NPC status. In Figure S15A, the status of NPC (early vs advance) in each tumor should be shown. For the single-cell RNA-seq, it is noted that more than 1/3 of the cases are no tumor cells and 1/3 of the cases contain few tumor cells. Nevertheless, the high variety of the tumor quality might affect the interpretation of the results in Figure 2b. For supplementary Figure S15f, the authors should also conduct the CD3 and panCK staining/EBERs assay to determine and indicate the portion of the infiltrating lymphocytes and tumor cells and their correlation with the presence of TLS, tumor status (early vs advance) and OS.

Response: Thanks for the suggestions. In this Revision, we have analysed and compared the proportions of tumour cells, total T-lymphocytes, and total B-lymphocytes between the early and advanced NPC groups, revealing no significant differences (**Supplementary Figure 15a**; reproduced as below **Response Figure 5**). These results underscore the robust association of GC-B proportion with NPC status. It's true that the numbers of tumour cells can vary among individuals, with a significant proportion of samples displaying no tumour cells in the scRNA-seq data. In contrast, each sample contains a sufficient total cell count, predominantly immune and stromal cells. This is likely due to the fragile nature of malignant cells during the microfluidic partitioning for single-cell capture and subsequent processing, rather than a reflection of tumour quality.

For the **Supplementary Figure 15a**, we have indicated the disease status of NPC in each sample. Furthermore, we have conducted additional multiplex IHC staining assays for CD3, PanCK, and CD20 (**Response Figure 6a**). These assays demonstrated a significant difference in the proportion of infiltrating immune cells within the tumour cell aggregates (TCA) region, reflected by the immune cell infiltration scores (percentage of immune cell infiltration), between early and advanced stages of tumours (**Response Figure 6b**). Moreover, NPC tumours with TLS exhibited greater immune cell infiltration and NPC patients with high immune infiltration had better overall survival, although these results do not reach statistical significance, likely due to the limited sample size (**Response Figure 6c**). Please kindly review these updates in the Results section on Page 18, Lines 5-12 and in the **Supplementary Figures 16a-d**.

Response Figure 5: Proportions of cell types in NPC tumours and their associations with NPC stages. a. Bar plots showing the cell proportions (y-axis) of each cell type for each sample (x-axis) in Bulk RNA-seq (top panel), Microarray (middle panel), and Single-cell RNA-seq (bottom panel) cohorts. Cell types are colour-coded and indicated on the right. b. Box plots showing the tumour, T, and B and cell signatures in samples from the Bulk RNA-seq (top panel), Microarray (middle panel), and Single-cell RNA-seq (bottom panel) cohorts. P values are derived from two-sided student t-tests.

Response Figure 6: Immune cell infiltration in tumour cell aggregate (TCA) and their clinical relevance. a. Multiplex IHC staining of T, B, and malignant cells in NPC tissue biopsy. Cells were coloured according to their staining with CD3 (yellow), CD20 (red), and PanCK (cyan) proteins as indicated on top. Images are representative of three biological replicates. Scale bar is 240µm. b. Box plots showing the immune cell infiltration scores in samples with or without TLS (left panel) and at advanced or early stage (right panel). c. Kaplan-Meier survival curves of NPC cohorts with patients stratified by the intensity of immune cell infiltration in TCA. Survival duration and probability are indicated at the x- and y-axis, respectively. P value and HR were calculated using a two-sided cox test. OS, overall survival.

2. For the FFPE NPC samples of Visium cohort (ST5-19), the EBV-high and -low status were defined indirectly by the 100DEGs genes, the accuracy of the method is questionable as they are not directly detected the EBV gene expression or the EBV copy number. They should clearly describe the limitations before they presented the results.

Response: Thanks for the comment. The Visium assay for FFPE sample is based on probe capturing for human genes, including no probes for EBV genes. To assess the EBV infection status for the NPC FFPE samples tested by Visium, we proposed a set of DEGs between EBV-high and -low tumour cells as a surrogate signature for EBV infection status. Both scRNA-seq and Stereo-seq technologies, which rely on poly-A tail capture sequencing, allow for the detection of EBV-encoded genes. We identified the top 100 DEGs by comparing the EBV-high and -low tumour cells using the scRNA-seq data, and observed a high correlation between the

expression of the top 100 DEGs and EBV-encoded genes (**Response Figure 7a**). Moreover, validated the correlation between the top 100 DEGs signature and EBV infection status using the Stereo-seq cohort data (**Response Figure 7b**). These results strongly support the use of the top 100 DEGs signature as a robust surrogate for EBV infection status.

In this Revision, we have clarified that “We further evaluated the EBV infection status in the two ST cohorts, by using the top 100 DEGs between EBV-high and -low tumour cells derived from scRNA-seq data as a surrogate signature for the Visium FFPE cohort that does not capture EBV genes and using directly EBV gene expression for the Stereo-seq cohort”. Please kindly review this update in the Results section on Page 10, Lines 11-14. Additionally, we have acknowledged that “the 10x Visium platform for FFPE sample relies on probe capture of RNA molecules and does not include probes for EBV genes. Although we employed cross-validation approaches using the stereo-seq platform and EBER staining to verify EBV infection status, there are inherent limitations in accurately defining the EBV infection status solely with the top 100 DEGs signature”. Please kindly review the Discussion section on Page 23, Lines 15-19.

Response Figure 7: Scatter plots showing the pair-wise correlation between EBV-related tumour cell signature and EBV-encoded molecules in NPC scRNA-seq (a) and stereo-seq (b) cohorts. The R values represented Pearson’s correlation. The P values were calculated by two-sided Pearson correlation analysis.

For the fresh-frozen (FF) NPC samples of the stereo-seq cohort (ST2, ST3), the results of

Figure 3 and in Figure S9a are not consistent. The regions/spots with high RPMS1/A73 in ST3/ST2 were also defined as EBV-low? The data of the Samples ST1 should also be shown.

Response: Thanks for the feedback. In the previous version of our manuscript, the regions/spots with high expression of EBV-encoded genes in Figure S9a were identified as EBV-high-TCA regions in Figure 3. However, we realized that the colour-coding in these figures might have caused some confusion, as blue represented EBV-high regions in Figure 3 but indicated low EBV expression in Figure S9a. In this revised manuscript, we have adjusted the colour-coding in Figure S9a to improve clarity, using yellow for EBV high and blue for EBV low regions; additionally, we included a column of the sections showing EBV-status to facilitate a more intuitive comparison (**Supplementary Figure 9a**, reproduced as below **Response Figure 8**).

Moreover, we have reclassified EBV-high and -low regions based on the presence or absence of detected EBV molecules, moving away from using the top 100 DEGs in the previous manuscript. This change enhances the reliability of our results while has no impact on the previous conclusions. Additionally, following the Reviewer's suggestion, we have included the data for the ST1 sample. Please kindly review the updated changes in **Supplementary Figure 9a**.

Response Figure 8: Representative images of spatial feature plots showing the different TCA and expression levels of EBV molecules in NPC tumours in the Stereo-seq cohorts. Regions were grouped according to EBV status in TCA (left). EBV-encoded genes are indicated on top. Each type of TCA is coloured as indicated. Filled colours from blue to red represent scaled expression levels from low to high.

If the probe for detection of EBER2(or EBER1) is poly-A dependent, the data of EBER2 (or EBER1) should be removed. Why relatively high lytic genes (e.g. BARF0 and LF2) were detected in the NPC samples? It is possible that there are problems in EBV gene annotation as RPMS1/A73 are overlapped with the BARF0 and LF2 genome regions, but in different DNA stands.

Response: Thanks for the comments. As suggested, we have removed EBER2/1 data from the relevant analyses. It's important to note that the exclusion of EBER2/1 data did not

significantly impact the original conclusions.

Previous study using Smart-seq2 technology³ has reported simultaneous expression of lytic genes (such as BARF0 and LF2) and latent genes in some NPC tumour cells. Our Stereo-seq data revealed co-expression of lytic genes and latent genes at certain spots, while other spots showed high expression of latent genes alone (**Response Figure 9a**). Furthermore, our bulk RNA-seq data indicated that some patients exhibited simultaneous expression of both type of genes, whereas others showed high expression of only latent genes (**Response Figure 9b**). Together, these results underscore the heterogeneity of EBV infection status at both the cellular and individual levels.

Additionally, the algorithms used in Cell Ranger Count are specifically designed to identify genes transcribed from both sense and antisense strands (**Response Figure 9c**). Therefore, the coexistence of both lytic and latent genes may reflect the complex nature of EBV infection in NPC development, rather than issues with gene annotation.

Response Figure 9: a. Heatmap showing the expression levels of EBV-encoded molecules (y-axis) for each spot (x-axis) in NPC stereo-seq cohort. b. Heatmap showing the expression levels of EBV-encoded molecules (y-axis) for each patient (x-axis) in NPC bulk RNA-seq cohort. c. Box plots showing the expression levels of genes in 10x scRNA-seq data. Orange indicates genes located on the sense strand and deep blue indicates genes located on the antisense strand.

In FigureS9c, the author used the similar Yellow color to show the EBV-high-TCA-wP and EBV-high-TCA-woP group in the figure, the relationship between these two groups cannot be recognized!

Response: Thanks for pointing out this ambiguous color-coding. In this Revision, we have used high-contrast colours to clearly distinguish between the EBV-high-TCA-wP and EBV-high-TCA-woP groups. Please kindly review these updates in the main **Figures 3a-b** and the **Supplementary Figure 9c**.

In Figure9e, it is noted that the pattern of expression of all EBV lytic (BARF0, BALF3/4) and latent genes (EBER1, EBER2, EBNA's, LMP-1/BNLF2a/b) were almost identical and with the similar expression level in the case M39, M55. The results suggest the occurrence of errors during analysis.

Response: Thanks for the comment. This mentioned apparent similarity may be due to the dot plot representation at a lower resolution. Using the same spatial transcriptome data, we have now used heatmap to present the expression patterns of the EBV lytic genes (BARF0, BALF3/4) and latent genes (EBER1, EBER2, EBNA's, and LMP-1/BNLF2a/b) in EBV+ GaC cells from samples M39 and M55 (**Response Figure 10**). This analysis reveals distinct expression levels and patterns for these EBV lytic and latent genes in the GaC samples (**Response Figure 10**). Please note that we have removed the results related to EBER1 and EBER2 as suggested by this Reviewer #3. Please kindly review these updates in **Supplementary Figure 9**.

Response Figure 10: Heatmap showing the expression levels of EBV-encoded molecules (y-axis) for each spot (x-axis) in EBV+ GaC ST cohort.

Why there are high expression of BALF3/4 and BARF0 lytic genes in the latent EBV-infected GC cells? EBVaGC should express abundant RPMS1 and A73, why these major EBV latent gene products were not shown in the figures. Again, EBER1/EBER2 detection in the samples M39 and M55 is poly-A dependent, please remove it from the analysis since EBERs are non-poly-A RNAs. Similar problems also occur in the analysis of the microdissection cohort in figure

S9g and h. The authors did not address the problems in the detection of abundant lytic genes in the NPC samples. Overall, in this study, EBV-status (high vs low) were not accurately defined in the NPC/GC samples using the stereo and visium methods.

Response: Thanks for the comments. The observed high expression of the lytic genes BALF3/4 and BARF0 in the latent EBV-infected GaC cells from samples M39 and M55 may reflect a unique EBV infection status. Given that these two GaC samples have corresponding scRNA-seq data, we compared their EBV gene expression levels (specifically RPMS1 and A73) with those in other EBV+ GaC samples containing scRNA-seq data. We noted lower levels of RPMS1 and A73 in M39 and M55 compared to other samples (**Response Figure 11**), highlighting the heterogeneity of EBV infection status in EBV+ GaC.

Following Reviewer #3's suggestion, we excluded EBER1 and EBER2 data in all relevant analyses for the scRNA-seq, microdissection, spatial transcriptome cohorts that rely on poly-T capturing technologies. Please note that we only had access to processed data for the microdissection cohort⁴.

Regarding the detection of lytic genes in the NPC samples, please refer to the response to this Reviewer's previous comment (**Response Figure 9**). Given that multiple studies have reported the simultaneously expression of EBV lytic and latent genes in tumour cells^{3, 5, 6}, we suggest that EBV infection status exhibits variability at both cellular and individual levels.

In our study, we determined the EBV status (high vs. low) in the NPC/GaC samples based on the expression of EBV molecules for the Stereo and Visium cohorts. Our findings were corroborated using the top 100 DEGs signature, microdissection cohort, and EBER staining, accurately reflecting the spatial heterogeneity of EBV infection status in tumour cells. Please refer to the response to this Reviewer's point 2 and kindly review the updated data presented in the main **Figure 3** and **Supplementary Figures 8-10**.

Response Figure 11: Bar plot showing the expression levels of RPMS1/A73 for EBV+ GaC samples (unpublished data)

3. In all information analysis of single cell sequencing, bulk-RNA sequencing, Stereo-seq and Visium cohorts using the poly-A captured methods, the data of EBER1/EBER2 should be removed from the calculation since these EBV-encoded LncRNAs are lack of poly-A tails.

Response: Thanks for the suggestion. We have repeated the analyses after removing

EBER1/2 and have updated all results accordingly. Please note that the exclusion of EBER1/2 does not affect our overall conclusions. Please kindly review the updated data in **Supplementary Figure 9**.

4. For the validation of EBV-high and -low samples, the authors should conduct the IHC of BZLF1 and EA-D to confirm the present of NPC cells with lytic induction. If possible, they should examine both EBER and RPMS1/A73 in the FFPE samples by in-situ hybridization analysis.

Response: Thanks for the suggestion. Indeed, a previous study has reported the co-expression of lytic (BZLF1, BRLF1, BMRF1 and BLLF1) and latent (LMP1, ENBA1, and EBNA2) genes in NPC cells⁷. In our study, we aimed to define EBV-high and -low states based on the comprehensive expression of EBV-encoded molecules, irrespective of whether they are lytic or latent genes. We established EBV-high and -low states based on the expression of EBV molecules in the Stereo and FF Visium cohorts and validated these classifications using the top 100 DEGs signature, bulk RNA-seq, and EBER staining. Although FFPE Visium technology has limitations in directly detecting EBV genes, the top 100 DEGs signature provided a viable alternative for assessing EBV-high and -low states.

Following the suggestion, we performed EBER staining on samples from the NPC ST cohort (FFPE Visium). We observed both EBV-high and -low states in these samples (**Response Figure 12**). Please note that these samples were consecutive tissue sections and may not completely match the sections analysed previously. We have acknowledged this limitation in the discussion section, read as “the 10x Visium platform for FFPE sample relies on probe capture of RNA molecules and does not include probes for EBV genes. Although we employed cross-validation approaches using the stereo-seq platform and EBER staining to verify EBV infection status, there are inherent limitations in accurately defining the EBV infection status solely with the top 100 DEGs signature”. Please kindly review the Discussion section on Page 23, Lines 15-19, and **Supplementary Figure 8b**.

Response Figure 12: Representative images of EBERs staining with different intensity (strong and weak) in NPC biopsy tissue samples. The dark brown arrows represent the EBV-high state, while the light brown arrows represent the EBV-low state. Indicated scale bars are 400 μ m and 240 μ m for the top panel and 200 μ m and 120 μ m for the bottom panel.

Reviewer #5 (Remarks to the Author):

All my concerns were addressed to a reasonable degree.

To better understand the rationale behind the relevance of small quantitative differences between GC B cells, please include a summary of explanation given in this rebuttal in the discussion of the main manuscript.

Response: Thanks for the suggestion. In this Revision, we have included a summary of explanation for the biological relevance of small quantitative differences in GC B cells in the Discussion section, read as "In our study, we observed a 70% decrease in the proportion of GC B cells from early to late-stage NPC. Despite their generally low abundance across various cancer types, GC cells exhibit a robust proliferative capacity, capable generating a substantial number of memory B cells and plasma cells. These cells play a pivotal role in modulating the tumour microenvironment through the production of antibodies, cytokines, and other factors". Please kindly review this added content in the Discussion section on Page 19, Lines 16-21.

Reviewer #6 (Remarks to the Author):

The authors have provided detailed answers to my questions and improved their manuscript by adding new experiments and analysis. The additional data make the paper and the findings stronger.

Response: Thanks for the favourable comments.

Minor comments:

Please rephrase the following highlights of the manuscript:

- GC reaction is LINKED TO the differentiation of plasma cells producing antitumour antibodies from TLS to TCA with heterogeneous juxtapositions in NPC
- The antibodies ARE LINKED TO the apoptosis of EBV^{high} malignant cells in EBV-related epithelial tumours and are associated with immunotherapy response in NPC
- In TLS CXCL13⁺ CAFs promote the adhesion and antibody production of B cells, which ARE ASSOCIATED WITH the stem-like CXCL13⁺CD8⁺ T cells

Please also tone down these conclusions throughout the manuscript.

Please tone down the following sentence: "These data suggest that EBV⁺ NPC malignant cells may actively recruit plasma cells through the CCL2-CCR2 axis, forming a niche with rich plasma cells in EBV⁺ TCA."

Response: Thanks for these suggestions. We have rephrased these highlights and toned down these conclusions throughout the manuscript following the suggestions. Please kindly review these updates in the Highlight section and the Results section on Page 11, Lines 15-18.

Regarding Fig. 5D, please provide the H&E staining. It is hard to see if this is a TLS or a blood vessel.

Response: Thanks for the suggestion. We have added a more representative TLS image and included the results of HE staining (**Response Figure 13**), showing a typical TLS. Please kindly review these in the main **Figure 5d**.

Multi-IF: CD20 FAP- α CXCR5 CXCL13

HE staining

Sample 1

Multi-IF: CD20 FAP- α CXCR5 CXCL13

HE staining

Sample 2

Response Figure 13: Multiplex IHC staining of CXCR5+ B cells and fibroblasts in NPC tissue biopsy. Cells were coloured according to their staining with CD20 (cyan), FAP- α (orange), and CXCR5 (red), and CXCL13 (red) proteins as indicated on top. The corresponding HE staining images are shown (bottom). Images are representative of three biological replicates. Scale bars are 500 μ m and 240 μ m as indicated.

Regarding Fig. 7B, please provide the H&E staining (I am not sure if this is a TLS) and use the name of the protein (TIM3), instead of the gene (HAVCR2).

Response: Thanks for the suggestion. We have added a more representative TLS image and included the results of HE staining (**Response Figure 14**). Please kindly review these in the main **Figure 7b**.

Response Figure 14: Multiplex IHC staining of CXCL13+CD8+ T cells in NPC tissue biopsy. Cells were coloured according to their staining with CD8 (purple), CXCL13 (red), CD20 (orange), PD1 (green), and TIM3 (yellow) proteins as indicated on top. The corresponding HE staining images are shown (bottom). Images are representative of three biological replicates. Scale bars are 500 μ m and 240 μ m as indicated.

Reference:

1. Lindner AK, *et al.* CXCR3 Expression Is Associated with Advanced Tumor Stage and Grade Influencing Survival after Surgery of Localised Renal Cell Carcinoma. *Cancers (Basel)* **15**, (2023).
2. Wang X, *et al.* The role of CXCR3 and its ligands in cancer. *Front Oncol* **12**, 1022688 (2022).
3. Jin S, *et al.* Single-cell transcriptomic analysis defines the interplay between tumor cells, viral infection, and the microenvironment in nasopharyngeal carcinoma. *Cell Res* **30**, 950-965 (2020).
4. Tay JK, *et al.* The microdissected gene expression landscape of nasopharyngeal cancer reveals vulnerabilities in FGF and noncanonical NF-kappaB signaling. *Sci Adv* **8**, eabh2445 (2022).
5. Liu Y, *et al.* Tumour heterogeneity and intercellular networks of nasopharyngeal carcinoma at single cell resolution. *Nat Commun* **12**, 741 (2021).
6. Li YQ, *et al.* Single-Cell Analysis Reveals Malignant Cells Reshape the Cellular Landscape and Foster an Immunosuppressive Microenvironment of Extranodal NK/T-Cell Lymphoma. *Adv Sci (Weinh)* **10**, e2303913 (2023).
7. Lin W, *et al.* Establishment and characterization of new tumor xenografts and cancer cell lines from EBV-positive nasopharyngeal carcinoma. *Nat Commun* **9**, 4663 (2018).

REVIEWERS' COMMENTS

Reviewer #1 (Remarks to the Author):

The authors have addressed all my concerns in an acceptable way.

Reviewer #2 (Remarks to the Author):

The authors have addressed my comments in this new version of the paper.

Reviewer #3 (Remarks to the Author):

For this revised version, the authors have addressed most of my concerns. Nevertheless, according to my comments on the problems of heterogeneous tumor contents in single-cell RNA-seq data and the authors' reply, there are "a significant proportion of samples displaying no tumour cells in the scRNA-seq data". Although the authors explained that "This is likely due to the fragile nature of malignant cells during the microfluidic partitioning for single-cell capture and subsequent processing, rather than a reflection of tumour quality." Since the single-cell RNA-seq data did not provide the evidence for the presence of tumor cells in these samples, the authors should exclude the cases "without any tumor cells" from their analysis. Furthermore, the criteria for the "High / Low immune cell infiltration" cases and the immune cell infiltration score shown in sFigure 16 should be defined. What is the immune cell infiltration score of the two cases shown in sFigure 16a? In Response Figure 7, what is the parameter for X axis? The definition of the parameter should also be clearly defined.

Reviewer #6 (Remarks to the Author):

After two rounds of revision, all my concerns were addressed properly.

REVIEWERS' COMMENTS

Reviewer #1 (Remarks to the Author):

The authors have addressed all my concerns in an acceptable way.

Response: Thanks.

Reviewer #2 (Remarks to the Author):

The authors have addressed my comments in this new version of the paper.

Response: Thanks.

Reviewer #3 (Remarks to the Author):

For this revised version, the authors have addressed most of my concerns. Nevertheless, according to my comments on the problems of heterogeneous tumor contents in single-cell RNA-seq data and the authors' reply, there are "a significant proportion of samples displaying no tumour cells in the scRNA-seq data". Although the authors explained that "This is likely due to the fragile nature of malignant cells during the microfluidic partitioning for single-cell capture and subsequent processing, rather than a reflection of tumour quality." Since the single-cell RNA-seq data did not provide the evidence for the presence of tumor cells in these samples, the authors should exclude the cases "without any tumor cells" from their analysis.

Response: Thanks for the comments. As noted by Reviewer #5, single-cell RNA sequencing (scRNA-seq) is a valid technology for profiling the tumor microenvironment (TME) composition, provided that findings from scRNA-seq are validated using other methods. In our study, the scRNA-seq data was derived from tumor samples with pathological confirmation^{1, 2, 3 4}. Although there is variation in tumor cell content observed in the scRNA-seq, our main findings, including essential TLS components and their interactions, are consistent across these samples. Moreover, we have validated our scRNA-seq findings using additional methods such as bulk-RNAseq, IHC, IF, and functional assays. As pointed out by Reviewer #5, microfluidic-based scRNA-seq may have technical limitations in processing simultaneously multiple cell types with variability in size, tractability, and fragility, leading to a capture bias that results in fewer tumour cells compared to immune and stromal cells.

In summary, we have retained all single-cell samples derived from tumours for analysis. Please refer to Reviewer 5's comments and Page 23, Lines 19-22 for details.

Furthermore, the criteria for the "High / Low immune cell infiltration" cases and the immune cell infiltration score shown in sFigure 16 should be defined. What is the immune cell infiltration score of the two cases shown in sFigure 16a? In Response Figure 7, what is the parameter for X axis? The definition of the parameter should also be clearly defined.

Response: Thanks for the comment. We defined immune cell infiltration scores of 1, 2, 3, and 4 as representing 0~25%, 26~50%, 51~75%, and 76~100% of the tumour area infiltrated by immune cells, respectively. For the two cases in sFigure 16a, the immune cell infiltration scores are 1 and 2. We then used ROC analysis to determine the optimal cut-off (1.5) for immune cell infiltration scores associated with survival. Cases with the scores above or below this cut-off are defined as having High or Low immune cell infiltration, respectively. In this revision, we have included these definitions in Method section, Page 23, Lines 19-22.

In Response Figure 7, the parameter for the X axis is the expression level (transcript per million, TPM) of the EBV-encoded molecules as indicated at the top. We have added this description in **Response Figure 1** in this revision.

Response Figure 1: Scatter plots showing the pair-wise correlation between EBV-related tumour cell signature and the expression levels of EBV-encoded molecules (TPM) in NPC scRNA-seq (a) and stereo-seq (b) cohorts. The R values represented Pearson's correlation. The P values were calculated by two-sided Pearson correlation analysis.

Reviewer's #5 comments:

Variation in tumor cell content is a common and expected outcome when using microfluidic systems to generate single-cell RNA-seq libraries from unselected samples. This variability may arise from the processing challenges associated with large, adhesive tumor cells in

microfluidic systems, as mentioned by the authors. Additionally, it is probable that in samples with a high proportion of immune cells, these cells are preferentially processed over larger and less tractable cell types. There is also the potential for loss of read-outs during the bioinformatics quality control phase, where tumor cells might not have the same mitochondrial gene and other QC cut-offs as other cell types.

These technical considerations could be addressed during the experimental design phase. One potential strategy is to process separate samples for immune and non-immune cells in different microfluidic lanes, as demonstrated by Kürten et al. *Nature Communications*, 2021. However, even this approach does not guarantee uniform recovery of cancer cells from each patient, as shown in Supplementary Figure 1C of Kürten et al. Achieving a balanced representation of cancer cells would require either an even more selective cell input (e.g., using only EPCAM+ cells) or employing alternative methodologies such as plate sorting, where each well represents a single-cell reaction (Puram et al., *Cell*, 2017).

Regarding whether samples with no tumor content should be excluded, this decision should be based on the conclusions drawn from the data and their validation methods. Single-cell RNA-seq is not suited for quantitative analyses of tumor microenvironments due to the reasons previously mentioned. However, its strength lies in providing a detailed transcriptomic profile at the single-cell level. Provided that conclusions derived from scRNA-seq data are corroborated by other methods such as bulk RNA-seq and immunohistochemistry, the use of scRNA-seq for exploratory analysis is a valid and intended application of this technology.

In conclusion, I do not believe that the absence of tumor cells necessitates the removal of samples for technical reasons. Although I believe I do not have access to the full manuscript in its latest version, it appears the authors have validated their scRNA-seq findings with additional methodologies. To address the reviewer #3 concerns fully, a more extensive discussion of the aforementioned technical issues may be beneficial.

Response: Thanks for the thoughtful comments. We fully agree with the points raised by Reviewer #5. Single-cell RNA sequencing (scRNA-seq) is a valid technology for exploratory analysis of TME composition. As in our study, findings from the scRNA-seq data are typically validated by other methods, such as bulk RNA-seq, IHC, and functional assays. In line with Reviewer #5's suggestions, we acknowledge that the technical limitations of microfluidic-based single-cell platforms may result in variation in tumor cell content. Please refer to Page 23, Lines 19-22 for further details.

Reviewer #6 (Remarks to the Author):

After two rounds of revision, all my concerns were addressed properly.

Response: Thanks.

1. Jin S, *et al.* Single-cell transcriptomic analysis defines the interplay between tumor cells, viral infection, and the microenvironment in nasopharyngeal carcinoma. *Cell Res* **30**, 950-

965 (2020).

2. Chen YP, *et al.* Single-cell transcriptomics reveals regulators underlying immune cell diversity and immune subtypes associated with prognosis in nasopharyngeal carcinoma. *Cell Res* **30**, 1024-1042 (2020).
3. Gong L, *et al.* Comprehensive single-cell sequencing reveals the stromal dynamics and tumor-specific characteristics in the microenvironment of nasopharyngeal carcinoma. *Nat Commun* **12**, 1540 (2021).
4. Liu Y, *et al.* Tumour heterogeneity and intercellular networks of nasopharyngeal carcinoma at single cell resolution. *Nat Commun* **12**, 741 (2021).